# Two subduction-related heterogeneities beneath the Eastern Alps and the Bohemian Massif imaged by high-resolution P-wave tomography

Jaroslava Plomerová[1], Helena Žlebčíková[1], György Hetényi[2], Luděk Vecsey[1], Vladislav Babuška[1,†] and AlpArray-EASI and AlpArray working groups[+]

[1] Institute of Geophysics, Czech Academy of Sciences, 141 31 Prague, Czech Republic
[2] Institute of Earth Sciences, University of Lausanne, 1015 Lausanne, Switzerland
† deceased
[+] For the complete team list visit the link which appears at the end of the paper

*Correspondence to*: Jaroslava Plomerová (jpl@ig.cas.cz)

**Abstract.** We present high-resolution tomographic images of the upper mantle beneath the E. Alps and the adjacent Bohemian Massif (BM) in the North based on data from the AlpArray-EASI and AlpArray Seismic Networks. The tomography locates the Alpine high-velocity perturbations between the Periadriatic Lineament and the Northern Alpine Front. The northward-dipping lithosphere keel is imaged down to ~200-250 km depth, without signs of delamination, and we associate it with the Adriatic plate subduction. Detached high-velocity heterogeneity, sub-parallel to and distinct from the E. Alps heterogeneity is imaged at ~100 - 200km depths beneath the southern part of the BM. We associate this heterogeneity with the western end of a SW-NE striking heterogeneity beneath the south-eastern part of the BM, imaged in models of larger extent. The strike, parallel with the Moldanubian/Brunovistulian mantle-lithosphere boundary in the BM and with the westernmost part of the Carpathian front, leads us to consider potential scenarios relating the heterogeneity to (1) a remnant of the delaminated European plate, (2) a piece of continental-and-oceanic lithosphere mixture related to the building of the BM, particularly to the closure of the old Rheic ocean during the MD/BV collision or (3) a lithospheric fragment going through to the NW between the E. Alps and W. Carpathians fronts in a preceding subduction phase. The study is dedicated to our outstanding and respected colleague Vladislav Babuška, who coined innovative views on the European lithosphere and died on March 30, 2021.

## 1 Introduction

Teleseismic body-wave tomography represents a powerful tool to study regional velocity structure of the upper mantle and to image velocity anomalies, particularly those related to subducted plates in collision zones. The Alps have developed at a collision zone of the Eurasian and Adriatic plate since the Variscan orogeny (Fig.1). The classical concept assumed the European lithospheric slab subducted south-eastward/southward along the entire Alps (Laubscher, 1970; Mueller, 1982)

without any segmentation. However, interactions of the European lithosphere with the translating and rotating Adriatic plate and several micro-plates involved in the collision, tearing and retreat of the slabs, resulted in the bended (arcuate) shape of the western Alpine mountain range on the surface and in the complicated geometry of the rigid lithosphere penetrating into the ductile mantle.

Early tomographic models of Europe (Aki et al., 1977, Spakman, 1990) have been replaced with more advanced models of segmented Alpine slab during last decades (Wortel and Spakman, 2000; Kissling et al., 2006; Malusa et al., 2021, for reviews). For the first time, Babuška et al. (1990) imaged the Alpine slabs separated into two segments, one beneath the Western and second beneath the Eastern Alps, with opposite  polarity and a gap in between them. The density of stations and teleseismic rays enabled to resolve the high-velocity heterogeneities only at 1.5° by 1.5° grid, but the bended shape of the south-eastward dipping subduction of the European plate in the Western Alps and steep northward dipping lithosphere beneath the Eastern Alps, with a gap between the two Alpine slabs, were evident. More recent tomographic studies from data recorded in regional passive seismic experiments with densely spaced stations resolved the Alpine subductions at finer grids and confirmed the suggested change in polarity between the distinct Western and Eastern Alpine lower lithosphere roots (Lippitsch et al., 2003; Mitterbauer et al., 2011; Karousová et al., 2013; Zhao et al.,2016; Hua et al, 2017). The standard isotropic regional velocity tomography (e.g., Piromallo and Morelli, 2003; Koulakov et al., 2009) based on pre-AlpArray data (Hetényi et al., 2018a), imaged the south-eastward dipping curved slab of the Eurasian lithosphere in the Western Alps and the northward dipping plate beneath the Eastern Alps of similar sizes (geometry), though interpret them differently. Dando et al. (2011) interpret high-velocity heterogeneities at the bottom of their regional tomographic model of the upper mantle beneath the Alpine-Pannonian region (Lombardi et al., 2009, Hetényi et al., 2009) as  a graveyard of relict downwelling of continental lithosphere..

While resolution of early tomography of the Alpine region allowed researchers to deal only with the large and most distinct heterogeneities in the upper mantle, accumulation of high-quality data from the dense AlpArray network motivated us to search finer images of the upper mantle and to search answers on segmentation of the Alps, dip directions of the subductions, their relevance to European or Adriatic plate, extent of slab delamination and particularly, to elucidate the smaller size heterogeneity beneath the Bohemian Massif north of the E. Alps subduction. We concentrate in this paper on imaging the European/Adria plate collision across the Eastern Alpine transect aiming at understanding orogen-forming processes and present high-resolution tomographic images of the upper mantle beneath the E. Alps and the adjacent Bohemian Massif (BM) in the North. We show that thanks to data from the AlpArray-EASI (Hetényi et al., 2018b) and AlpAray Seismic (AASN) networks (AlpArray Seismic Network, 2014; 2015; Hetényi et al., 2018a), the tomography localizes the western end of a high-velocity heterogeneity imaged at ~100 - 200km depths beneath the south-eastern part of the BM (referred to as HV-BM throughout the paper), sub-parallel to and distinct from the E. Alps high-velocity heterogeneity (HV-EA). Kästle et al. (2018) identified approximately 1° to the north of the Periadriatic Fault a similar subparallel fast velocity heterogeneity in

their surface-wave tomography. Considering the NE continuations of the HV-BM as imaged in body-wave tomography of a larger extent (e.g., Karousová et al., 2013; Paffrath et al., 2021), the heterogeneity strikes with the SW-NE trend, in parallel with the boundary of the Moldanubian (MD) and Brunovistulian (BV) mantle lithosphere in the BM, and the westernmost part of the Carpathian front. Finding a secondary slab/thickened and cold lithosphere north of the Alps strengthen our motivation to determine its proper location and discuss hypotheses regarding its origin. Besides linking the shallow heterogeneity beneath the Eastern Alps to subduction of the Adriatic plate, we also present and discuss in our paper three potential scenarios of the origin of the positive heterogeneity located beneath the southern Bohemian Massif (HV-BM).

## 2 Data

High spatial density of stations, involved in passive seismic experiments, high-quality of recorded data and dense ray-coverage of the upper mantle under the study are essential pre-requisites for reliable high-resolution tomographic imaging. The AlpArray passive seismic experiment, realized in a broad European cooperation (Hetényi et al., 2018a), provided the necessary high-quality recordings for such a study. We collected recordings from stations of the AlpArray Seismic Network (AASN, doi.org/10.12686 /alparray/z3_2015) and AlpArray-EASI network (doi.org/10.12686 /alparray/xt_2014). All the AASN stations installed in a 200km-wide band (Fig.1) along the densely spaced stations of the AlpArray-EASI complementary experiment (Hetényi et al., 2018b) were selected for this study. The N-S band of the EASI stations is oriented perpendicularly to the E. Alpine chain (crest) and runs through the BM in the north to the Adriatic see in the south at length of ~540 km.

We have applied several procedures to check the data quality (Vecsey et al., 2017), particularly data completeness and correct timing, to eliminate periods with the uncorrected leap seconds or failures of clock synchronizations. In station metadata files we fixed for example wrong channel naming, station mislocation, and incorrect units for poles and zeros (mixing Hz and radians/sec units). We collected seismograms from 1920 earthquakes recorded at 240 temporary and permanent stations involved in the AlpArray experiments, from which we selected a subset of 201 top-quality earthquakes from epicentral distances greater than 30° with as uniform as possible distribution relative to the region of the E. Alps (Fig. S1). On this data set we picked coherently teleseismic P-wave arrival times with a fully automatic picker TimePicker 2017 (Vecsey et al., 2021) developed in the ObsPy/Python platform (Krischer et al., 2015). The TimePicker 2017 is based on two-step signal cross-correlations, and allows us to measure absolute arrival times (Fig. S2a). A low-noise beam trace created from stacked cross-correlated and shifted traces of an event serves as a reference in the second cross-correlation step and beam forming in the P-arrival time picking. This means, there is no subjective a priori selected single reference trace. The final time of each extreme (green P1 and P2 in Fig. S2a) and its error estimate is computed from the normal distribution, which approximates a mixture of normal distributions of partial picks. We have applied the picker on full data set of 1920

teleseismic events recorded by the AASN. Figure S2b shows uncertainties of the measured P-wave arrivals, means and medians for both the complete dataset as well as for events selected for tomography (see below).

Each earthquake in the sub-set of selected 201 earthquakes was recorded at least by 50 stations of the array, i.e., at least 20% of all the stations in the area. In this data set, 130 rays per event sampled the mantle on average. The conditions assure sufficient stability of the reference level in computing the relative traveltime residuals. The north-south elongated shape of the region oriented across the Alpine structures and perpendicular to the strike of the presumed subduction does not guarantee the same resolution along and across the strike. To eliminate mapping effects of heterogeneities aside the model into its internal part and to enhance the resolution in direction of the subductions, we selected additional rays coming from the northern and southern 60° wide azimuth bins. Only rays propagating through the model within the two azimuthal fans (see Fig. S1b) have been included in the final tomographic inversions, tested, discussed and interpreted further. This data comes from 244 earthquakes, each of them being recorded by 120 stations on average, i.e., by 50% of stations in the region. The dataset with enhanced data from N and S provide sharper picture of the high-velocity heterogeneities associated with the subducting slabs (see Fig. S5).

Teleseismic data cannot resolve velocities in the crust itself due to their sub-vertical propagation at the very shallow depths. To avoid mapping effects from the crust into the velocity perturbations in the upper mantle (e.g., Karousová et al., 2012), we have introduced crustal corrections (Fig. S3) for sediments (thickness, velocities) and variations in thickness and velocity of the crust. We compiled parameters for the crust corrections from different sources: from Karousová et al. (2012, and references therein) for the BM, from, e.g., Di Stefano et al. (2011), Hua et al., (2017), Tesauro et al.(2008) south of the BM. Crust corrections along the EASI transect were based on model by Hetényi et al. (2018b). Carefully pre-processed P-wave travel-time residuals calculated relative to the IASPEI'91 velocity model (Kennett and Engdahl, 1991), corrected for the crust, normalized to average residual per event and cleaned from outliers serve as input to the inversion. With this approach we gathered a high-quality and uniform dataset of travel time residuals for a proper tomographic inversion to resolve structures in our target region.

## 3 Method

We retrieved the velocity perturbations in the upper mantle by isotropic mode of the coupled anisotropic-isotropic tomographic code AniTomo (Munzarová et al., 2018a) derived from the broadly used Telinv code (e.g., Weiland et al. 1995; Arlitt et al. 1999; Lippitsch et al. 2003; Sandoval et al. 2004; Shomali et al. 2006; Eken et al. 2007; Karousová et al. 2012; 2013; Plomerová et al. 2016; Silvennoinen et al. 2016; Chyba et al. 2017).

Weak anisotropy with hexagonal symmetry, both with the high-velocity *a* axis or with the low-velocity *b* axis generally oriented in 3D, are assumed in coupled anisotropic-isotropic tomographic code AniTomo. Velocity at each point can be expressed as

$$v = \bar{v}\left(1 + \frac{k}{2}cos2\alpha\right)$$  (1)

where $\bar{v}$ is isotropic component of anisotropic velocity, $k$ is strength of anisotropy and $\alpha$ is an angle between the symmetry
axis and wave-propagation direction (for details see Munzarová et al., 2018a). The linearized relation between a travel time residual $\Delta t$ and perturbations of four anisotropic parameters $\Delta\bar{v}$ (isotropic component of the anisotropic velocity), $\Delta k$ (strength of anisotropy), $\Delta\lambda$ and $\Delta\theta$ (azimuth and inclination of the symmetry axis) at each grid node (indexed with i) attains a form

$$\Delta t = \sum_i \left(\frac{\partial t}{\partial \bar{v}}\right)_i \Delta\bar{v}^i + \sum_i \left(\frac{\partial t}{\partial k}\right)_i \Delta k^i + \sum_i \left(\frac{\partial t}{\partial \lambda}\right)_i \Delta\lambda^i + \sum_i \left(\frac{\partial t}{\partial \theta}\right)_i \Delta\theta^i.$$  (2)

Due to the elongated shape of the region (Fig.1), which is not suitable for a coupled anisotropic-isotropic tomography and out of the scope of this paper, we apply on the AlpArray-EASI data only the isotropic mode of the AniTomo code, in the first step. Then formula (2) reduces to the relation between the travel-time residual and the isotropic-velocity perturbations

$$\Delta t = \sum_i \left(\frac{\partial t}{\partial \bar{v}}\right)_i \Delta\bar{v}^i,$$  (3)

The system of linear equations is then solved with the standard damped least-square method (e.g., Menke, 1984)

$$\mathbf{m} = \left(A^T W_D A + \varepsilon^2 W_M\right)^{-1} A^T W_D \mathbf{d},$$  (4)

where $\mathbf{m}$ is vector of model parameters $\Delta\bar{v}$ at all nodes. Data vector $\mathbf{d}$ contains travel-time residuals $\Delta t$ and matrix $A$ stores the partial derivatives from equation (2) or (3). Errors of arrival-time measurements are considered in weighting matrix $W_D$. Damping factor $\varepsilon^2$ stabilizes the ill-posed problem. Horizontal smoothing of model parameters can be achieved via matrix $W_M$. The inverse in equation (4) is approximated by truncated singular value decomposition. 3D ray-tracing bending
technique Simplex (Steck and Prothero, 1991), in which ray paths are distorted by sinusoidal signals, is applied. Reliability of the model parameters for a given ray distribution and inversion setup can be assessed with resolution matrix $R$

$$R = \left(A^T W_D A + \varepsilon^2 W_M\right)^{-1} A^T W_D A.$$  (5)

The area of about 400 000 km$^2$, centred at 13.3°E 48.5°N, is approximated by 30-by-30 km cell size, horizontally. The images are calculated down to 435 km depth with 30 km spacing. To minimize creating false perturbations, we invert for the
velocity perturbations only in the central 5 x 25 x 13 cells, which are well-sampled by criss-crossing rays (Fig. S1), i.e., in nodes between depths of 60 km and 420 km.. The model covers the Eastern Alps and a core of the BM, an area of ca. 140 400 km$^2$ in total. Variance reduction of the final model for the chosen damping parameter attains 66% (Fig. S4).

## 4 Results

The distinct, high-velocity, northward dipping, ~140km broad perturbations related to the E. Alpine root, imaged in the upper ~250 km of the mantle by previous tomography (Babuška at al., 1990, Karousová et al., 2013; Hetényi et al., 2018b), had a tendency to split, when we exploited data from the EASI experiment and nearby permanent stations (Plomerová et al., EGU 2018). However, "only" adding data from the AASN lead to the clear visualization of two separate sub-parallel high-velocity heterogeneities beneath the broader E. Alpine region, both dipping to the north and each about 80 km thick (Figs. 2

and 3), with a low-velocity separation zone of ~80-100 km extent.

A clear decrease of amplitudes with depth dominates in the horizontal depth slices (Fig. 2 – left) through the EASI-AA velocity-perturbation model. They exceed +/- 1% only exceptionally below 220 km depth (two deepest layers shown in Fig. 2). Negative perturbations in the two uppermost mantle layers concentrate along the Eger Rift (ER) and can be related to the

lithosphere thinning in this region relative to the MD part of the BM (e.g., Plomerová and Babuška, 2010; Plomerová et al., 2016). At greater depth, the lower velocities dominate in the sub-lithospheric mantle beneath the whole BM (e.g., Amaru, 2007; Fichtner and Villasenor, 2015). The distinct positive velocity perturbations related to the E. Alpine root are located north of the PAL at ~47° N and are distinct down to 225km depth. At greater depths both the positive and weak negative perturbations are arbitrarily mixed and do not indicate any continuous object. Relatively smaller-size positive velocity

heterogeneity, north of the strongest large one, lies beneath the southern BM at 100-200 km depth.

To have a better sense of both lateral and vertical changes of the perturbations and thus to the dip of the high-velocity heterogeneities, we contour the 1.5% positive perturbations between 90 and 210 km depth (Fig. 2-right). The contours clearly mark the northward dip of the EA slab, particularly in its eastern part (east of 13.3°E). A dip of the western rim of the

slab is not clear from the contour curves only, as it appears to become steeper and thins significantly with depth. Similarly, it is difficult to judge a dip direction of the smaller-size, positive heterogeneity around ~48.7°N (HV-BM) in this visualization. To image the dip-directions of the heterogeneities, we present five N-S vertical cross-sections through the EASI-AA model (Fig. 3 a-c), perpendicular to the strike of the mountain belt. The images of slab geometry suggest changes along-strike of the Alpine orogen, even on short distances, but one has to keep in mind changes in resolutions toward the margins of the

model as well (see sections with synthetic tests). In general, the positive perturbations reach down to ~220 km. The perturbations beneath the HV-BM are slightly weaker in comparison with those in the HV-EA heterogeneity and disconnected from shallow parts. The HV-BM can be mapped only below ~100 km depth down to ~220km. On the other hand, the strongest heterogeneity in the south does not exhibit signs of delamination, and thus through its dip and existing connection to crustal levels we associate it with a continuous subduction of the Adriatic plate. All perturbations below ~250

km are very weak without any clustering or evident association with the stronger heterogeneities above this level. The limits of the numerically obtained well-resolved area are reported on the figures, with grey shading outside it.

The two positive-velocity perturbations seem to immerse in the northward direction, into the sub-lithospheric mantle, at an apparent dip of ~45° or more. The general dip of these heterogeneities marked in the cross-sections, changes only slightly in direction toward the Central Alps. In the central cross-sections, the HV-EA heterogeneity appears shorter than in the easternmost cross-section, while the detached lithosphere fragment reaches slightly deeper. Cross-sections through a 3D visualization of along strike changes of the velocity perturbation in the EASI-AA model can be found in Figure S9

## 5 Resolution Tests

We have performed several synthetic tests, to evaluate the resolution of the tomography results, particularly its ability to detect the two separate sub-parallel slabs beneath the E. Alps and BM and their dip direction. The polarity reversal of the northward subduction beneath the E. Alps relative to the south-eastward subduction in the Western Alps is of particular importance. The polarity flip is still questioned by some authors (e.g., Kind et al., 2021 this issue) in spite of long-lasting various inferences speaking for the change in subduction polarity beneath the W. and E. Alps with a gap between them (e.g., Babuška et al., 1990 Lippitsch et al., 2003; Zhao et al., 2016; Paffrath et al., 2021 this issue, and references therein).

Test 1 (Figs. 4, S6a-d) was designed to compare data-retrieved perturbations with those resulting from one or two narrow vertical synthetic heterogeneities, without imposing any polarity of the subductions. The model with one 5% velocity heterogeneity does not reproduce the velocity perturbations retrieved from real data. The model with two steep heterogeneities mimics the perturbations much better both in the central part of the model and its margins. The perturbations retrieved from the synthetic vertical heterogeneities remain vertical for the real ray geometry or with a weak southward dipping tendency in the westernmost profile, which contradicts the northward dip of real perturbations. Evidently, there is no northward smearing due to the ray geometry.

After accepting the existence of two heterogeneities, we have tested their relative orientations. Test 2 (Figs. 4, S6a-d), assumes two heterogeneities as in Test 1, but with 27° southward dip. The resulting perturbations do not reproduce the northward dip of the real perturbations. On the other hand, Test 3 with the two heterogeneities dipping to the north at 27° mimics the dip of real perturbations very well. Test 4 with two bi-vergently dipping heterogeneities (towards each other) match the geometry of the real perturbations only at shallow depths, above ~150km, but the deeper part of the northern heterogeneity is completely missing.

The above performed synthetic tests corroborate that the data from the AlpArray and EASI networks are able to image two separate northward dipping sub-parallel slabs beneath the E. Alps and southern rim of the BM. The two slabs are separated from each other, and the northern one is not connected with the shallow parts of the lithosphere (above ~100km). The

difference between the dip directions of the subductions beneath the E. Alps and the subductions beneath the W. and Central Alps is undoubtedly real and it is not produced by potential smearing due to ray geometry.

Besides the specific tests described above we also performed standard checkerboard tests to assess resolution capability of the network (Fig. S7). The checkerboard test confirms the positive and negative perturbations are retrieved well down at

least to 240 km with a weak vertical smearing (Fig. S7a – horizontal slices). Also the vertical cross-section through the central part of the model (Fig.S7b) images the synthetic perturbations reliably.

## 6 Imaging the high-velocity perturbations in different tomography model

The present shape of the Alpine mountain chain, characterised by the curved Western Alps and east-west striking Central and Eastern Alps, reflects a multi-phase action of tectonic forces during the collision of the European and Adriatic plates, the

AlCaPa micro-plate and numerous lithosphere fragments in the regions, as well as the Piemont oceanic lithosphere. The processes are imprinted in the complex architecture of the broader Alpine region, both in the crust (Handy et al., 2010; Rosenberg and Kissling, 2013; Schlunegger and Kissling, 2015) and in the mantle (e.g., Kissling et al., 2006). Continuing debates on exact setting the Moho depth in the Alps and on the "gap" near the Tauern window (Spada et al., 2013, Hetényi et al., 2018b; Brückl, Tectonics 2010) document the complex structure of the Alpine orogen.


Though the upper mantle structure is less diverse in comparison with the crust, in general, ongoing studies of the Alpine upper mantle continue to reveal new and more detailed features in geometry of the lower lithosphere, dip direction of the Alpine slabs, tears or detachments of the slabs and interactions of the Alps with the Apennines and Dinarides. The complex structure of the fragmented Alpine slab(s) and the broader Europe/Adria collision zone is now visualized in tomography

snapshots. The current stage of knowledge from results of various disciplines – seismology, geology, petrology, tectonics, paleo-magnetism, geochemistry, GPS studies etc. - reflect differences in the segmented slab responses to the acting forces. In recent studies, Paffrath et al. (2020) suggest the reversed slab polarity relative to the Western Alps already in the Central Alps, as opposed to the formerly documented polarity reversal further to the east - beneath the E. Alps (e.g., Babuška et al., 1990, Lippitsch 2003; Zhao et al., 2016). Mock et al. (2020) note discordance between the slab geometry at depth and the

boundary between the Eastern and Central Alps observed at the surface. The boundary between the central and eastern slab segments of the Alps, as defined either by slab gaps due to delamination (Handy et al., 2021) or a change in dip direction of the subductions,, coincides with the Giudicare-Brenner fault system, in general (e.g, Rosenberg et al., 2018).

As early as 1990 Babuška et al. (1990) suggested segmentation of the Alpine slab into the western and eastern parts with

opposite polarities, and a gap in between them, in tomography of central Europe, in which the authors inverted crust-corrected and source-side clustered travel time residuals. This model (Fig. 5) has rather high variance reduction, 70%, but a resolution only at 1.5°x1.5° grid due to station spacing available at that time. The Alpine slab segmentation has been

confirmed in the upper mantle tomography of the Alpine orogen by Lippitsch et al. (2003). Their regional tomography with finer lateral grid of 50x50 km has the highest resolution around the 12° E, just between the two strongest high-velocity

perturbations beneath the Central and Eastern Alps, where the passive seismic network TRANSALP was deployed. Later tomography of the upper mantle which included the E. Alps from data of regional passive experiments (Dando et al., 2011; Mitterbauer et al., 2011; Karousová et al., 2013) also retrieved the northward dipping high-velocity heterogeneity of similar geometries (Fig. 5) in 250 km of the upper mantle. . The imaged triangular shape of the LAB model (e.g., Babuška and Plomerová 1992) beneath the thickened lithosphere of the E. Alps, detached from that beneath the W. and C. Alps, led

Babuška et al. (1990) to suggest three main phases in building the E. Alps lithosphere root: (1) NW translation of the Adria and its thrusting over the (subducting?) European plate in the W. Alps, (2) fragmentation of northern Adria along a deep-seated fault (possibly the Giudicarie Fault, or at least a spatially nearby structure) and (3) counter-clockwise rotation of the Adria resulting in a start of its subduction after a collision with the European plate in the Eastern Alps (e.g. also Ustaszewski et al., 2008), potentially above a delaminated European lithosphere residing at greater depths (Paffrath et al., 2021, Handy et

al, 2021). Complexity of this region is reflected in ambiguous views on the crust, being interpreted as a triple-junction of three crustal terranes (Brückl et al., 2010), although the deformation style between the E. Alps and the Pannonian Basin is usually considered diffuse on the surface. Instead, e.g., Ustaszewski et al. (2008) and Horváth et al. (2015) assume the Pannonian Basin underlain by gradually thinned European and Adriatic crust, which has been recently imaged by Kalmár et al. (2021).


The most recent tomography of the entire Alps and surrounding regions (Paffrath et al., 2021 this issue, Handy et al., 2021) exploit data from the AlpArray seismic network and AlpArray complementary experiments. The large-scale AlpArray tomography and the EASI-AA model along the EASI with similar gridding exhibit remarkable coincidence of perturbation patterns. In both tomography images, the HV-EA is located between the PAL in the south and the NAF in the north, but the

large-scale tomography (Paffrath et al., 2020; 2021 this issue) images the E. Alps subduction below 150 km depth with low velocity perturbations (red) above it. At this depth, horizontal slices show characteristic separation of the Wester, Central and Eastern Alps, which becomes more distinct deeper in the mantle. Similarly, Zhao et al. (2016) show only weak positive perturbations at 100 km depth beneath the E. Aps, but this region lies in a relatively less well-resolved part of their model.

A local discrepancy between models is present in the northern BM, where the large tomography returns positive perturbation, but the EASI-AA tomography maps the low-velocities there in accordance with other studies (e.g., Plomerová et al., 2016). However, the large and EASI-AA tomography detect positive perturbations beneath the Moldanubian part of the southern BM. The perturbations continue further to the south-west in the EASI-AA in comparison with Paffrath (2020) tomography. Similar local high-velocity perturbations down to ~250 km depth, isolated from the EA heterogeneity, have

been detected also in body-wave tomography by Dando et al. (2011) and by Karousová et al. (2013) (Fig. 5) as well as in surface-wave tomography by Kästle et al. (2018, 2020), though the surface waves localize it south-west of the BM.

The north oriented dip of the EA subduction, imaged in early tomography studies (Babuška et al., 1990, Aric et al., 1989), was questioned for a long time, until Lippitsch et al. (2003) clearly imaged the northward dipping structures using

TRANSALP data as well. Subsequent regional passive experiments provided body-wave data for tomography of the E. Alps and surrounding regions (namely in the Bohemian Massif and Pannonian Basin (BOHEMA (e.g., Karousová et al., 2013, Plomerová et al., 2016), ALPASS (e.g., Mitterbauer et al., 2011) and CBP (e.g., Dando et al., 2011) passive experiments) and imaged again the northward dip of the EA slab. Most of them agree in interpreting it being of Adriatic plate origin (Dando et al., 2011; Karousová et al., 2013). However, Mitterbauer et al. (2011), and similarly in recent tomography based

on the newest Alp Array data by Paffrath et al., (2021 this issue), the positive perturbations below 150km are associated with a delaminated EU slab (see also Handy et al., 2021 this issue). Kästle et al. (2020) relate the HV-EA mainly with the European plate subductions as well, and leave no or only minor role to the Adriatic subduction. The authors explain the northward dipping subduction modelled beneath the E. Alps by imaging a combination of the short Adriatic and deep delaminated, potentially overturned European slabs.


Synthetic tests from the EASI-AA data set in our study proved that the EASI-AA models is capable to image the two sub-parallel northward dipping heterogeneities – the large and strong southern one beneath the E. Alps with connection to shallow depths, and the weaker northern one beneath the southern BM apparently disconnected. To understand the positive perturbations beneath the southern BM, we compare it with results from the large-scale Paffrath et al. (2021) tomography

and with the regional tomography of the BM (Karousová et al., 2013) of similar resolution to ours (Fig. 6).  The strongest positive perturbations related to that heterogeneity overlap in the models, though they are of unrealistically large extent in the Paffrath's et al. model (2021, this issue). Paffrath et al. (2021) show the strong positive perturbations also beneath the north-western part of the BM (beneath the Eger Rift), where the lithosphere thins significantly (~80 km). This is well imaged by negative perturbations in the EASI-AA and in BOHEMA models (Karousová et al., 2013; Plomerova et al., 2016)..

Images of perturbations at 180km depth stress the importance of using data from stations in the northern part of the EASI array. Data from these stations capture the positive velocity perturbations much farther to the SW, beneath the southern rim of the BM, than the other two tomography studies. Cross-sections through the BM regional tomography (see Fig. 5) locate the increased velocities within the low velocity BM upper mantle (REF e.g., Amaru 2007, Fichtner and Villasenor, 2015). The SW-NE elongated shape of the heterogeneity follows the boundary between the MD and BV mantle lithosphere

domains (Babuška and Plomerová, 2013).

Different types of seismic waves propagate with different velocities and sample the mantle volume in different directions and wavelength, which affects the velocity/velocity perturbation images of the upper mantle and corresponding resolution. A cross-section along 13.5°E through the MeRE2020 model, a shear-wave velocity perturbation model from Rayleigh phase

velocities (El-Sharkawy et al., 2020), runs parallel to the EASI transect in our P-wave model. The authors relate positive

perturbations eastward of the cross-section at ~45°N to the northernmost part of the Dinaride slab and images a short (only down to 150km depth) European slab, without any delamination. On the other hand, the body-wave tomography by Paffrath et al. (2021) sees the top of high-velocity heterogeneity beneath the E. Alps at 150km depth. The heterogeneity extends down to ~300km. The authors interpret it as the delaminated European plate lithosphere. There is obvious contradiction between these two interpretations.  To test whether a potential delamination can be detected in our EASI-AA tomography, we performed additional synthetic tests (Fig. S8). In each test the top of the steep 5% high-velocity heterogeneity shallows from 150km to 60km depth. The tests demonstrate that the slab detachment larger than the 30km grid would be revealed in our EASI-AA model. Two northward dipping slabs (Fig. 7), the northern small-size delaminated one, approximated with 3% velocity increase, and the southern one attached, with 5% velocity increase, mimic well the velocity perturbations retrieved beneath the Bohemian Massif and beneath the Eastern Alps from real data (Fig. 3).

The inconsistency between the above mentioned results in the upper ~150 km of the upper mantle could come from different sources, e.g., a sub-optimal ray coverage in that area and depth range, but also can reflect differences in crustal corrections applied.  Certain role can play also the difference in size of the arrays. Velocity perturbations in P-wave tomography are linked to averages within each layer (a plane of grid nodes), which might differ in a large array from that of a smaller-size array. Data from large-scale arrays allows to model heterogeneities down to greater depths than a smaller-size regional tomography, which can, on the other hand, provide finer images. While tomography images presented here have the best resolution above 200 km and allows to image a shallow slab beneath the Eastern Alps attached to Adriatic plate, the large-scale model from the entire AlpArray has the best resolution at depths 200-400 km (Paffrath et al., 2021 this issue) and thus is able to images delaminated European slab beneath the Eastern Alps.

 The role of applying proper crustal corrections is significant in teleseismic regional tomography.  Not applying any crustal corrections or applying inadequate ones can strongly affect velocity perturbations within the upper ~100 km of the upper mantle (e.g., Karousová et al., 2013), which is the zone, where the models discussed above differ. "Overcorrecting" the travel times due to a crustal model used, regardless of a method of correcting itself, can delete/erase or substantially reduce positive perturbations in the upper 100 km, if too slow and thick crust would be considered and vice versa.   From this point of view, developing a uniform detailed and reliable model of the European crust is urgently needed.

**7 Potential scenarios for geneses of the high-velocity heterogeneities**

Various evolution scenarios for the E. Alps slab exist, but there are none for the HV-BM beneath the southern BM. For detailed scenarios of the Alps subduction and the Europe-Adria plate collision we refer to Schmid et al., 2004; Handy et al., 2010; 2015; Le Breton et al., 2017; Kissling and Schlunegger, 2018; Rosenberg et al., 2018; Paffrath et al., 2021 and references therein. Their models of Alpine orogeny include subductions of the European and Adriatic plates, slab roll-back,

tearing, break-offs and delamination, widespread intra-crustal and crust–mantle decoupling, as well as the NW translation and counter-clockwise rotation of the Adriatic plate. All these processes, accompanied by thermal erosion or/and deglaciation uplifts, are reflected in different structure of the WA and EA. But how to interpret the positive velocity perturbations within the low-velocity upper mantle beneath southern BM (HV-BM)?

The elongated HV-BM is ~80km broad and extends, considering both the EASI-AA and Paffrath's (2021) model, approximately between 12.5°-16°E over a length of ~300 km on the surface. The elongated shape of this heterogeneity strikes in the SW-NE azimuth, and extends from ~100 km down to ~200 km depth. We estimate the total volume of the HV-BM at 1.5 million km$^3$; these dimensions are comparable to a small lithospheric segment. The smaller part of the HV-BM imaged in EASI-AA model extends at a low angle relative to the Eastern Alpine front. But considering its full size, it is subparallel to the MD/BV contact in the BM as well as the Western Carpathians front. The high-velocity material hovers in the low-velocity upper mantle beneath the BM. We outline three potential scenarios for explanation of an origin of this positive heterogeneity.

The simplest explanation would be to consider it as a fragment of the delaminated part of the European plate subductions, as suggested in Handy et al. (2015) for the high-velocity heterogeneity beneath the Eastern Alps at greater depth (Fig.8a). In their model, the second break-off or delamination of the European slab at ~25Ma opened space for the northward subduction of the Adriatic plate, pushed by Africa from the south and rotated by pulling due to subduction of its SE rim beneath the Hellenides. The delaminated piece of the continental lithosphere has continued sinking into the mantle in the model since then. However, the HV-BM is located at shallow depth (above ~200km, not really compatible with sinking) and is too far to the north (~49°N) from the Periadriatic Fault System. Therefore, an association of the HV-BM with the delaminated fragment of the EU subduction is not likely. Also, the clearly imaged separation (negative anomaly) between the subducting HV-EA and the HV-BM is a feature that would not be explained in this scenario.

The BM is an assemblage of fragments of continental lithosphere with their own large-scale anisotropic fabrics (e.g., Babuška and Plomerová 2013, 2020). Changes in the fabrics delineate boundaries between the mantle lithosphere domains. Location of the HV-BM, following the boundary of the MD/BV mantle lithosphere domains, evokes a possible link to the MD/BV collision, which is related to the closure of the Rheic ocean in late Devonian-Middle/Late Carboniferous (Babuška and Plomerová, 2013; and discussion and references therein). The HV-BM is disconnected from the BM lithosphere in tomography cross-sections (see, e.g., Figs. 3, 5). The continual subduction of the oceanic plate due to negative buoyancy would lead to a removal of the denser materials from shallow depths since then. The Phanerozoic continental mantle lithosphere may mechanically decouple from the upper or the full crust, and since this lower lithosphere is denser than the asthenosphere it subducts. Densification of the lower crust through metamorphic reactions may enhance this process if the convergence is fast (e.g., 2 cm/yr in Hetényi et al., 2011). The Phanerozoic continental mantle lithosphere, composed of

originally lighter rocks than those in the asthenosphere, becomes denser due to metamorphic phase changes as it subducts. This is the general process, however at low convergence rates it is able to slowly return from the negative to positive buoyancy range (Bonma et al., 2019). Such a process could "stop" subductions, and allow for a long-term survival of this lithosphere material in the asthenosphere, with the high-velocity anomaly caused by chemical composition rather than temperature. Thus the HV-BM could represent a remnant of the Rheic oceanic closure and/or a relic of the MD/BV collision, captured in the slow sublithospheric BM mantle (Fig 8b).

A third scenario  (Fig. 8c) could be some relatively light material, brought to its current position beneath the BM not too long ago to survive there (i.e., not to sink rapidly).To get such material there is difficult, but not impossible considering the evolution of the European plate and the Carpathians. The shape of the originally linear, W-E running Alpine-Carpathians front has changed since the time of the second European slab break-off, AlCaPa lateral escape and beginning of the Adria subduction (Handy et al., 2015). The Carpathians front curved significantly and migrated to the north. Differences in the roll-back subductions of the Alps (e.g., Kissling and Schlunegger, 2018 and reference therein) and the Carpathians, northward push of Adria and European slab delamination beneath the EA could have formed complex flows in the asthenosphere (e.g., Vignaroli et al., 2008), which  could "transport" a purely oceanic lithosphere or a mix of oceanic and continental lithosphere fragments through the open space between the E. Alpine and Carpathian slabs north-northeastward into the mantle beneath the BM.  The rotational displacement of the Adriatic Plate indenter  provided an additional driving force for modifications of the Alpine-Carpathian-Dinaridic orogenic system (Ustaszewski et al., 2008). Remnants of the Penninic/Piemont oceanic lithosphere (Brückl et al., 2014 – Fig. 9 there), squeezed east of 14°E between the AlCaPa and European plates in the W. Carpathians and pushed from the south by the Adria, can offer a possible explanation for the origin of the SW-NE elongated HV-BM striking sub-parallel to the W. Carpathian front.

Finding an unambiguous model of the complex Alpine orogeny and structure of the upper mantle in the broader surroundings of the Alps requires multi-method and inter-disciplinary research that covers various spatial scales. Combination of gravity and seismic data represents one of such approaches (e.g., Lowe et al, 2021, this issue; Scarponi et al., 2021).  Lowe et al. (2021, this issue) converted modified standard isotropic velocity perturbation models into velocity and then density models. Those are after that used to calculate the gravity signal, predicted up to 40 mGal for various slab configurations mimicking the Alps. The applied methods include severe simplifying assumptions. Nevertheless, neither including pre-defined slab geometries nor accounting for compositional and thermal variations with depth brings satisfactory results, which would allow them to distinguish between their two different slab configurations. The freshly compiled pan-Alpine surface-gravity database (Zahorec et al., 2021 in print) will undoubtedly provide new impetus for structural investigations combining seismology and gravity. Regarding the gravity effect of the HV-BM, the expected signal is too weak to appear there clearly, as crustal effects are predominant in that area of the BM.

Anisotropic nature of the Earth has been proved as a general characteristic in different seismological studies. Anisotropy influences mainly velocities and polarizations of seismic waves. Seismic anisotropy of the Earth's upper mantle carries a key information for deciphering tectonic history of the lithosphere–asthenosphere system (e.g., Babuška and Cara 1991; Sobolev, 1999; Fouch & Rondenay 2006; Long & Becker 2010, Babuška and Plomerová, 2020 and references therein). However, effects of directional dependences of velocities are not considered in standard isotropic tomography images. Only long-wavelength shear-velocity models from surface waves include azimuthal and/or radial anisotropy in the mantle, traditionally. Ignoring seismic anisotropy and assuming isotropic wave propagation or considering only azimuthal and/or radial anisotropy leads to significant isotropic and anisotropic imaging artefacts that may lead to spurious interpretations (VanderBeek and Faccenda, 2021). In this study of the broader region around the E. Alps we have applied the isotropic mode of a coupled anisotropic–isotropic teleseismic P-wave tomography developed by Munzarová et al. (2018a). In spite of the general good agreement with the high-resolution large-scale isotropic tomography (Paffrath et al., 2021, this issue), the images can be biased due to seismic anisotropy (Eken et al., 2012; Qorbani et al., 2015, 2016; Bokelmann et al., 2021). Laterally varying anisotropy, which correlates with tectonics of the region, has been indicated in shear-wave splitting (e.g., Link and Rumpker, 2021). After collecting sufficient amount and well-distributed high-quality data we will run the coupled anisotropic-isotropic mode of the code, , successfully applied in the northern Fennoscandia (Munzarová et al, 2018b). Resulting 3D anisotropic model of the region, which will map laterally and vertically varied anisotropy with symmetry axes oriented generally (i.e., inclined) in 3D. This further investigation may help in deciding among the drafted scenarios for the origin of the HV-BM, or point to new ones.

## 8 Conclusions

The here presented teleseismic P-wave tomography of the upper mantle beneath the Eastern Alps and the Bohemian Massif locates the Alpine high-velocity perturbations between the Periadriatic Lineament (PAL) and the Northern Alpine Front (NAF). The northward-dipping slab is imaged down to ~200 km, without signs of delamination from the Adriatic plate. Therefore, association of the high-velocity heterogeneity with the Adriatic plate subduction seems to be obvious. The fine-gridded EASI-AA model of velocity perturbations images at depths of ~100-200km the individual high-velocity heterogeneity beneath the southern part of the Bohemian Massif. Its eastward continuation is visualized in other tomography results as well. Interpreting this heterogeneity as a remnant of the delaminated European plate seems unlikely. The SW-NE trend of the heterogeneity strike, in parallel with the Moldanubian/Brunovistulian mantle lithosphere boundary in the Bohemian Massif or with the westernmost part of the Carpathian front, leads us to consider it as a piece of a mixture of the continental and oceanic lithosphere related to building of the BM, particularly to the closure of the old Rheic ocean during the MD/BV collision, or, as a lithospheric fragment going through to the NW between the E. Alps and W. Carpathians fronts in a preceding subduction phase.

Team list: The complete member list of the AlpArray Working Group can be found at http://www.alparray.ethz.ch

460    Author contribution: JP processed the P-wave residuals, analysed and interpreted results, and wrote the ms., HZ ran the AniTomo code, GH participated in interpretations and writing ms., LV developed and applied the P-wave arrival time picker, VB participated in early-stage discussions.

Competing interest: There is no competing interest

465

Code/Data availability: TimePicker 2017 will be accessible via web or upon request

**Acknowledgements.** We thank M. Handy and the anonymous reviewer for their valuable criticism and suggestions, which led to important clarifications and improvement of the manuscript, as well as E. Kästle for his editorial work. Research 470    within this study was supported by the Grant Agency of the Czech Republic (grant No. 21-25710) and station operation was supported by projects CzechGeo/EPOS-Sci CZ.02.1.01/0.0/0.0/16_013/0001800 (OP RDE), CzechGeo/EPOS LM2010008 and LM2015079. We acknowledge the operation of the temporary seismic network XT of the AlpArray-EASI complementary experiment and the AlpArray Seismic Network Z3. Contributions from all permanent seismic networks used in this study: BW – Department of Earth and Environmental Sciences, Geophysical Observatory, University of 475    München, 2001; CR – University of Zagreb, 2001; CZ – Institute of Geophysics of the CAS, 1973; GR – Federal Institute for Geosciences and Natural Resources, 1976; IV – INGV Seismological Data Centre, 2006; MN – MedNet Project Partner Institutions, 1990; NI – OGS and University of Trieste, 2002; OE – ZAMG, 1987; SI – ZAMG, 2006; SL – Slovenian Environment Agency, 1990; SX – University of Leipzig, 2001; TH – Friedrich-Schiller-Universität Jena, 2009, are acknowledged as well.

480

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

Figure Captions

**Figure 1:** Elevation map of the western part of the Bohemian Massif and the Eastern Alps, with the EASI and AASN seismological stations (left), and the grid for modelling structure of the region (right). Inverted and non-inverted nodes are filled in dark green and in yellow, respectively. The top-right location map also presents the large-scale tectonic context. Labels of main tectonic faults and units: PAL - Periadriatic Lineament, NAF - Northern Alpine Front, SAF - Southern Alpine Front, SEMP - Salzach–Ennstal–Mariazell–Puchberg fault, TW - Tauern Window (dashed), DF- Danube fault, Id - Idria fault, MD – Moldanubian part of Bohemian Massif (BM).

**Figure 2:** Depth slices through the new EASI-AA velocity perturbation model (left) along with depth contours of the 1.5% perturbations in a map view (right). Arrow marks the northward dip of the central and eastern part of the E. Alpine heterogeneity. Labels of main tectonic faults and units: PAL - Periadriatic Lineament, NAF - North Alpine Front, SAF - Southern Alpine Front, SEMP - Salzach–Ennstal–Mariazell–Puchberg fault, TW - Tauern Window (dashed), DF- Danube fault, Id - Idria fault, MD – Moldanubian and TB - Teplá-Barrandian units of the Bohemian Massif (BM), ER – Eger Rift.

**Figure 3:** Five north-south oriented vertical cross-sections through the EASI+AA velocity perturbation model: (a) along the EASI profile, (b) east of the EASI and (c) west of the EASI. Location of the cross-sections is marked in the map in the upper right part of the figure. The along-longitude cross-sections run from 51.65N to 45.35N. The three strongest positive velocity perturbations are highlighted. Fault abbreviations as earlier, Eu – Europe, Ad – Adria. Less-well resolved regions are shaded. Green dots mark Moho depths in the model used for calculation of crustal corrections (see also Fig. S3). Travel times are not inverted for velocity perturbations in the crust (nodes at 30 km depth).

**Figure 4:** Synthetic tests of tomography capability to resolve one or two sub-parallel heterogeneities (TEST1) and their dip directions (TEST2, TEST3 and TEST4) in the central cross-section along the EASI. The similar, north-south cross-sections parallel to the central EASI profile on its east and west, corresponding to profiles of Figure 3, are in Supplements S6a-d. The along-longitude cross-sections run from 51.65N to 45.35N. Fault labels as in Fig. 2.

**Figure 5:** Cross-sections through various regional teleseismic P-wave tomography results showing segmentation of the Western and Eastern Alps and changes in the slab dip directions (a-b), the northward dip of the steep slab (HV-EA) beneath the Eastern Alps (b-f) and the small size HV_BM heterogeneity beneath the Bohemian Massif (c,f,g). Cross-sections (a-b) from Babuška et al. (1990), with the crust in part (b) from Hetényi et al. (2018b) and perturbation contours from Karousová et al et al. (2013). The profile locations in a map view are in the upper right. The dark-blue and light-blue highlights refer to the location of the positive perturbations beneath the E. Alps and the Bohemian Massif ( HV-EA and HV-BM), respectively. The full set of colour palettes can be found in the original publications.

**Figure 6:** Horizontal slices at 120 km (a) and 180 km (b) depth through velocity-perturbation model EASI-AA (left) (this paper), model by Karousová et al. (2013) (right) and by Paffrath et al. (2021 this issue) (center), on which the HV-BM from the other two models is added as orange contours. Labels as in Fig. 2.

Figure 7: Velocity perturbations along the five vertical along-longitude cross-sections calculated from real data (upper row) and those (middle row) calculated from two synthetic 3% and 5% heterogeneities (bottom), plotted over the retrieved perturbations. Green dots mark Moho depths in the model used for calculation of crustal corrections (see also Fig. S3). The along-longitude cross-sections run from 51.65N to 45.35N.

**Figure 8:** Interpretation of the slab at shallow depth beneath the Eastern Alps (HV-EA), (Adriatic vs. European provenance), and potential scenarios of the origin of the heterogeneity beneath south-eastern Bohemian Massif (HV-BM). (a) European slab delamination at 20Ma and configuration of the European-Adria plate collision at present redrawn from Handy et al. (2015) complemented by location of the high-velocity heterogeneities from models EASI-AA (this paper) and Paffrath et al. (2020); (b) a scenario considering closure of the Rheic ocean and collision of the Brunovistulian micro-plate with the Moldanubian part of the BM in a portion of schematic cartoon by Babuška and Plomerová (2013), with a piece of remnant lithosphere image as the HV-BM in this paper; (c) a scenario related to fragmentation of the Alpine and Carpathian front. Differences in the roll-back subductions of the Alps and the Carpathians (e.g., Kissling and Schlunegger, 2018 and reference therein), northward push of Adria and European slab delamination beneath the E. Alps (Handy et al., 2015) could have formed complex flows in the asthenosphere (e.g., Vignaroli et al., 2008), which might "transport" lithosphere fragments toward the northwest. For more details see the text.

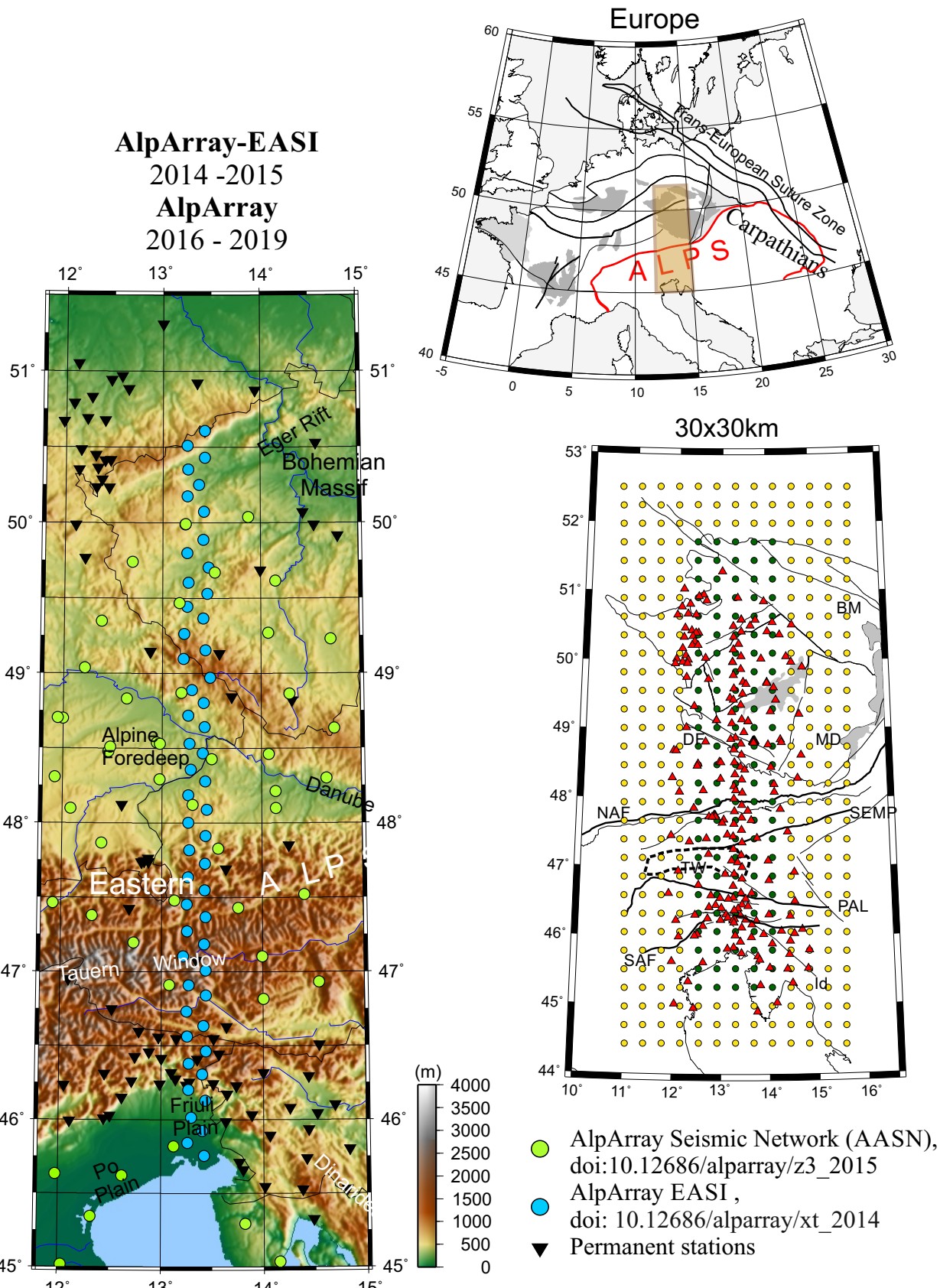

**Figure 1:** Elevation map of the western part of the Bohemian Massif and the Eastern Alps, with the EASI and AASN seismological stations (left), and the grid for modelling structure of the region (right). Inverted and non-inverted nodes are filled in dark green and in yellow, respectively. The top-right location map also presents the large-scale tectonic context. Labels of main tectonic faults and units: PAL - Periadriatic Lineament, NAF - Northern Alpine Font, SAF - Southern Alpine Front, SEMP - Salzach–Ennstal–Mariazell–Puchberg fault, TW - Tauern Window (dashed), DF- Danube fault, Id - Idria fault, MD – Moldanubian part of Bohemian Massif (BM).

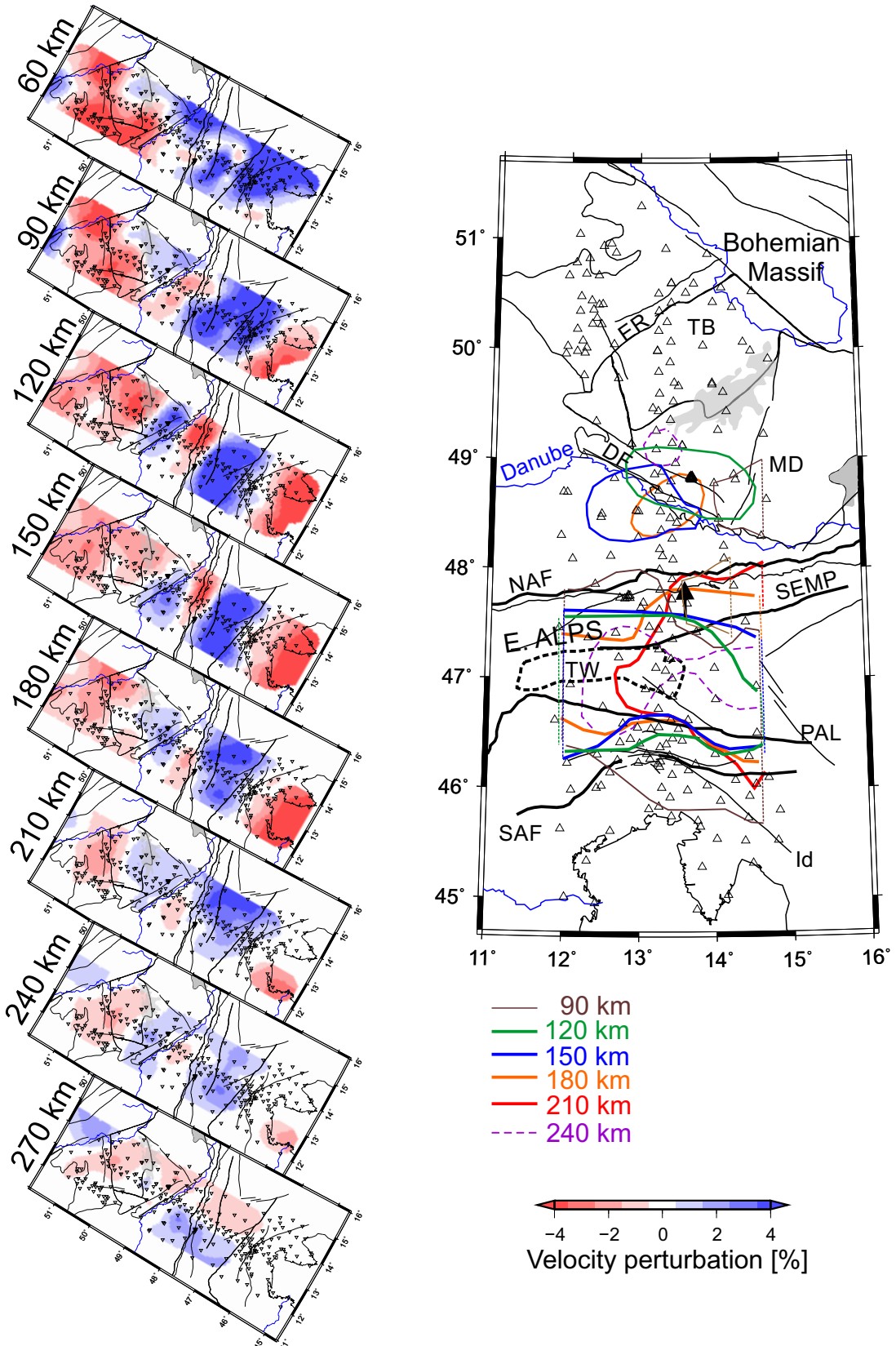

**Figure 2:** Depth slices through the new EASI-AA velocity perturbation model (left) along with depth contours of the 1.5% perturbations in a map view (right). Arrow marks the northward dip of the central and eastern part of the E. Alpine heterogeneity. Labels of main tectonic faults and units: PAL - Periadriatic Lineament, NAF - North Alpine Front, SAF - Southern Alpne Front, SEMP - Salzach–Ennstal–Mariazell–Puchberg fault, TW - Tauern Window (dashed), DF- Danube fault, Id - Idria fault, MD – Moldanubian and TB - Teplá-Barrandian units of the Bohemian Massif (BM), ER – Eger Rift.

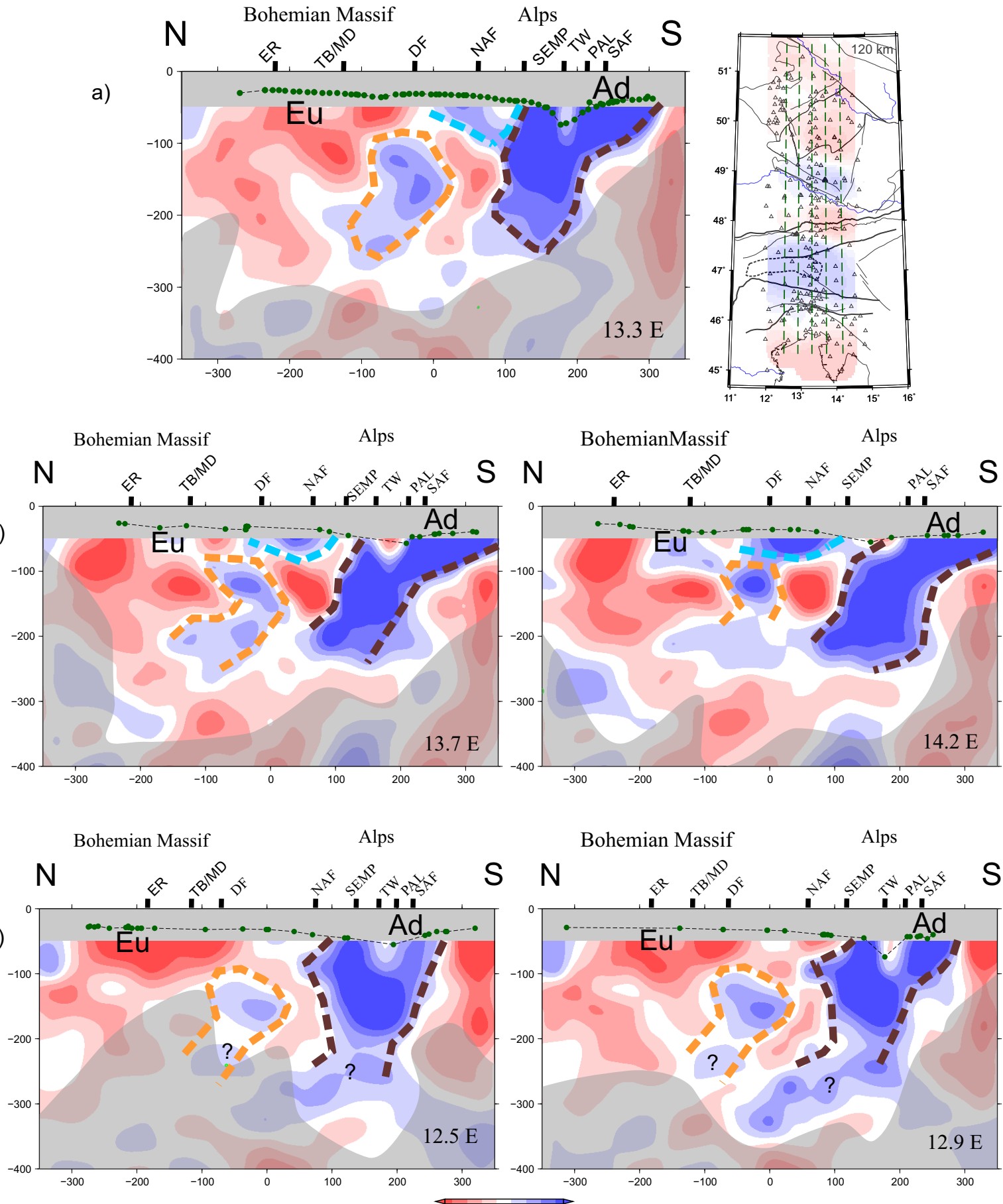

**Figure 3:** Five north-south oriented vertical cross-sections through the EASI+AA velocity perturbation model: (a) along the EASI profile, (b) east of the EASI and (c) west of the EASI. Location of the cross-sections is marked in the map in the upper right part of the figure. The along-longitude cross-sections run from 51.65N to 45.35N. The three strongest positive velocity perturbations are highlighted. Fault abbreviations as earlier, Eu – Europe, Ad – Adria. Less-well resolved regions are shaded. Green dots mark Moho depths in the model used for calculation of crustal corrections (see also Fig. S3). Travel times are not inverted for velocity perturbations in the crust (nodes at 30 km depth).

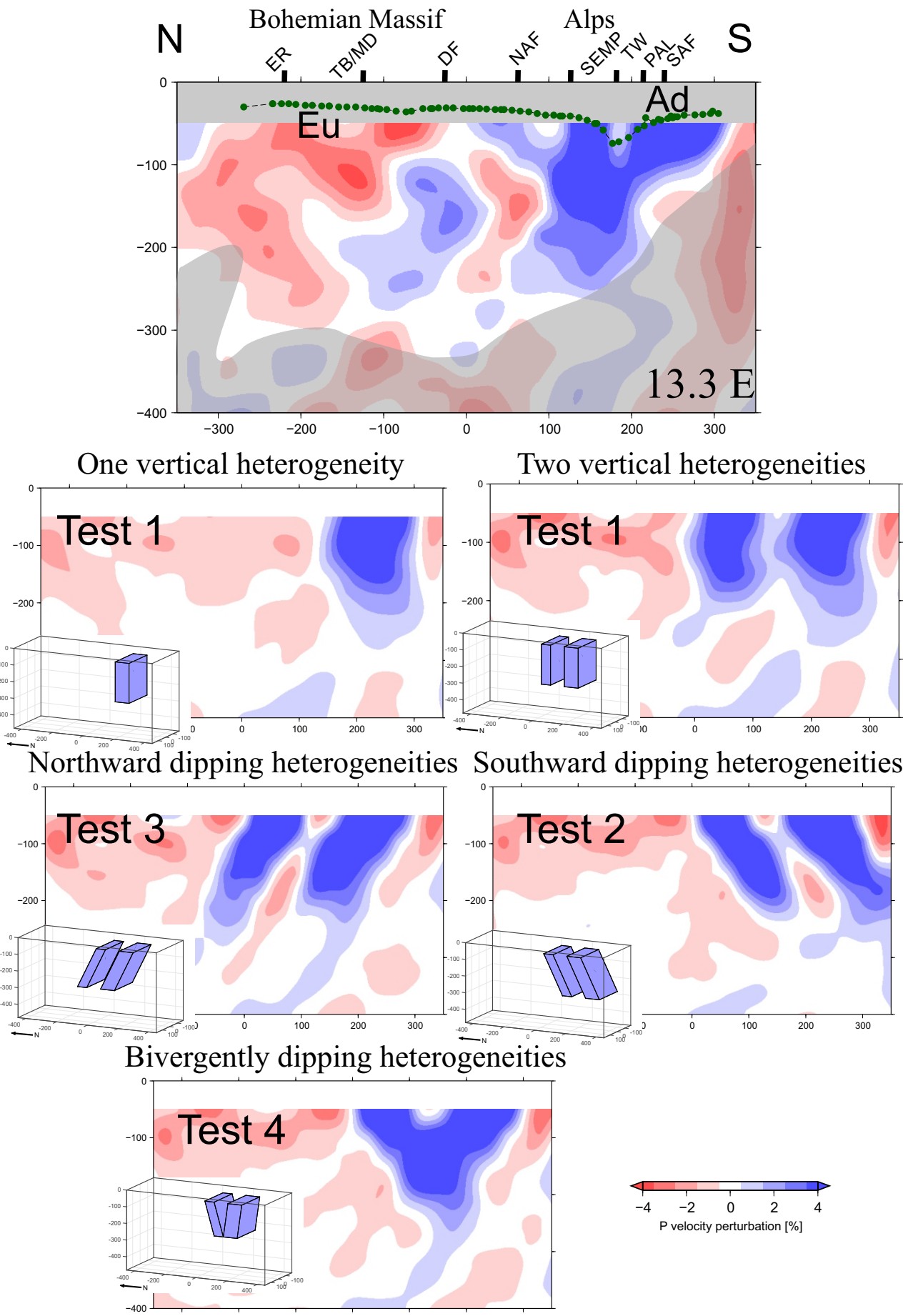

**Figure 4:**. Synthetic tests of tomography capability to resolve one or two sub-parallel heterogeneities (TEST1) and their dip directions (TEST2, TEST3 and TEST4) in the central cross-section along the EASI. The similar, north-south cross-sections parallel to the central EASI profile on its east and west, corresponding to profiles of Figure 3, are in Supplements S6a-d. The along-longitude cross-sections run from 51.65N to 45.35N. Fault labels as in Fig. 2.

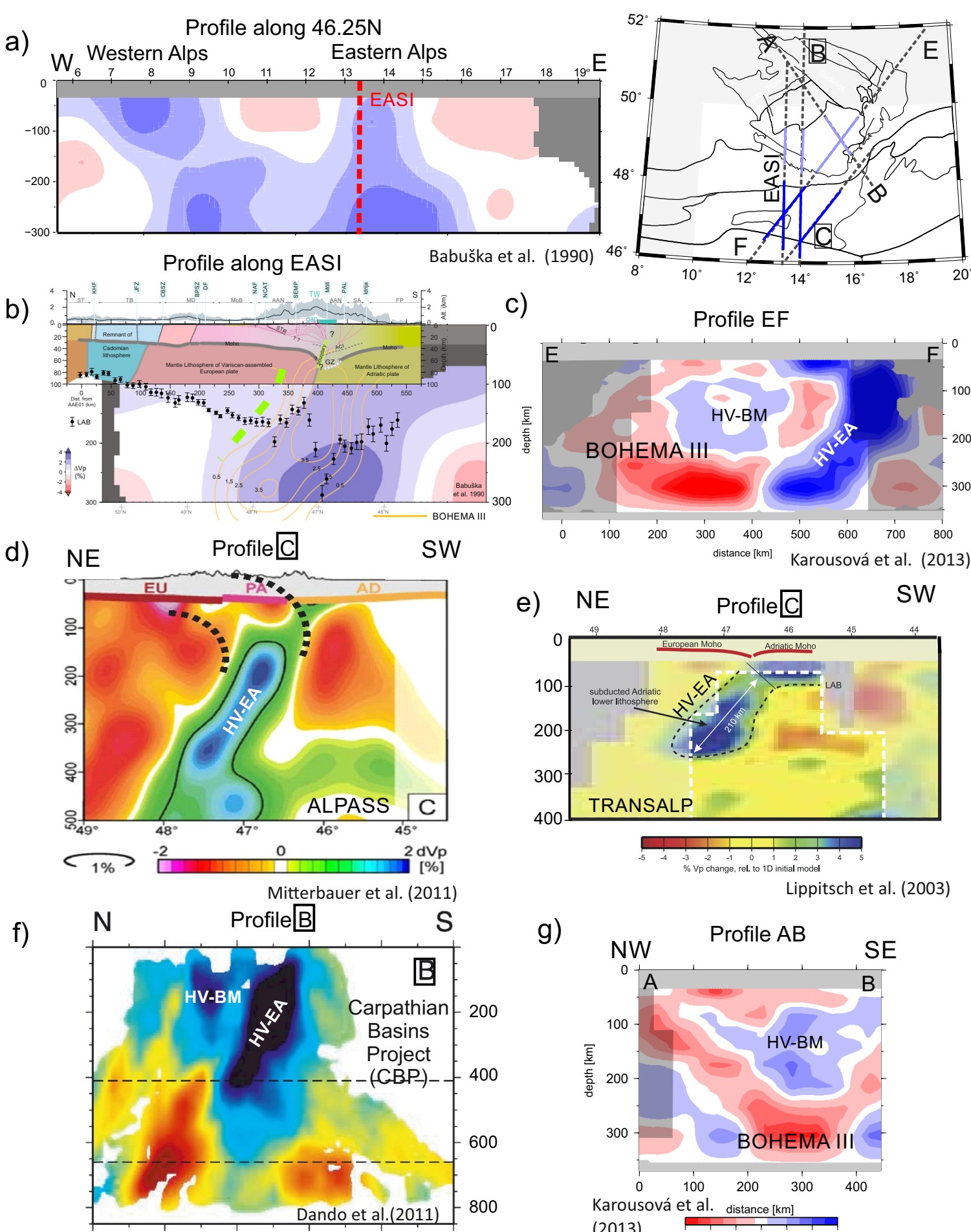

**Figure 5:** Cross-sections through various regional teleseismic P-wave tomography results showing segmentation of the Western and Eastern Alps and changes in the slab dip directions (a-b), the northward dip of the steep slab (HV-EA) beneath the Eastern Alps (b-f) and the small size HV_BM heterogeneity beneath the Bohemian Massif (c, f, g). Cross-sections (a-b) from Babuška et al. (1990), with the crust in part (b) from Hetényi et al. (2018b) and perturbation contours in (b) from Karousová et al et al. (2013). The profile locations in a map view are in the upper right. The dark- and light-blue highlights refer to the location of the positive perturbations beneath the E. Alps and the Bohemian Massif (HV-EA and HV-BM), respectively. The full set of colour palettes can be found in the original publications.

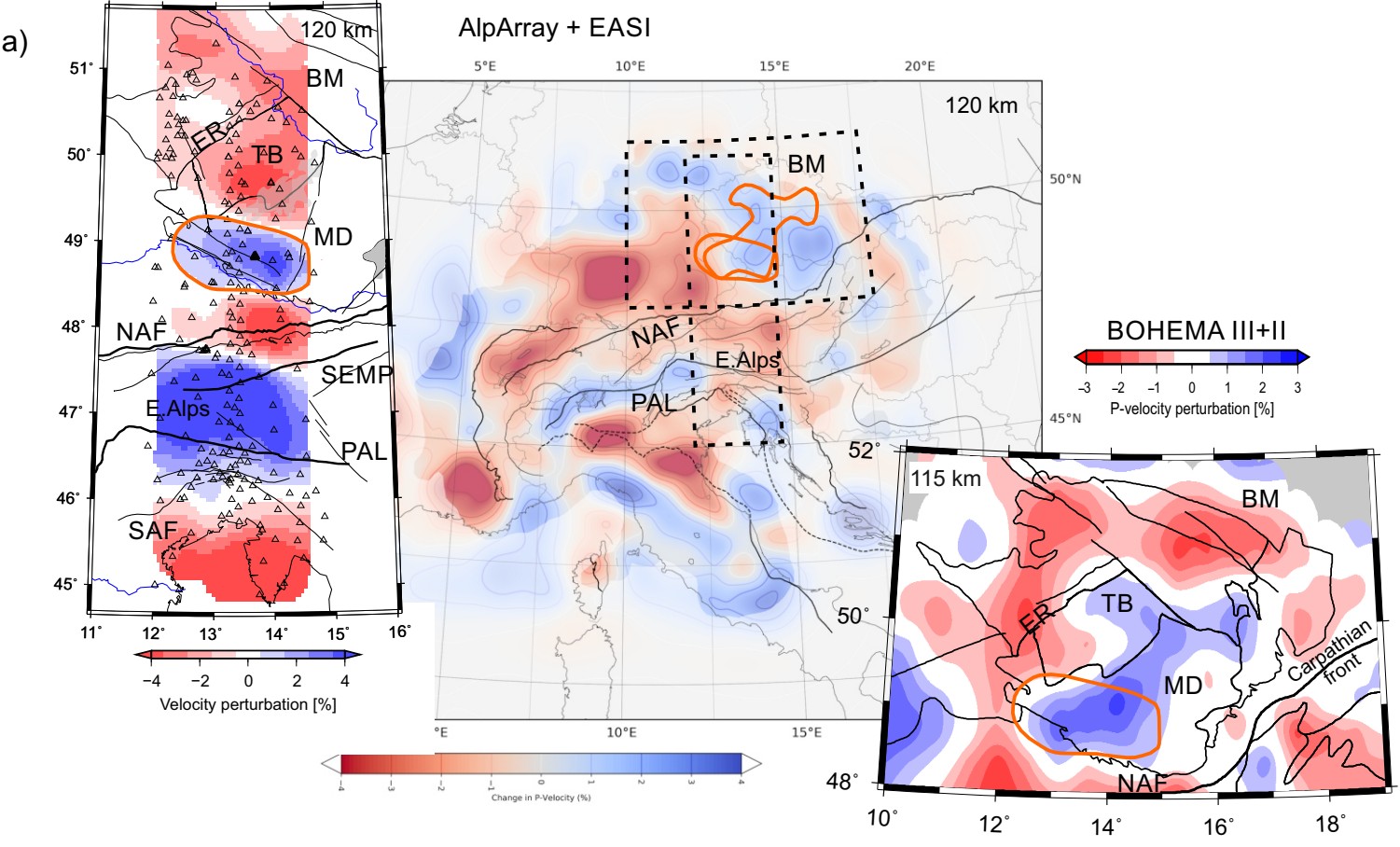

**Figure 6:** Horizontal slices at 120 km (a) and 180 km (b) depth through velocity-perturbation model EASI-AA (left) (this paper), model by Karousová et al. (2013) (right) and by Paffrath et al. (2021 this issue) (center), on which the HV-BM from the other two models is marked by orange contours. Labels as in Fig. 2.

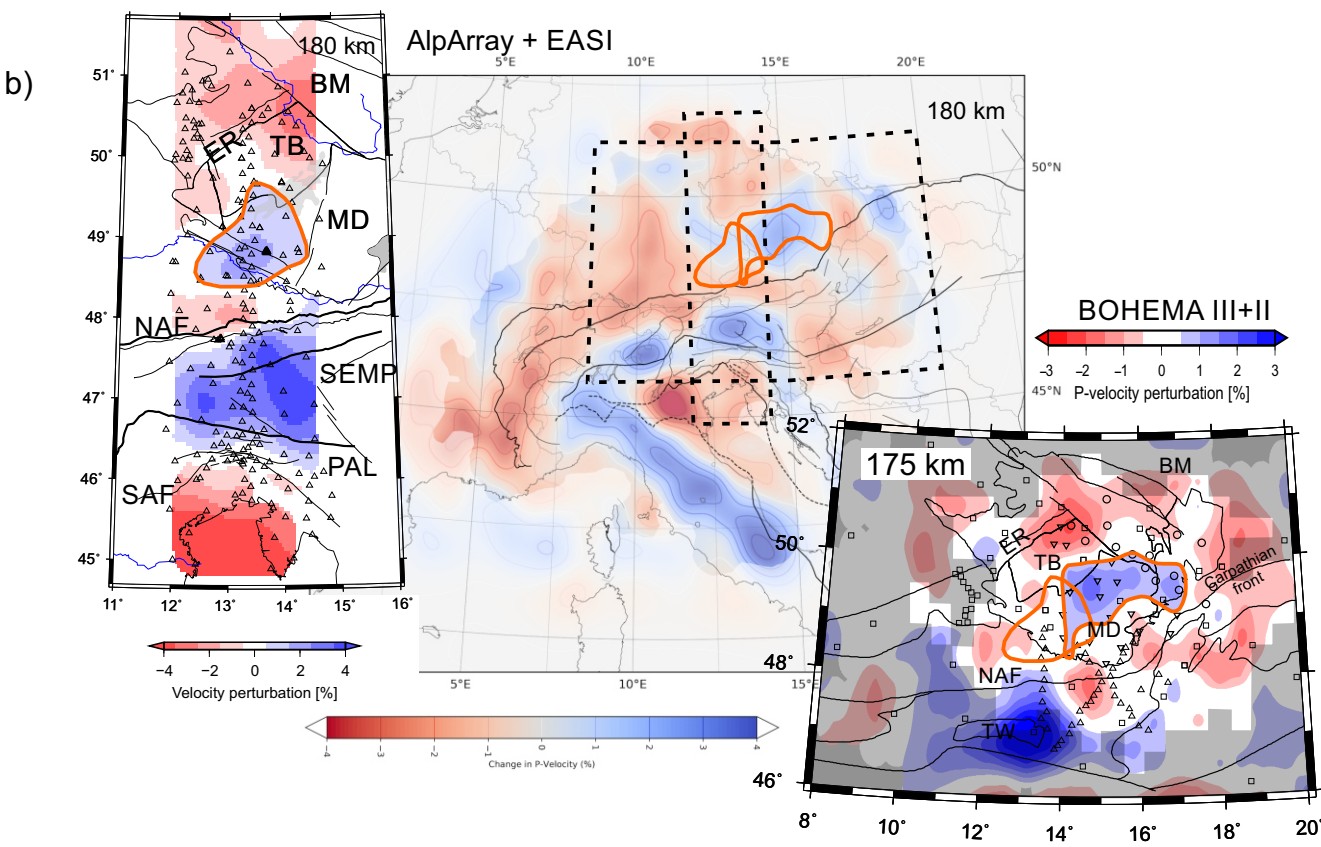

**Figure 6:** continuation part (b)

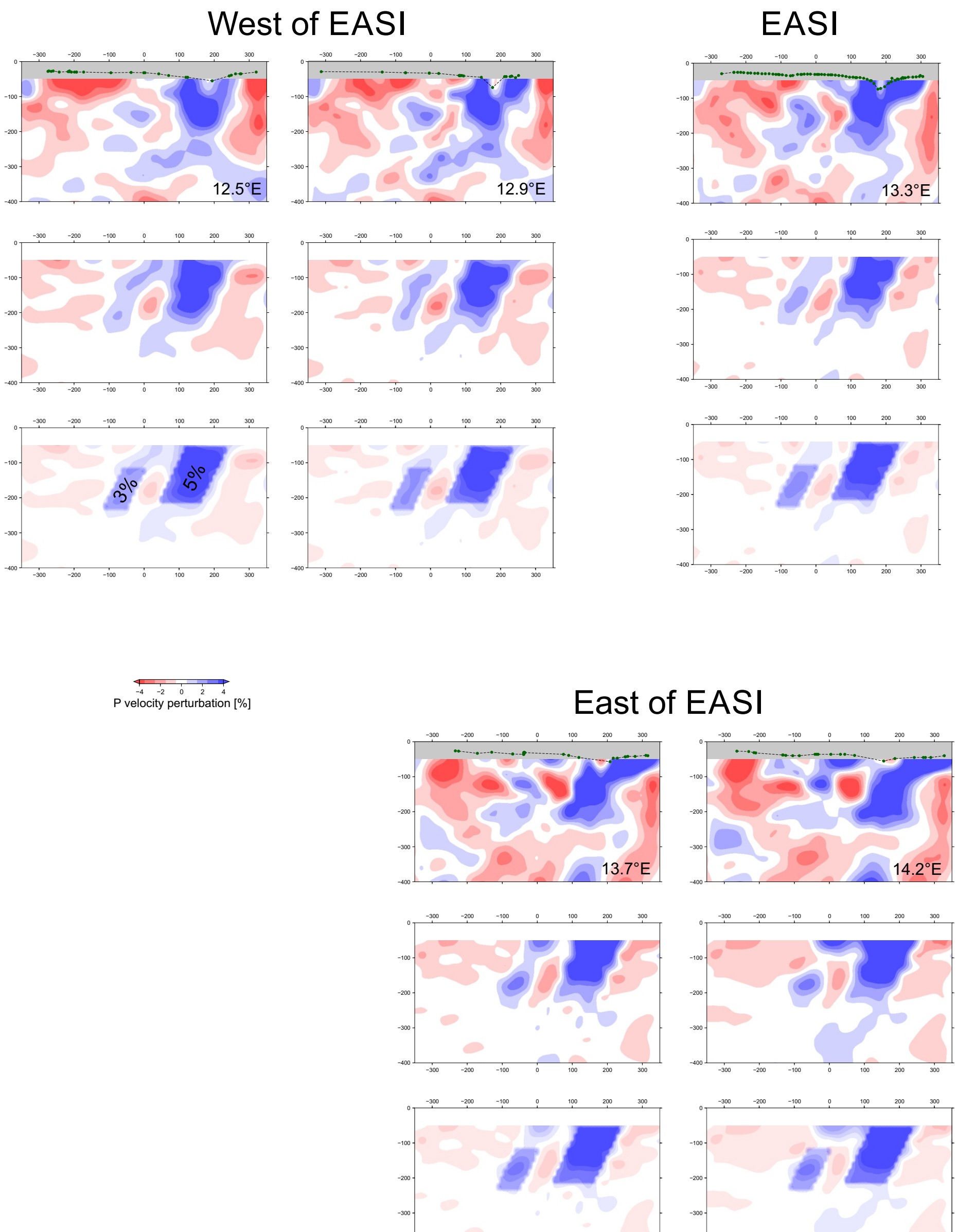

**Figure 7**: Velocity perturbations along the five vertical along-longitude cross-sections calculated from real data (upper row) and those (middle row) calculated from two synthetic 3% and 5% heterogeneities (bottom), plotted over the retrieved perturbations. Green dots mark Moho depths in the model used for calculation of crustal corrections (see also Fig. S3). The along-longitude cross-sections run from 51.65N to 45.35N.

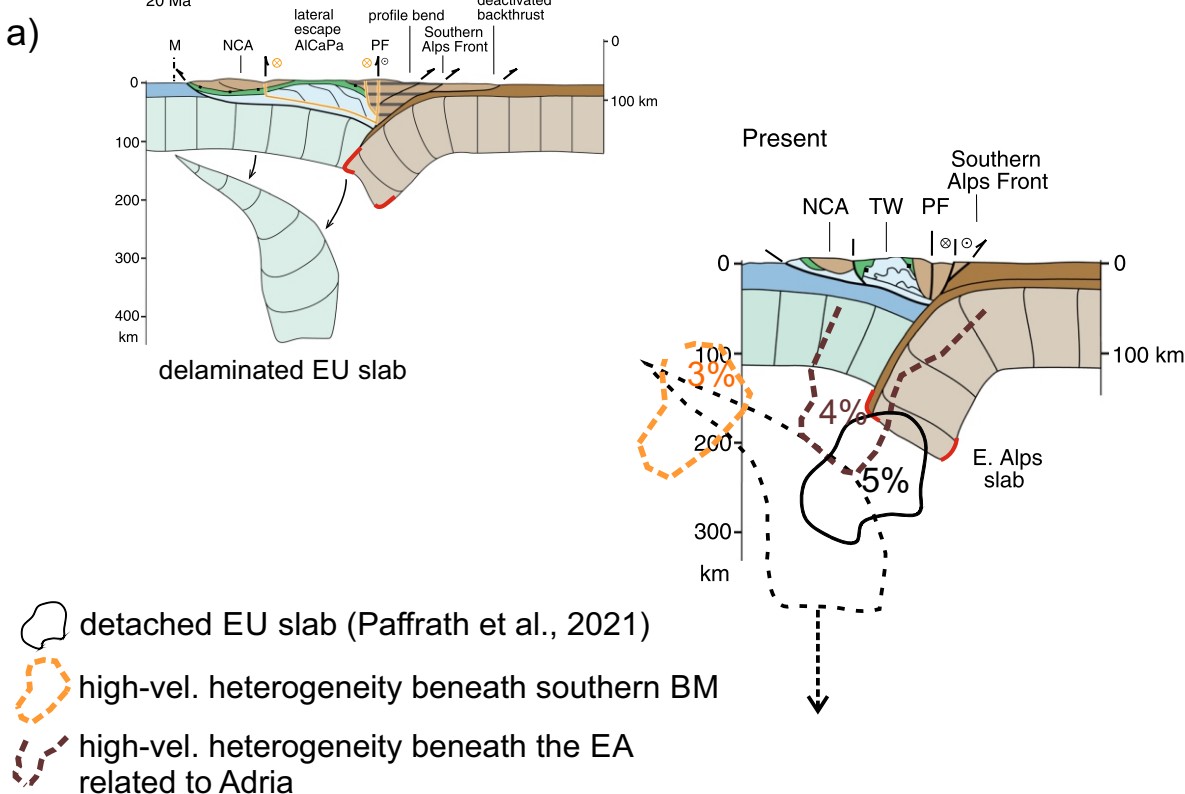

**Figure 8:** Interpretation of the slab at shallow depth beneath the Eastern Alps (HV-EA), (Adriatic vs. European provenance), and potential scenarios of the origin of the heterogeneity beneath south-eastern Bohemian Massif (HV-BM). (a) European slab delamination at 20Ma and configuration of the European-Adria plate collision at present redrawn from Handy et al. (2015) complemented by location of the high-velocity heterogeneities from the  EASI-AA (this paper) and the Paffrath et al. (2020) models; (b) a scenario considering closure of the Rheic ocean and collision of the Brunovistulian micro-plate with the Moldanubian part of the BM in a portion of schematic cartoon by Babuška and Plomerová (2013), with a piece of remnant lithosphere image as the HV-BM in this paper; (c) a scenario related to fragmentation of the Alpine and Carpathian front. Differences in the roll-back subductions of the Alps and the Carpathians  (e.g., Kissling and Schlunegger, 2018 and reference therein), northward push of Adria and European slab delamination beneath the E. Alps (Handy et al., 2015) could have formed complex flows in the asthenosphere (e.g., Vignaroli et al., 2008), which  might "transport"  lithosphere fragments toward the northwest.  For more details see the text.

## Assemblage of microplates in the Bohemian Massif

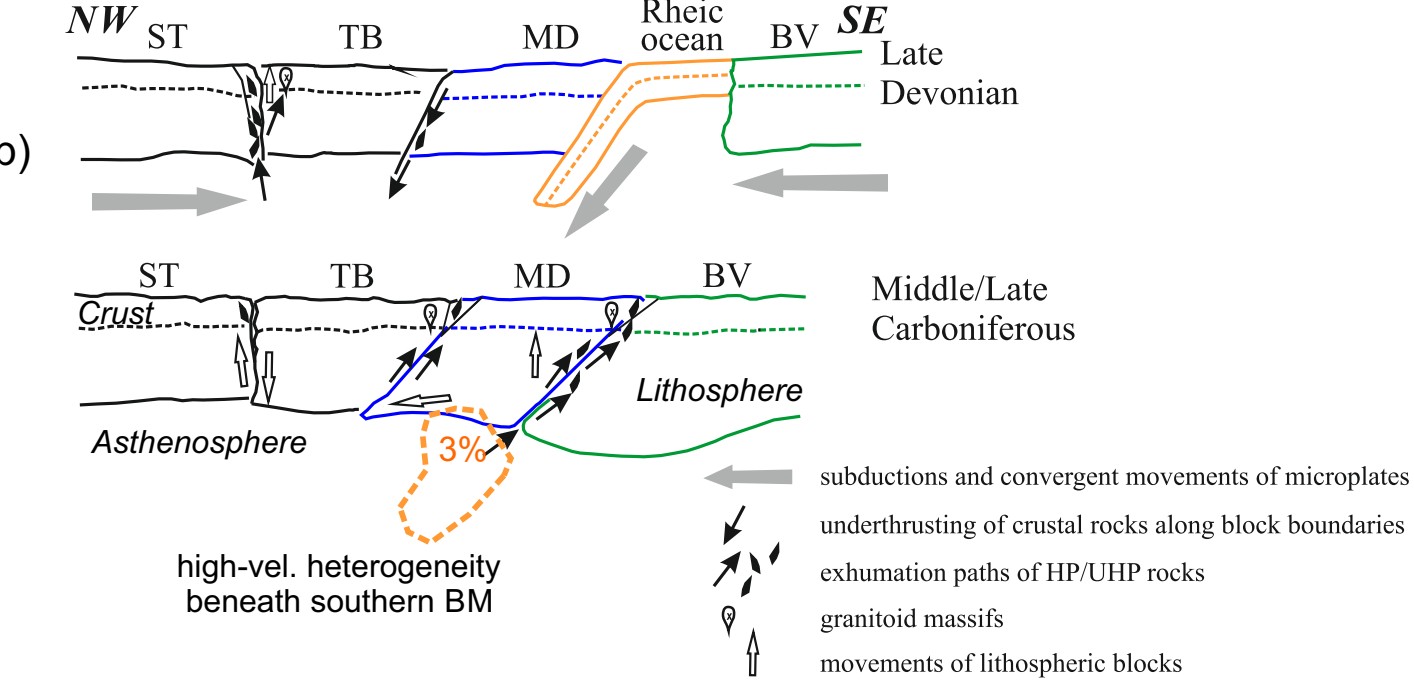

**Figure 8, part b.**

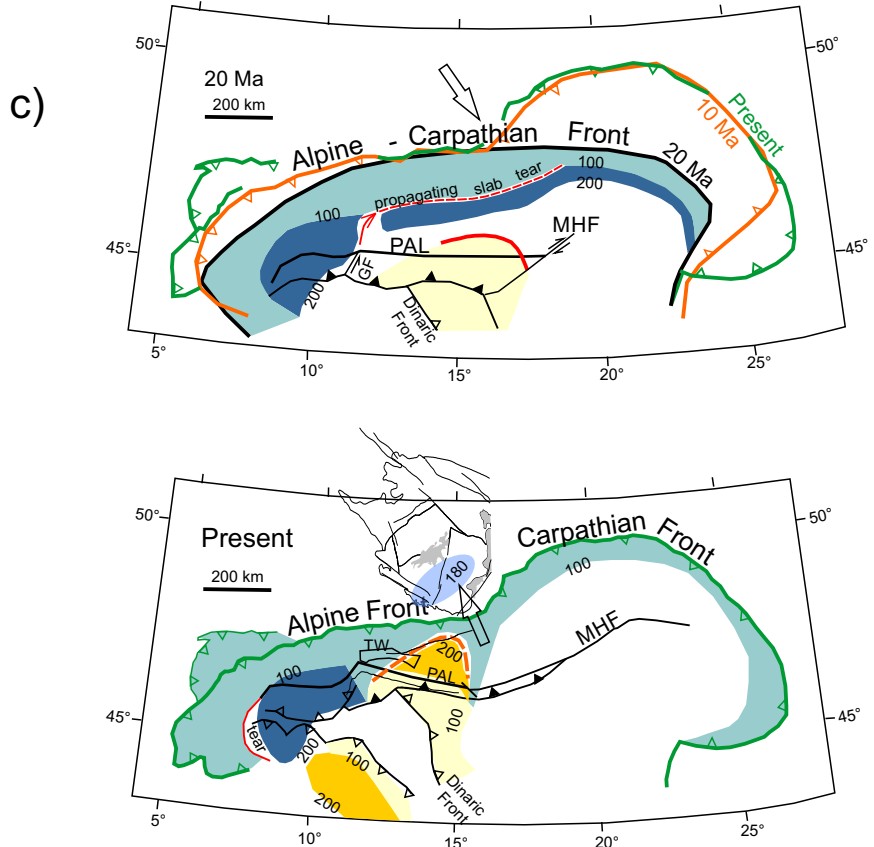

c)

Figure 8, part c.