# Peer review of "Two subduction-related heterogeneities beneath the Eastern Alps and the Bohemian Massif imaged by high-resolution P-wave tomography"

_Solid Earth, 2021_

## Referee Comment (RC1)

**Reviewer Comments on se-2021-56**
**Mark R. Handy**
**Black – lines and text from Plomerova et al.**
**Red – my comments**

This paper provides readers with new insight of seismological P-wave anomalies beneath the Alps and their eastern foreland. It employs AlpArray data as well as data from the EASI "swath" experiment across the Eastern and Southern Alps. As such it is a welcome contribution to the growing wealth of literature from these ambitious projects.

As a tectonist, I have concentrated on structural aspects of the authors' interpretation, leaving an assessment of seismological aspects to other qualified reviewers and the topical editor. My main criticisms pertain to the authors' interpretation/understanding of tectonic models, to their nomenclature and to the clarity of their text and figures. I have grouped my comments under headings relating to the main themes in the manuscript. My comments follow immediately on the pertinent lines in the manuscript.

Introduction:

Lines 31-32: "The classical concept assumed the European lithospheric slab subducted south-eastward/southward along the entire Alpine chain (Mueller, 1982) without any fragmentation." I would specify "Alps" rather than "the entire Alpine chain" because Mueller was referring only to the Alps sensu stricto, not to other parts of the Alpine chain which also includes the Apennines and Carpathians. The latter involve subduction of Adriatic lithosphere. It would also be appropriate to cite colleagues who previously proposed lithospheric subduction in the Alps (e.g., Laubscher 1970) inspired by the much earlier "classical" work of Ampferer and Argand.

Line 40: "…reversed polarity…" Here, I would recommend using the descriptive term "opposite dip" rather than "reverse polarity". The latter is more interpretative and implies a change in the direction of subduction.

Line 43: "….with a gap between the two Alpine keels…" Please define what you mean by "keel". You probably mean slabs. Note that Mitterbauer et al. (2011) preferred an interpretation in which there is no slab gap between the Central and Eastern Alps.

Line 58: Nomenclature – "the south-eastern part of the BM (referred to as HV-BM throughout the paper)…" Please keep things simple by using BM = Bohemian Massif. Or if you really need this long abbreviation (HV-BM), then write it completely to make your usage clear.

Resolution tests:

Lines 201-205: "The above performed synthetic tests corroborate that the data from the AlpArray and EASI networks are able to image two separate northward dipping sub-parallel slabs beneath the E. Alps and southern rim of the BM. The two slabs are separate from each other, and the northern one is not connected with the shallow parts of the lithosphere (above ~100km). The flip of the subduction polarity beneath the E. Alps relative to the W. Alps is undoubtedly real and it is not produced by potential smearing due to ray geometry." I would distinguish between slab dip (the direction that the slab is presently slanted) and subduction polarity (the original direction of subduction). The steep northward dip of the slab beneath the Eastern Alps is a robust feature of all models so far, but the original direction of subduction is debatable (see comments below). For comparison of end-member models with different slab dips, e.g., resolution tests of Paffrath et al. (in rev.).

Connection of surface to subsurface dynamics:
Lines 226-229: "The uplift rates in the W. and C. Alps exhibit at least 50% contribution by convective processes (due to slab detachment) and dynamic contributions (due to the sub-lithospheric mantle flow), while isostatic response due to ice unloading during deglaciations dominates in the E. Alps (Sternai, 2019)." There are other modelling studies that differ from those of Sternai 2019 in both their approach and their conclusions; they warrant reading and citation in this context (e.g., Champognac et al. 2007, Fox et al. 2015, Mey et al. 2016).

Slab polarity and the boundary between the Eastern and Central Alps:
Lines. 222-224: "In recent studies, Paffrath et al. (2020) suggest the reversed slab polarity relative to the Western Alps already in the Central Alps, as opposed to the formerly documented polarity reversal further to the east - beneath the E. Alps (e.g., Babuška et al., 1990, Lippitsch 2003; Zhao et al., 2016)." This is a misunderstanding/misreading, because Paffrath et al. don't claim that there was a reversal in slab polarity below the Central Alps! Please read carefully. Rather, they refer to the part of the slab beneath the Eastern Alps when they write „The new model does not require a slab polarity switch due to the detachment from the crust and the nearly vertical dip of the eastern Alpine slab. Tectonic arguments (in Handy et al., in rev.  with Solid Earth) rather suggest a European provenance of the slab".

Lines 224-226: "Mock et al. (2020) pointed out the discordance between the slab geometry at depth and the boundary between the Eastern and Central Alps observed in the surface geology and, similarly to Rosenberg et al. (2018), shift the boundary between the E. and C. Alps further to the east, at the Giudicare-Brenner fault system." The authors cited do not redefine the boundary between Eastern and Central Alps, they merely note a discordance between the slab geometry at depth and the boundary between the Eastern and Central Alps observed at the surface. The boundary between the E. and C. Alps is defined at the surface in map view by the western edge of the continuous Austroalpine nappe stack located in Eastern Switzerland. This traditional definition does not take into account the effect of erosion which removed Austroalpine nappes originally exposed somewhat further to the west. The former existance of these nappes to the west is supported by the provenance of components in the Molasse Basin and by the Austroalpine Klippe in the Central Alps. Regarding the Alpine slab, the boundary between the central and eastern slab segments (as defined either by a slab gap or a change in dip direction of the slab) coincides generally with the Giudicarie-Brenner faults. In light of recent evidence for a laterally continuous Alpine slab with a change in dip (Handy et al., in review with SE), it would be clearer to avoid relating surface and subsurface boundaries between Central and Eastern Alps.

Lines 238-241: "Later tomography of the upper mantle which included the E. Alps from data of regional passive experiments (Dando et al., 2011; Mitterbauer et al., 2011; Karousová et al., 2013) also retrieved the northward dipping high-velocity heterogeneity of similar geometries (Fig. 5) and associated it mostly with the Adria plate subduction." It is important to avoid equating a switch in the dip direction (polarity) of the Alpine slab with a gap in this slab. In the interpretation of Handy et al. (in rev. with Solid Earth), the slab changes its dip direction along strike without evidence for lateral separation into two segments. Rather, the along-strike change in dip direction of the slab coincides generally with an eastward transition in its degree of detachment from the orogenic lithosphere.

Lines 241-246: "....suggested three main phases in building the E. Alps keel: (1) NW translation of the Adria and its thrusting over the (subducting?) European plate in the W. Alps, (2) fragmentation of northern Adria along a deep-seated fault (possibly the Giudicarie Fault, or at least a spatially nearby structure) and (3) counter-clockwise rotation of the Adria and its subduction beneath the European

plate in the Eastern Alps, with a triple-junction of three crustal terranes in its eastern rim proposed by Brückl et al. (2010), although the deformation style between the E. Alps and the Pannonian Basin is usually considered diffuse on the surface." Two points need clarification here: (1) Based on recent P-wave tomography from the newest Alp Array data (Paffrath et al. in rev), the length of slab beneath the Eastern Alps, irrespective of its dip direction, far exceeds the amount of shortening in the eastern Southern Alps. This precludes a primarily Adriatic origin of this slab, as discussed in Handy et al. in rev, Solid Earth; (2) there is no unequivocal evidence at the surface or at depth to support the idea of three "crustal terrains" with distinct Moho boundaries near the transition from E. Alps to the Pannonian Basin. Rather, the Pannonian Basin is underlain by thinned European and Adriatic crust and lithosphere that acquired their reduced thicknesses in Miocene time (e.g., Ustaszewski et al. 2008). The decrease in Moho depth going from the Eastern Alps to the Pannonian Basin is gradual, not abrupt (e.g., Horvath et al. 2015), indicating extensional flow of the lower crust and upper mantle.

Lines 267-277: "The north oriented dip of the EA subduction, imaged in early tomography studies…..Most of them agree in interpreting it being of Adriatic plate origin (Dando et al., 2011; Karousová et al., 2013). However, Mitterbauer et al. (2011), and similarly in recent tomography by Paffrath et al., (2021 this issue), the positive perturbations are associated with a delaminated EU slab. Kästle et al. (2020) relate the HV-EA mainly with the European plate subductions as well, and leave no or only minor role to the Adriatic subduction. The authors explain the northward subduction modelled beneath the E. Alps by imaging a combination of the short Adriatic and deep delaminated, potentially overturned European slabs." Here, the authors should mention that recent P-wave tomography based on the newest Alp Array data (Paffrath et al. in rev) supports a primarily European origin of the north-dipping slab beneath the Eastern Alps (see discussion in Handy et al., in rev. with Solid Earth). This conclusion is based on the length of imaged slab beneath the Eastern Alps (several hundred km) which far exceeds the amount of shortening (c. 50 km) in the overlying crust of the eastern Southern Alps. In this interpretation, the steep northward dip of the slab segment beneath the E. Alps is attributed to asthenospheric flow during or after Adriatic indentation (Handy et al., rev.).

Lines 401-402: "The northward-dipping lithosphere keel is imaged down to ~200-250 km, without signs of delamination, and we associate it with the Adriatic plate subduction." As above, it is unclear what the authors mean by "keel". It appears to mean slab, but this should be clarified.

Interpretations of anomalies beneath the Bohemian Massif:
Lines 281-285: "To understand the positive perturbations beneath the southern BM, we compare it with results from the large-scale Paffrath et al. (20201) tomography and with the regional tomography of the BM (Karousová et al., 2013) of similar resolution to ours (Fig. 6). The strongest positive perturbations related to that heterogeneity overlap in the models, though they are of unrealistically large extent in the Paffrath's et al. model (2021, this issue)." Why is this extent in the Paffrath et al. model unrealistic? "There, they include also the ER with the thinnest BM lithosphere, well imaged by negative perturbations in the EASI-AA and in BOHEMA models (Karousová et al., 2013)."

Lines 308-311: "The role of applying proper crustal corrections is significant in teleseismic regional tomography. Not applying any crustal corrections or applying inadequate ones can strongly affect velocity perturbations within the upper ~100 km of the upper mantle (e.g., Karousová et al., 2013), which is the zone, where the models discussed above differ. From this point of view, developing a uniform detailed and reliable model of the European crust is urgently needed." I agree and find the comparison of models in this chapter valuable. I've sent the authors some PPT slides of a recent talk

showing the correspondence between models regarding the degree of connectivity of slabs with their orogenic lithosphere. Both the recent P-wave tomography employing 3D crustal models (Paffrath et al. in rev.) and the surface wave models of Kaestle et al. (2018, 2020) indicate slab detachment in the Western and Eastern Alps, with local slab attachment only in the western Central Alps. This coincidence of images with different methods suggests that the crustal correction of the P-wave model of Paffrath et al. is robust.

Lines 332-333: "The simplest explanation would be to consider it as a fragment of the delaminated part of the European plate subductions, as suggested in Handy et al. (2015) (Fig.7a )." This appears to be a misunderstanding: The +Vp anomaly below the BM is not the anomaly referred to in Handy et al. 2015 (their Fig. 11a), which was a European slab hanging directly below the Eastern Alps in Miocene time. Note that Handy et al. 2015 never considered a +Vp anomaly below the BM in their model. Instead, the Miocene +Vp anomaly below the Eastern Alps in their figure is torn European lithosphere that sank and no longer exists. The model of Handy et al. 2015 is actually consistent with the statement in Line 338 that "Therefore, an association of the HV-BM with the delaminated fragment of the EU subduction is not likely."

Points pertaining to rock physics:
Lines 347-348: "The Phanerozoic continental mantle lithosphere, composed of originally lighter rocks than those in the asthenosphere (Mantle lithosphere is generally denser rather than lighter than asthenospheric mantle) and becomes denser due to metamorphic phase changes as it subducts" (reference?)

Lines 360-363: "The roll-back subduction of the Carpathians, accompanied by a substantial asthenospheric flow, could open a space between the E. Alpine and Carpathian slabs for the northwestward "transportation" of a purely oceanic lithosphere or a mix of oceanic and continental lithosphere fragment into the mantle beneath the BM." No reason or mechanism is provided here to explain why oceanic or mixed ocean-continent lithosphere should be "transported" northward. Please elaborate.

Lines 380-396: The authors emphasize the importance of accounting for seismic anisotropy when interpreting body-wave tomography: "Ignoring seismic anisotropy and assuming isotropic wave propagation or considering only azimuthal and/or radial anisotropy leads to significant isotropic and anisotropic imaging artefacts that may lead to spurious interpretations (Vanderbeek and Faccenda, 2021)." The reader is left without a clear explanation of how anisotropy would change the images of +Vp anomalies which are the basis for interpretations. It would be helpful to discuss this more specifically instead of making vague references to future studies in Lines 392-395.

Figures and Captions: This needs significant improvement
**Fig. 1** (right side): Label the main faults and use transparent colour to mark the Alps between the Northern and Southern Alpine Fronts.
**Fig. 2** (right panel): Label the Southern Alpine Front (SAF), use transparent colour to mark the Alps between the Northern and Southern Alpine Fronts. It would help to mark the outlines of the Tauern Window (TW). I would eliminate the triple junction, because there is no structural evidence for a Pannonian Plate sensu Brückl et al. (see comments above).
**Fig. 3**: Label the Southern Alpine Front (SAF) on all cross sections. In the caption, indicate what DF means (this is not included in any of the earlier captions). Indicate the traces of all cross sections on a map or inset map. Include coordinates of the endpoints in the caption or directly on the figures. This is important for others who wish to compare their results with the authors'.

**Fig. 4**: Label all the faults shown in the other cross sections. Include coordinates of the endpoints in caption or directly on the figures. Again, this is important for others who wish to compare their results with the authors'.

**Fig. 5**: The labelling of the cross sections should be made clearer. Label each section only once so it corresponds to the letters of the traces on the map. The orogen-parallel cross section in a) should have the longitudinal degrees correspond to the longitudinal units on the map on the lower right.

**Fig. 6**: Add Variscan unit boundaries on the main horizontal slice of Paffrath to enable better comparison with the other slices shown. Distinguish Alpine and Variscan boundaries, either by using lines with different thicknesses (Alps thick, Variscan thin) or colours.

**Fig. 7**: Check the position of the cross sections with respect to the seismic anomalies, because they look misaligned. The northern tip of the delaminated EU slab in a) is too far to the north in b) with respect to the Northern Alpine Front (NAF). Handy et al. 2015 proposed that this slab sank beneath the core of the Alps, not that it migrated to the north with respect to the front. However, if this is your idea (meaning the authors of this text), then please state this clearly.

**Fig. 8**: Show the end points of the cross sections on an inset map so that the location of the. anomalies and unit boundaries can be compared, otherwise the model can't be compared with other data and the reader is helplessly lost. In b), please also consider other interpretations of Late Devonian to late Carboniferous paleogeography and subduction directions of the Rheic Ocean, e.g., Schulmann et al. 2009, 2014, Ballevre et al. (2009), Zeh & Will (2011) or in Franke et al. (2000). What effect do these scenarios have on your interpretation of the 3% positive Vp anomaly beneath the Moldanubikum?

---

## Referee Comment (RC2)

**Review Plomerova et al., SE-2021-56**

The paper presents a teleseismic tomography of travel time residuals determined from records of the AlpArray backbone and the complementary experiment EASI that focuses on a narrow NS swath along the 13.3 E meridian. Most notable features in the derived model of P-wave velocity are two parallel positive anomalies dipping towards the north, one below the Alpine chain and one below the Bohemian Massif. The results provide new fuel for the long-lasting debate on slab polarity beneath the Alps and deserve publication after a substantial revision.

As a seismologist I have concentrated on aspects of data processing and inversion. There are several severe issues regarding the inversion procedure, the correction for crustal structure and the resolution tests that need to be dealt with. Moreover, data, data processing, uncertainties and crustal model need to be documented in more detail. Finally, the paper deserves a much better organization of thoughts and arguments in particular in the introduction and section 6.

**Introduction:**

The introduction is surprisingly short. I miss a proper description of the state of the art (what is known), of controversial issues (what is debated) and of how the manuscript contributes to answering some of these open questions. The introduction does not pose any research questions and thus entirely lacks a motivation for the work presented. I also miss a proper description of previous work done with EASI data and its relevance for the current work. At the end, a brief description of the contents of the paper would be welcome. Instead, the last paragraph rather resembles an abstract.

The meaning of the lines in the tectonic map in Fig 1 needs to be explained in the caption.

**Data:**

I assume that also data of the complementary EASI experiment are used. This should be made clear in the text.

Which procedures for checking data quality have been applied? The paper should be self-contained to some extent.

The description of the picking scheme and the event distributions should be put into main paper. Which trace is used as reference for cross-correlation and beam forming? Quantitative statements regarding error estimation of the picks and their probabilistic combinations should be made.

The description of the enhanced data set is confusing. Is the enhanced dataset just the original one with 201 events (in Figure S1 209 events) plus another 43 events within the 60 degree back-azimuth cone? Or is it a completely new selection from the overall available events containing only events within the 60 degree cone? This should be clarified in the manuscript. Regarding the figures S1, the latter seems to be the case.

There are two figures S1 in the supplement. Renumber to S1 and S2.

Information on crustal structure is taken from several sources. How is the 3D crustal model put together? How are transitions between subregions treated? How are crustal discontinuities treated, as transition regions or as real discontinuities? At least two horizontal sections through the 3D crustal velocity model should be shown. To which depth does the crustal model reach? Does it also contain parts of the uppermost mantle?

How is the correction for crustal structure done? One correction for all events assuming vertical incidence or event-wise corrections taking correct incidence angles into account?

The meaning of the end of the following sentence (line 100) ".... proper tomographic inversion in our target region to resolve structures, including the spatial limits of our images" is unclear? What are the spatial limits of the images? Do the authors want to say that resolution is also good at the model boundaries?

Estimated uncertainties of the travel time residuals should be documented in the manuscript. Show a histogram of the uncertainty distribution and give average and median values

**Method:**

It is stated that the matrix $W_m$ in equation (4) provides horizontal smoothing. What about vertical smoothing? Is there also a damping term included in $W_m$, and if yes, what is the weight of smoothing relative to damping?

With 13 cells in vertical direction, I calculate a model depth of 13*30=390 km and not 435 km as stated in the text. Is there a specific reason for choosing 390 km (or 435 km) as the bottom of the model. Is this depth still warranted by intersection of rays given the length of the EASI profile?

Confining the inversion domain to a narrow stripe in NS-direction causes a problem for rays incident from easterly or westerly directions. They can carry significant travel time residuals but only pass through the uppermost parts of the inversion domain. Thus, the entire residual accumulated along the path through the mantle beneath the Alps needs to be explained by heterogeneities in the uppermost part of the model. This is probably impossible and definitely undesired. Basically, the travel time residual of the rays incident form the side could be much more easily explained by heterogeneities outside the model domain and hence do only bring little information for the uppermost model parts. A remedy for this problem could either be to discard the rays from the side or to extend the inversion domain in EW-direction to allow heterogeneities there. They will not be interpretable because of the lack of ray crossing but at least they inhibit a mapping of the entire travel time residual they carry into the uppermost parts of the model. This problem could be one reason for the rather low misfit reduction of 66 percent. Restricting ray incidence to a cone around the NS direction as done for data set 2 may mitigate this problem. What is the rationale behind choosing a 60 degree cone for data set 2?

What is the damping factor in Fig. S4? Is it epsilon**2 from equation (4)? Why do the authors plot data variance instead of misfit which is normalized to the data uncertainties. What is model variance? Is it the squared norm of the velocity perturbations? A definition of this quantity should be provided. The authors should provide a value of misfit normalized to the picking uncertainties to get an impression whether there is overfitting or maybe even severe underfitting. Regarding Fig. S4. I wonder that the authors get increasing data variance and decreasing model variance for decreasing damping factor. I would expect exactly the opposite. This strange behaviour needs to be explained by the authors.

Why do the authors only consider 2 iterations? Did they try more iterations and how does the inversion behave then?

**Results:**

In the "Data" section two dataset were presented. Which one was used to create the model shown in the figures? Fig. S5 which compares vertical sections obtained from the 2 data sets is only mentioned once. The issue is never discussed in the results or the interpretative section.

Regarding the vertical sections in fig. 3, I recognize significant perturbations in the gray-shaded upper 50 km of the model which appears to be the crustal domain. As the effect of the crust was subtracted from the travel-time residuals, why do the authors still allow perturbations in the upper 50 km? Theoretically, after crustal correction, the travel-time residuals should represent pure mantle structure and perturbations in the crust should be suppressed. How would the model change if perturbations in the upper 50 km were forced to zero? In particular, what would happen with the model between 50 km and 100 km depth, just beneath the crustal domain? The interpretation of a northward dip strongly depends on the velocity perturbations between 50 km and 100 km depth. Below the high velocity anomalies are rather vertical. Whether HV-EA is delaminated or not, also depends on the anomalies in this depth range. All the conclusions following in the paper about polarity flip, northward dip and detachment of the Eastern Alpine slab depend on this issue.

For a connection to crustal levels, one could plot the perturbations of the crustal model relative to a 1D-model of the crust into the vertical cross sections. But then, what to do with the anomalies that are already there?

What is the criterion for gray-shading in the model domain? Is it derived from the resolution matrix?

**Resolution tests**:

I would also like to see the results from a checkerboard test (with gaps) which nicely shows lateral and vertical smearing. In particular vertical smearing at shallow levels should be investigated because it may hide a detachment of the slab or falsely connect the high vp anomaly below 100 km with a shallower one further to the south giving the impression of a significant northward dip.

The remark that the polarity flip is more or less accepted is certainly a misconception. If I read Paffrath et al. (2021) correctly, they see a rather vertical, detached eastern Alpine slab and favour the interpretation of European provenance because its down-dip length can only be explained by the Tertiary shortening in the Eastern Alps accommodated by south-dipping subduction of European lithosphere. Basically, what is seen in the tomography is only the slab dip but tomography does not tell us the provenance of the slab. Even if it were dipping clearly northwards (in the tomographies by Mitterbauer (P4), Zhao and Paffrath, it is nearly vertical) it could still be overturned European lithosphere. Independent data are needed to decide this issue. The use of the term "polarity flip", however, already implies the interpretation of Adriatic provenance of the slab.

Resolution test 1 also mimics the real pattern quite well below 100 km depth while test 3 misses the increasing dip angle of the real pattern with depth. My remark regarding the treatment of crustal structure also applies to the resolution test. Since crustal corrections were subtracted before inversion of the real data, resolution test data should also be free of crustal contributions and anomalies in the crustal domain should be forced to zero.

**Imaging the high-velocity perturbations in different tomography model:**

$2^{nd}$ paragraph: If I read Paffrath et al (2021) correctly they do not postulate a polarity flip for the Central Alpine slab. They state a steep SE dip of the slab and also do not associate it with Adria. This should be corrected.

In general, I find this section not well structured. The polarity flip issue is discussed in several paragraphs interrupted by a discussion of the $2^{nd}$ high velocity anomaly underneath the BM. There seems to be no real ordering of thoughts and arguments. This part should be streamlined and rewritten in a concise and non-repetitive way with clear order and structure of arguments.

I am strongly worried by the resolution tests shown in Fig. S7. For each one of the detached slab test models there are strong smearing artifacts reaching up to the surface. The artifacts are strongest in the crustal layer and they mimic an either northward or southward dipping continuous slab. I cannot see that this resolution test proves that a detachment can be resolved by the inversion. It rather suggests the opposite. I wonder how the result of these tests would look like if crustal perturbations were forced to zero. Possibly, all artificial positive anomalies between 50 km and 100 km could become much stronger pretending a dipping high velocity feature.

---

## Author Comment (AC4)

Dear Reviewer 1,

Thank you for your comment on our manuscript. We report all your comments in red below, followed by our respective answers. The main manuscript is revised accordingly, some of the modified sentences are include below (underlined text).

Reply to RC1

Introduction:
**Lines 31-32:** "The classical concept assumed the European lithospheric slab subducted southeastward/southward along the entire Alpine chain (Mueller, 1982) without any fragmentation."

I would specify "Alps" rather than "the entire Alpine chain" because Mueller was referring only to the Alps sensu stricto, not to other parts of the Alpine chain which also includes the Apennines and Carpathians. The latter involve subduction of Adriatic lithosphere. It would also be appropriate to cite colleagues who previously proposed lithospheric subduction in the Alps (e.g., Laubscher 1970) inspired by the much earlier "classical" work of Ampferer and Argand.

**Reply:** We have modified the sentence as suggested:

The classical concept assumed the European lithospheric slab subducted south-eastward/southward along the entire Alps (Laubscher 1975; Mueller, 1982) without any segmentation.

*Line 40:* "…reversed polarity…" Here, I would recommend using the descriptive term "opposite dip" rather than "reverse polarity". The latter is more interpretative and implies a change in the direction of subduction.

**Reply:** We see your point and have modified the sentence as suggested, though "opposite" is not the exact expression and the term "reversed polarity" in sense of a change in dip direction is generally used in a broader sense than suggested in the review (e.g., Vignarolli et al., 2008). We substitute the word fragment, fragmented with more propper segment/segmented throughout most of the text.

For the first time, Babuška et al. (1990) imaged the Alpine slabs separated into two segments, one beneath the Western and second beneath the Eastern Alps, with opposite polarity and a gap in between them.

*Line 43:* "….with a gap between the two Alpine keels…" Please define what you mean by "keel". You probably mean slabs. Note that Mitterbauer et al. (2011) preferred an interpretation in which there is no slab gap between the Central and Eastern Alps.

**Reply:** we change word keel to slab as suggested, though the word keel is often used in many papers for dipping/thickened edge of the lithosphere (the whole lithosphere, not only crust; in that case the word "root" is usually used). We attempted to avoid frequent usage of word slab. We have seen the (vertical) slab gap even in our old studies in the 80s, which was later documented in other papers as well, e.g., Lippitsch et al. (2003).

New wording:

…. with a gap between the two Alpine slabs,……

*Line 58:* Nomenclature – "the south-eastern part of the BM (referred to as HV-BM throughout the paper)…" Please keep things simple by using BM = Bohemian Massif. Or if you really need this long abbreviation (HV-BM), then write it completely to make your usage clear.

**Reply:** Abbreviation "BM" is introduced for the Bohemian Massif as a whole in the abstract of the original submission (line 15) and Introduction (line 57). Abbreviation HV-BM is introduced as a logical expression for the "high-velocity heterogeneity beneath the southern part of the BM", i.e., only a part of the BM and at depth. Repeating "high-velocity heterogeneity beneath the southern part of the BM" throughout the text would be long and non-effective. Moreover, not all the BM mantle is fast, but rather slow (e.g., Piromallo & Morelli 2003; Amaru 2007; Koulakov et al. 2009). Using only BM for the high-velocity heterogeneity beneath a part of the BM would be confusing for those who are familiar with the slow sub-lithospheric upper mantle beneath the BM, in general. Therefore, we will keep the abbreviation HV-BM and complement the captions.

**Resolution tests:**

*Lines 201-205*: "The above performed synthetic tests corroborate that the data from the AlpArray and EASI networks are able to image two separate northward dipping sub-parallel slabs beneath the E. Alps and southern rim of the BM. The two slabs are separate from each other, and the northern one is not connected with the shallow parts of the lithosphere (above ~100km). The flip of the subduction polarity beneath the E. Alps relative to the W. Alps is undoubtedly real and it is not produced by potential smearing due to ray geometry." I would distinguish between slab dip (the direction that the slab is presently slanted) and subduction polarity (the original direction of subduction). The steep northward dip of the slab beneath the Eastern Alps is a robust feature of all models so far, but the original direction of subduction is debatable (see comments below). For comparison of end-member models with different slab dips, e.g., resolution tests of Paffrath et al. (in rev.).

**Reply:** We modify the sentence as suggested,

The difference between the dip directions of the subduction beneath the E. Alps and the subductions beneath the W. and Central Alps is undoubtedly real and it is not produced by potential smearing due to ray geometry.

**Connection of surface to subsurface dynamics:**
*Lines 226-229:* "The uplift rates in the W. and C. Alps exhibit at least 50% contribution by convective processes (due to slab detachment) and dynamic contributions (due to the sub-lithospheric mantle flow), while isostatic response due to ice unloading during deglaciations dominates in the E. Alps (Sternai, 2019)." There are other modelling studies that differ from those of Sternai 2019 in both their approach and their conclusions; they warrant reading and citation in this context (e.g., Champognac et al. 2007, Fox et al. 2015, Mey et al. 2016).

**Reply:** We chose to delete this paragraph, which does not concern the main topic of the study. More explanation would make the ms. unnecessarily longer.

**Slab polarity and the boundary between the Eastern and Central Alps:**
*Lines 222-224:* "In recent studies, Paffrath et al. (2020) suggest the reversed slab polarity relative to the Western Alps already in the Central Alps, as opposed to the formerly documented polarity reversal further to the east - beneath the E. Alps (e.g., Babuška et al., 1990, Lippitsch 2003; Zhao et al., 2016)." This is a misunderstanding/misreading, because Paffrath et al. don't claim that there was a reversal in slab polarity below the Central Alps! Please read carefully. Rather, they refer to the part of the slab beneath the Eastern Alps when they write „The new model does not require a slab polarity switch due to the detachment from the crust and the nearly vertical dip of the eastern Alpine slab. Tectonic arguments (in Handy et al., in rev. with Solid Earth) rather suggest a European provenance of the slab".

*Reply:* We have read carefully both the submitted paper by Paffrath et al. as well as studied the presentation at EGU2020 (reference given), where on slide 15 the authors compare their results with results of Lippitsch et al. (2003) along slice B running through the C. Alps and show:

Discrepancy in lithospheric slab orientation, slab connected to Adria?
Dipping direction seems to be reversed (NW) in comparison, however structure of this slab is highly complex with eastern part showing tendency to dip SE

Our sentence "In recent studies, Paffrath et al. (2020) suggest the reversed slab polarity relative to the Western Alps already in the Central Alps, as opposed to the formerly documented polarity reversal further to the east - beneath the E. Alps (e.g., Babuška et al., 1990, Lippitsch 2003; Zhao et al., 2016)" refers correctly to this presentation. The reference to Zhao et al., 2016 might be deleted as they do not have sufficient resolution beneath the E.Alps. We can also delete the entire sentence in case statements in slide 15 of the presentation are no more valid, as it is not essential for our ms., in which we focus on the E. Alps and the BM. At this stage of our ms. we do not say anything about the slab provenance.

*Lines 224-226:* "Mock et al. (2020) pointed out the discordance between the slab geometry at depth and the boundary between the Eastern and Central Alps observed in the surface geology and, similarly to Rosenberg et al. (2018), shift the boundary between the E. and C. Alps further to the east, at the Giudicare-Brenner fault system." The authors cited do not redefine the boundary between Eastern and Central Alps, they merely note a discordance between the slab geometry at depth and the boundary between the Eastern and Central Alps observed at the surface. The boundary between the E. and C. Alps is defined at the surface in map view by the western edge of the continuous Austroalpine nappe stack located in Eastern Switzerland. This traditional definition does not take into account the effect of erosion which removed Austroalpine nappes originally exposed somewhat further to the west. The former existance of these nappes to the west is supported by the provenance of components in the Molasse Basin and by the Austroalpine Klippe in the Central Alps. Regarding the Alpine slab, the boundary between the central and eastern slab segments (as defined either by a slab gap or a change in dip direction of the slab) coincides generally with the Giudicarie-Brenner faults. In light of recent evidence for a laterally continuous Alpine slab with a change in dip (Handy et al., in review with SE), it would be clearer to avoid relating surface and subsurface boundaries between Central and Eastern Alps.

Reply: Wee see your point and chose to keep a shortened delimitation of the C./E. Alps boundary, regardless of the interpretation. The rest of the paragraph has been deleted:

Mock et al. (2020) note a discordance between the slab geometry at depth and the boundary between the Eastern and Central Alps observed at the surface. The boundary between the central and eastern   slab segments of the Alps, as defined either by slab gaps, due to delamination (Handy et al.,  2021)  or a change in dip direction of the subductions, coincides with the Giudicare-Brenner fault system, in general (e.g,  Rosenberg et al.,  2018).

*Lines 238-241:* "Later tomography of the upper mantle which included the E. Alps from data of regional passive experiments (Dando et al., 2011; Mitterbauer et al., 2011; Karousová et al., 2013) also retrieved the northward dipping high-velocity heterogeneity of similar geometries (Fig. 5) and associated it mostly with the Adria plate subduction." It is important to avoid equating a switch in the dip direction (polarity) of the Alpine slab with a gap in this slab. In the interpretation of Handy et al. (in rev. with Solid Earth), the slab changes its dip direction along strike without evidence for lateral separation into two segments. Rather, the along-strike change in dip direction of the slab coincides generally with an eastward transition in its degree of detachment from the orogenic lithosphere.

**Reply:** We modified the sentence, in which we delete the last part. Reference to Dando et al. (2011) for the slab relation to the Adriatic plate was included by a mistake.

Later tomography of the upper mantle, which included the E. Alps, from data of regional passive experiments (Dando et al., 2011; Mitterbauer et al., 2011; Karousová et al., 2013) also retrieved the northward dipping high-velocity heterogeneity of similar geometries  (Fig. 5) in the 250 km of the upper mantle.

To the comment segmentation vs. change in dip direction:

Along-strike change in dip direction of the slab (without a gap) is conditioned by the slab detachment to the east.  In case of no or "thin" slab detachment (irresolvable at the current 30x30x30 km grid), as imaged in several tomographies including ours, the change in subductions calls for slab segmentation, or a "vertical" gap (separation) between the two subductions (regardless how they are interpreted). When interpreting results of body-wave tomography, we strongly follow principles of P- velocity perturbation images, i.e., deviations from a 1D reference velocity model. The damped deviations referes to each layer separately and do not allow direct comparison with absolute velocities in other models. One has to be careful with interprations of vertical cross-sections as well.

**Lines 241-246:** "….suggested three main phases in building the E. Alps keel: (1) NW translation of the Adria and its thrusting over the (subducting?) European plate in the W. Alps, (2) fragmentation of northern Adria along a deep-seated fault (possibly the Giudicarie Fault, or at least a spatially nearby structure) and (3) counter-clockwise rotation of the Adria and its subduction beneath the European plate in the Eastern Alps, with a triple-junction of three crustal terranes in its eastern rim proposed by Brückl et al. (2010), although the deformation style between the E. Alps and the Pannonian Basin is usually considered diffuse on the surface." Two points need clarification here: (1) Based on recent P-wave tomography from the newest Alp Array data (Paffrath et al. in rev), the length of slab beneath the Eastern Alps, irrespective of its dip direction, far exceeds the amount of shortening in the eastern Southern Alps. This precludes a primarily Adriatic origin of this slab, as discussed in Handy et al. in rev, Solid Earth; (2) there is no unequivocal evidence at the surface or at depth to support the idea of three "crustal terrains" with distinct Moho boundaries near the transition from

E. Alps to the Pannonian Basin. Rather, the Pannonian Basin is underlain by thinned European and Adriatic crust and lithosphere that acquired their reduced thicknesses in Miocene time (e.g., Ustaszewski et al. 2008). The decrease in Moho depth going from the Eastern Alps to the Pannonian Basin is gradual, not abrupt (e.g., Horvath et al. 2015), indicating extensional flow of the lower crust and upper mantle.

**Reply:** ad comment (1): we refer to our interpretation from the '90s, reflecting the LAB mapping and results of P-wave tomography, both showing differences between the Western and Eastern Alps. NW movement of Adria, its counterclockwise rotation and collision with the European plate is not questioned (e.g., Le Breton et al., 2017), including the amount of shortening (Ustaszewski et al., 2008). We modify the text as follows:

The imaged triangular shape of the LAB model (e.g., Babuška and Plomerová 1992) beneath the thickened lithosphere of the E. Alps detached from that beneath the W. and C. Alps led Babuška et al. (1990) to suggest three main phases in building the E. Alps mantle-lithosphere root: (1) NW translation of the Adria and its thrusting over the (subducting?) European plate in the W. Alps, (2) fragmentation of northern Adria along a deep-seated fault (possibly the Giudicarie Fault, or at least a spatially nearby structure) and (3) counter-clockwise rotation of the Adria resulting in a start of its subduction after a collision with the European plate in the Eastern Alps (Ustaszewski et al., 2008), potentially above a delaminated European lithosphere residing at greater depth (Paffrath et al., 2021; Handy et al., 2021).

Ad (2) triple junction (Brückl)

R**eply:** In tectonically complex regions, as the Alps and surrounding area, one can hardly expect that only one fixed opinion exists. We modified the end of the sentence related to the crust and complemented the references as suggested.

Complexity of this region is reflected in ambiguous views on the crust, being interpreted as a triple-junction of three crustal terranes (Brückl et al., 2010), although the deformation style between the E. Alps and the Pannonian Basin is usually considered diffuse on the surface. Instead, e.g., Ustaszewski et al.(2008) and Horvath et al. (2015) assume the Pannonian Basin underlain by gradually thinned European and Adriatic crust, which has been recently imaged by Kalmár et al. (2021).

**Lines 267-277:** "The north oriented dip of the EA subduction, imaged in early tomography studies…..Most of them agree in interpreting it being of Adriatic plate origin (Dando et al., 2011; Karousová et al., 2013). However, Mitterbauer et al. (2011), and similarly in recent tomography by Paffrath et al., (2021 this issue), the positive perturbations are associated with a delaminated EU slab. Kästle et al. (2020) relate the HV-EA mainly with the European plate subductions as well, and leave no or only minor role to the Adriatic subduction. The authors explain the northward subduction modelled beneath the E. Alps by imaging a combination of the short Adriatic and deep delaminated, potentially overturned European slabs." Here, the authors should mention that recent P-wave tomography based on the newest Alp Array data (Paffrath et al. in rev) supports a primarily European origin of the north-dipping slab beneath the Eastern Alps (see discussion in Handy et al., in rev. with Solid Earth). This conclusion is based on the length of imaged slab beneath the Eastern Alps (several hundred km) which far exceeds the amount of shortening (c. 50 km) in the overlying crust of the eastern Southern Alps. In this interpretation, the steep northward dip of the slab segment beneath the E. Alps is attributed to asthenospheric flow during or after Adriatic indentation (Handy et al., rev.).

**Reply:** The requested formulation is included in the original submission (see above). We extend the sentence and add the Handy et al., 2021 this issue refence as well).

… and similarly in recent tomography  based on the newest Alp Array data by Paffrath et al., (2021 this issue), the positive perturbations below 150km are associated with a delaminated European slab (see also Handy et al., 2021 this issue).

**Lines 401-402:** "The northward-dipping lithosphere keel is imaged down to ~200-250 km, without signs of delamination, and we associate it with the Adriatic plate subduction." As above, it is unclear what the authors mean by "keel". It appears to mean slab, but this should be clarified.

**Reply:** The sentence was modified (keel deleted):

The northward-dipping slab is imaged down to ~200-250 km, without signs of delamination from the Adriatic plate. Therefore, associations of the high-velocity heterogeneity with the Adriatic plate subduction seem to be obvious.

**Interpretations of anomalies beneath the Bohemian Massif**

**Lines 281-285:** "To understand the positive perturbations beneath the southern BM, we compare it with results from the large-scale Paffrath et al. (20201) tomography and with the regional tomography of the BM (Karousová et al., 2013) of similar resolution to ours (Fig. 6). The strongest positive perturbations related to that heterogeneity overlap in the models, though they are of unrealistically large extent in the Paffrath's et al. model (2021, this issue)." Why is this extent in the Paffrath et al. model unrealistic? "There, they include also the ER with the thinnest BM lithosphere, well imaged by negative perturbations in the EASI-AA and in BOHEMA models (Karousová et al., 2013)."

**Reply:** We modify the sentence to make it clearer and added reference Plomerova et al. (2016).

Paffrath et al. (2021) show the strong positive perturbations also beneath the north-western part of the BM (beneath the Eger Rift), where the lithosphere thins significantly (~80 km), which is well imaged by negative perturbations in the EASI-AA and in BOHEMA models (Karousová et al., 2013; Plomerova et al., 2016).

**Lines 308-311:** "The role of applying proper crustal corrections is significant in teleseismic regional tomography. Not applying any crustal corrections or applying inadequate ones can strongly affect velocity perturbations within the upper ~100 km of the upper mantle (e.g., Karousová et al., 2013), which is the zone, where the models discussed above differ. From this point of view, developing a uniform detailed and reliable model of the European crust is urgently needed." I agree and find the comparison of models in this chapter valuable. I've sent the authors some PPT slides of a recent talk showing the correspondence between models regarding the degree of connectivity of slabs with their orogenic lithosphere. Both the recent P-wave tomography employing 3D crustal models (Paffrath et al. in rev.) and the surface wave models of Kaestle et al. (2018, 2020) indicate slab detachment in the Western and Eastern Alps, with local slab attachment only in the western Central Alps. This coincidence of images with different methods suggests that the crustal correction of the Pwave model of Paffrath et al. is robust.

**Reply:** We extended the paragraph by a sentence as follows

"Overcorrecting" the travel times due to a crustal model used, regardless of a method of correcting itself, can delete/erase or substantially reduce positive perturbations in the upper 100 km, if too slow and thick crust would be considered and vice versa.

**Lines 332-333:** "The simplest explanation would be to consider it as a fragment of the delaminated part of the European plate subductions, as suggested in Handy et al. (2015) (Fig.7a )." This appears to be a misunderstanding: The +Vp anomaly below the BM is not the anomaly referred to in Handy et al. 2015 (their Fig. 11a), which was a European slab hanging directly below the Eastern Alps in Miocene time. Note that Handy et al. 2015 never considered a +Vp anomaly below the BM in their model. Instead, the Miocene +Vp anomaly below the Eastern Alps in their figure is torn European lithosphere that sank and no longer exists. The model of Handy et al. 2015 is actually consistent with the statement in Line 338 that "Therefore, an association of the HV-BM with the delaminated fragment of the EU subduction is not likely."

**Reply:** Our formulation was not clear enough, if it lead to the idea described by the referee. We did not mean that at all. We reformulate the sentence.

The simplest explanation would be to consider it as a small fragment of the delaminated part of the European plate subductions, as suggested in Handy et al. (2015) for the heterogeneity beneath the Eastern Alps at greater depth (Fig. 7a, now Fig. 8a ).

**Points pertaining to rock physics:**

**Lines 347-348:** "The Phanerozoic continental mantle lithosphere, composed of originally lighter rocks than those in the asthenosphere (Mantle lithosphere is generally denser rather than lighter than asthenospheric mantle) and becomes denser due to metamorphic phase changes as it subducts" (reference?)

**Reply:** The mantle lithosphere alone is on average a bit (ca. 70 kg/m3) denser then the asthenosphere. The mantle lithosphere remains above the asthenosphere as it is mechanically coupled to the crust, so the overall lithosphere density is less than the asthenosphere density. However, as soon as the upper or the full crust is decoupled from the mantle lithosphere, the lower lithosphere (denser than the asthenosphere) is able to sink. The ms. text is updated accordingly:

Updated text:

The Phanerozoic continental mantle lithosphere may mechanically decouple from the upper or the full crust, and since this lower lithosphere is denser than the asthenosphere it subducts. Densification of the lower crust through metamorphic reactions may enhance this process if the convergence is fast (e.g., 2 cm/yr in Hetényi et al., 2011).

New reference: Hetényi G, Godard V, Cattin R, Connolly JAD (2011) Incorporating metamorphism in geodynamic models: the mass conservation problem. *Geophys J Int* **186**:6-10. doi:10.1111/j.1365-246X.2011.05052.x

**Lines 360-363:** "The roll-back subduction of the Carpathians, accompanied by a substantial asthenospheric flow, could open a space between the E. Alpine and Carpathian slabs for the northwestward "transportation" of a purely oceanic lithosphere or a mix of oceanic and continental

lithosphere fragment into the mantle beneath the BM." No reason or mechanism is provided here to explain why oceanic or mixed ocean-continent lithosphere should be "transported" northward. Please elaborate.

**Reply:** there is no current data to demonstrate this, like unequivocal anisotropy due to mantle flow, as such signals have been overprinted. Nevertheless, our proposal here is based on Kissling and Schlunegger (2018)'s model of the Alps, accompanied by complex asthenosphere flow in context of slab edges (e.g., Vignaroli et al., 2008, their Figure 8 for the Western Alps. (https://doi.org/10.1016/j.tecto.2007.12.012) and other examples. For the context discussed in our manuscript, such flows could have happened around the retreating and /or delaminated slabs. The manuscript text now explicitely says this.

Manuscript change, the new sentences read as follows:
The Carpathians front curved significantly and migrated to the north. Differences in the roll-back subductions of the Alps (e.g., Kissling and Schlunegger, 2018 and reference therein) and the Carpathians, northward push of Adria and European slab delamination beneath the EA could have formed complex flows in the asthenosphere (e.g., Vignaroli et al., 2008), which could "transport" a purely oceanic lithosphere or a mix of oceanic and continental lithosphere fragments through the open space between the E. Alpine and Carpathian slabs north-northeastward into the mantle beneath the BM. The rotational displacement of the Adriatic Plate indenter provided an additional driving force for modifications of the Alpine-Carpathian-Dinaridic orogenic system.

New references:
Ustaszewski et al., 2008
Kissling and Schlunegger, 2018.
Vignaroli G, Faccenna C, Jolivet L, Piromalo C, Rossetti F (2008) Subduction polarity reversal at the junction between the Western Alps and the Northern Apennines, Italy. *Tectonophys* **450**:34-50. doi:10.1016/j.tecto.2007.12.012

**Lines 380 - 396:** "The roll-back subduction of the Carpathians, accompanied by a substantial asthenospheric flow, could open a space between the E. Alpine and Carpathian slabs for the northwestward "transportation" of a purely oceanic lithosphere or a mix of oceanic and continental lithosphere fragment into the mantle beneath the BM." No reason or mechanism is provided here to explain why oceanic or mixed ocean-continent lithosphere should be "transported" northward. Please elaborate.

**Reply:** The continental mantle lithosphere is formed by domains with their own large-scale fabric, oriented generally in 3D, i.e., with inclined symmetry axes of anisotropy ( e.g., Babuska and Plomerova, 2020). For example, In case of two adjacent lithosphere domains with fabrics, in which the dipping high-velocity directions face each other (i.e., they are oriented convergently), an artificial positive heterogeneity could be produce at greater depth in isotropic images of the upper mantle.

We added an explanation after the quoted sentence
"Ignoring seismic anisotropy and assuming isotropic wave propagation or considering only azimuthal and/or radial anisotropy leads to significant isotropic and anisotropic imaging artefacts that may lead to spurious interpretations (VanderBeek and Faccenda, 2021)."

In this study of the broader region around the E. Alps we have applied, in the first step, the isotropic mode of a coupled anisotropic–isotropic teleseismic P-wave tomography developed by Munzarová

et al. (2018).  In spite of the general good agreement with the high-resolution large-scale isotropic tomography (Paffrath et al., 2021, this issue), the images can be biased due to seismic anisotropy (Eken et al., 2012; Qorbani et al., 2015, 2016; Bokelmann et al., 2021).  Laterally varying anisotropy, which correlates with tectonics of the region, has been indicated in shear-wave splitting (e.g. Link and Rumpker, 2021).  After collecting sufficient amount and well-distributed high-quality data, we will run the coupled anisotropic-isotropic mode of the code and retrieve 3D anisotropic model of the region, which will allow for laterally and vertically varied anisotropy with axes oriented generally (inclined) in 3D. This further investigation may help in deciding among the drafted scenarios for the origin of the HV-BM, or point to new ones.

New reference: Link and Rumpker, 2021, feart, doi: 10.3389/feart.2021.679887

**Figures and Captions:**

**Fig. 1** (right side): Label the main faults and use transparent colour to mark the Alps between the Northern and Southern Alpine Fronts.

**Reply:** We add labels to main faults, but do not mask the Alps with any schematized colour symplification.

**Fig. 2** (right panel): Label the Southern Alpine Front (SAF), use transparent colour to mark the Alps between the Northern and Southern Alpine Fronts. It would help to mark the outlines of the Tauern Window (TW). I would eliminate the triple junction, because there is no structural evidence for a Pannonian Plate sensu Brückl et al. (see comments above).

**Reply:** The same as for Fig.1. Labels added, the Tauern window contoured. The triple junction deleted.

**Fig. 3**: Label the Southern Alpine Front (SAF) on all cross sections. In the caption, indicate what DF means (this is not included in any of the earlier captions). Indicate the traces of all cross sections on a map or inset map. Include coordinates of the endpoints in the caption or directly on the figures. This is important for others who wish to compare their results with the authors'.

**Reply:**  SAF Label added at all cross-sections, DF included among explanations in Figs.1 and 2, traces of all cross-section marked in the horizontal slice at 120km (new upper right inset). Geographic coordinates of endpoints of all the along longitudes crossections are complemented in the captions.

**Fig. 4**: Label all the faults shown in the other cross sections. Include coordinates of the endpoints in caption or directly on the figures. Again, this is important for others who wish to compare their results with the authors'.

**Reply:** Labels complemented, coordinates added in the caption.

**Fig. 5**: The labelling of the cross sections should be made clearer. Label each section only once so it corresponds to the letters of the traces on the map. The orogen-parallel cross section in a) should have the longitudinal degrees correspond to the longitudinal units on the map on the lower right.

**Reply:** Required change from landscape to portait destroyed the figures, which escaped my attention. Figure is redrawn with correct labelling. For better visibility of the heterogeneities we keep part (a) with a broader extent of longitudes.

**Fig. 6**: Add Variscan unit boundaries on the main horizontal slice of Paffrath to enable better comparison with the other slices shown. Distinguish Alpine and Variscan boundaries, either by using lines with different thicknesses (Alps thick, Variscan thin) or colours.

**Reply:** We prefer keeping the Paffraths et al. plots at as close as possible to their original form, the Alpine and Variscan boundaries are distinguished in the two other parts of our figure. We label all important faults and units.

**Fig. 7**: Check the position of the cross sections with respect to the seismic anomalies, because they look misaligned. The northern tip of the delaminated EU slab in a) is too far to the north in b) with respect to the Northern Alpine Front (NAF). Handy et al. 2015 proposed that this slab sank beneath the core of the Alps, not that it migrated to the north with respect to the front. However, if this is your idea (meaning the authors of this text), then please state this clearly.

**Reply:** This is probably a misunderstanding. We slightly corrected the position of the delaminated slab in the schematic cartoon plotted across backround from Handy et al. (2015) and complemented the caption.

**Fig. 8**: Show the end points of the cross sections on an inset map so that the location of the. anomalies and unit boundaries can be compared, otherwise the model can't be compared with other data and the reader is helplessly lost.

 In b), please also consider other interpretations of Late
Devonian to late Carboniferous paleogeography and subduction directions of the Rheic Ocean, e.g., Schulmann et al. 2009, 2014, Ballevre et al. (2009), Zeh & Will (2011) or in Franke et al. (2000). What effect do these scenarios have on your interpretation of the 3% positive Vp anomaly beneath the Moldanubikum?

**Reply:** Figure 7b is part of the schematic cartoon (not to scale) with reference on Babuska and Plomerova (2013), where details of the cross-sections are presented and other interpretations as requested are discusseed and referred in Section 6. We complement the captions.

We present some of definitely more potential scenarios, but at present stage, we do not have enough information to favour only one of them, neither dispute with other interpretation of the Reic Ocean subductions. Arguments for our model of BM we have presented, e.g., in Babuska and Plomerova (2013) and reference it in this ms.

---

## Author Comment (AC5)

Dear Reviewer 2,

Thank you for your comment on our manuscript. We report all your comments in italic below (preceded by "RC2"), followed by our respective answers. The main manuscript is revised accordingly, some of the modified sentences are include below (underlined text).

Reply to RC2

**Introduction:**

RC2: The introduction is surprisingly short. I miss a proper description of the state of the art (what is known), of controversial issues (what is debated) and of how the manuscript contributes to answering some of these open questions. The introduction does not pose any research questions and thus entirely lacks a motivation for the work presented. I also miss a proper description of previous work done with EASI data and its relevance for the current work. At the end, a brief description of the contents of the paper would be welcome. Instead, the last paragraph rather resembles an abstract.

**Reply:** We added sentences on actual stage of debates on the E. Alps slab (scientific questions), our motivation for the work and reformulate description of the content of the paper as requested. With that, we consider our description of the classical view and state-of-art of the tomography images of the Alps in the first two paragraphs of the Introduction as representative and sufficient.

While resolution of early tomography of the Alpine region allowed researchers to deal only with the most distinct heterogeneities in the upper mantle, accumulation of high-quality data from the dense AlpArray network motivated us to search finer images of the upper mantle and answers on segmentation of the Alps, dip directions of the subductions, their relevance to European or Adriatic plate, extent of slab delamination and particularly, to elucidate the smaller size heterogeneity beneath the Bohemian Massif north of the E. Alps subduction.

*RC2:* The meaning of the lines in the tectonic map in Fig 1 needs to be explained in the caption.

Reply: Captions complemented.

**Data:**

*RC2:* I assume that also data of the complementary EASI experiment are used. This should be made clear in the text.

**Reply:** Usage of the AlpArray-EASI data is presented in the 1st sentence of the abstract and in the 3rd par. of the Introduction of the original submission, as well as in the Section Data. Nevertheless we modified the original lines 69-72 of Section 2. Data. It sounds in the revised version as follows:

We collected recordings from stations of the AlpArray Seismic Network (AASN, doi.org/10.12686 /alparray/z3\_2015) and AlpArray-EASI network (doi.org/10.12686 /alparray/xt\_2014). All the AASN

stations installed, in a 200km-wide band (Fig. 1) along the densely spaced stations of the AlpArray-EASI complementary experiment (Hetényi et al., 2018b), were selected for this study.

*RC2:* Which procedures for checking data quality have been applied? The paper should be self-contained to some extent.

**Reply:** We checked data for its completeness and correct timing (uncorrected leap second, failures in clock synchronizations) and station metadata to fix several mistakes (e.g., wrong channel name, station mislocation, poles and zeros given in Hz instead of radians/s).

We have applied several procedures to check the data quality (Vecsey et al., 2017), particularly data completeness and correct timing, to eliminate periods with the uncorrected leap seconds or failures of lock synchronizations. In station metadata files we fixed for example wrong channel naming, station mislocation, and incorrect units for poles and zeros (mixing Hz and radians/s units).

*RC2: The description of the picking scheme and the event distributions should be put into main paper.*

**Reply:** We consider this part as valuable for deep specialists and we prefer keeping it in the supplementary materials, which are easily accessible, for those who are interested. We do not want to extend further the main text due to an earlier experience with Solid Earth about manuscript length.

RC2: Which trace is used as reference for cross-correlation and beam forming?

Reply: explanation added

A low-noise beam trace created from stacked cross-correlated and shifted traces of an event serves as a reference in the second cross-correlation step and beam forming in the P-arrival time picking. This means, there is no subjective a priori selected single reference trace.

RC2: Quantitative statements regarding error estimation of the picks and their probabilistic combinations should be made.

**Reply:** We modified the text, added a new part to Fig.S2 (part S2b, Histograms of uncertainties, means, medians) and extended caption of Figure S2:

The final time of each extreme (green P1, P2 in Fig. S2a) and its error estimate is computed from the normal distribution which approximates a mixture of normal distributions of partial picks.

Figure S2b shows uncertainties of the measured P-wave arrivals, means and medians for both the complete dataset as well as for events selected for tomography (see below).

Each of the red, black and blue picks is complemented by its error estimate defined as a standard error of the normal distribution. Time error of the red extreme depends on a signal noise level (see cyan basins, their height is given by a noise magnitude - red dashed lines), errors of the black and blue correlation picks come from coherence of the signal with the beam. The final time of each extreme (green P1, P2) and its error estimate is computed from the normal distribution which approximates a mixture of normal distributions of partial picks (red, black and blue P1, P2), .....

RC2: The description of the enhanced data set is confusing. Is the enhanced dataset just the original one with 201 events (in Figure S1 209 events) plus another 43 events within the 60 degree back-azimuth cone? Or is it a completely new selection from the overall available events containing only events within the 60 degree cone? This should be clarified in the manuscript. Regarding the figures S1, **the latter seems to be the case**.

**Reply:** We stated on lines 86-88 of the original submission "To enhance the resolution in direction of the subducted plates, we selected further rays coming from the northern and southern 60° wide azimuth bins to be included in the tomographic inversion."

Of course, the reason to take only rays from the northern and southern 60° wide azimuth bins was to minimize effects of heterogeneities east and west outside the elongated model. Thus you are right: "the latter is the case". Besides showing the event distributions in Fig. S1, we also include Fig. S5 (original submission, and text there on L90-91), which shows the enhanced resolution along the vertical N-S cross-sections, if all rays are included (left, rays as in FigS1a) and if rays from sides of the model are excluded (right, rays as in Fig. S1b). This means, the original submission has already contained the above recommendation of the RC2. We modify the text and caption of Fig. S5 to prevent misunderstanding:

To eliminate mapping effects of heterogeneities aside the model into its internal part and to enhance the resolution in direction of the subductions, we selected additional rays coming from the northern and southern 60° wide azimuth bins. Only rays propagating through the model within the two azimuthal fans (see Fig. S2) have been included in the final tomographic inversions, tested, discussed and interpreted further. This data comes from 244 earthquakes, each of them being recorded by 120 stations on average, i.e., by 50% of stations in the region.

**Figure S5**: Velocity perturbations along Profile EASI through models calculated for rays from all directions (left) and for rays from the northern and southern 60° wide azimuth bins (right, see also Fig. S1). Relatively less-well resolved regions along the profile are shaded.

**RC2: There are two figures S1 in the supplement. Renumber to S1 and S2.**

**Reply:** Figure S1 has parts a) and b). The second label was missing in the second part of the figure of the original submission, now complemented. Caption and numbering of following figures remain correct.

RC2: Information on crustal structure is taken from several sources. How is the 3D crustal model put together? How are transitions between subregions treated? How are crustal discontinuities treated, as transition regions or as real discontinuities? At least two horizontal sections through the 3D crustal velocity model should be shown. To which depth does the crustal model reach? Does it also contain parts of the uppermost mantle?

How is the correction for crustal structure done? One correction for all events assuming vertical incidence or event-wise corrections taking correct incidence angles into account?

**Reply:** we compile information for each station and its surrounding, there are no transitions between subregions, individual crust parameters go to Moho depth, i.e., it does not contain the uppermost mantle. The absolute residuals relative to the IASP91 model are corrected for the real crust deviations from the model. For each ray, refractions in the crust according to incidence angle and azimuth are considered. We never applied one correction for all events assuming vertical incidence, but event-wise corrections, which take correct incidence angles into account.

RC2: The meaning of the end of the following sentence (line 100) ".... proper tomographic inversion in our target region to resolve structures, including the spatial limits of our images" is unclear? What are the spatial limits of the images? Do the authors want to say that resolution is also good at the model boundaries?

Reply: Of course, we did not want to say anything like that. We simplify the sentence.

With this approach we gathered a high-quality and uniform dataset of travel time residuals for a proper tomographic inversion to resolve structures in the target region.

RC2: Estimated uncertainties of the travel time residuals should be documented in the manuscript.Show a histogram of the uncertainty distribution and give average and median values.Reply: As above, new Figure S2b with histograms of uncertainties, mean and medianhas been included in the revised version.
**RC2**: It is stated that the matrix W\_m in equation (4) provides horizontal smoothing. What about vertical smoothing?

**REPLY:** There is only horizontal smoothing. There is no option for vertical smoothing in the original code for teleseismic tomography that we use. Both the Telinv and the AniTomo codes have been created without such a possibility in order not to enhance sub-vertical smearing, which is given by the ray geometry.

**RC2:** Is there also a damping term included in W\_m, and if yes, what is the weight of smoothing relative to damping?

**Reply:** Smoothing matrix forces the model parameters at each inverted node to be close to an average of the model parameters in the surrounding inverted nodes. Such a form of the smoothing matrix is fixed in AniTomo and in Telinv and there is no option to change any weight there. Damping factor epsilon\*\*2 is just a constant multiplying every element of the smoothing matrix.

**RC2:** With 13 cells in vertical direction, I calculate a model depth of 13\*30=390 km and not 435 km as stated in the text. Is there a specific reason for choosing 390 km (or 435 km) as the bottom of the model. Is this depth still warranted by intersection of rays given the length of the EASI profile?

**REPLY:** The number of 13 refers only to layers of grid nodes, in which the inversion for velocity perturbations is allowed, i.e., at the 60 km, 90 km, ..., 390 km and 420 km. The depth of 435 km corresponds to the bottom of the grid cells that belong to the nodes at 420 km depth. We clarified this in the text. There are still a lot of rays at the depth of 420 km, crossing mainly in the central part of the array. In the northern and the southern parts of the array, the ray crossing is less good compared to the upper parts, of course. See also the new Fig. S7 showing the whole model from the checkerboard test.

The area of about 400 000 km2, centred at 13.3°E 48.5°N, is approximated by 30-by-30 km cell size, horizontally, and with 30 km spacing, vertically. The images are calculated down to 435 km depth on a vertical grid of 30 km spacing. To minimize creating false perturbations, we invert for the velocity perturbations only in the central 5 x 25 x 13 cells, which are well-sampled by criss-crossing rays (Fig. S1), i.e., in nodes between depths of 60 km and 420 km. The model covers the Eastern Alps and a core of the BM, an area of ca. 140 400 km2 in total. Data variance reduction of the final model for the chosen damping parameter attains 66% (Fig. S4).

RC2: What is the damping factor in Fig. S4? Is it epsilon\*\*2 from equation (4)?

**Reply:** Yes, damping factor is epsilon \*\*2 from equation (4).

Why do the authors plot data variance instead of misfit which is normalized to the data uncertainties. What is model variance? Is it the squared norm of the velocity perturbations? A definition of this quantity should be provided. The authors should provide a value of misfit normalized to the picking uncertainties to get an impression whether there is overfitting or maybe even severe underfitting.

**Reply:** Evaluation of data variance in tomographic codes AniTomo and Telinv includes normalization to the data uncertainties. We modified caption of Fig. S4. Reduction of data variance of 66 % has been mentioned at the end of Section 3.

**Figure S4:** Data and model variance trade-off curve evaluated for various values of damping of the isotropic-velocity perturbations and numbers of iterations. The data variance and model variance are squared norms of the time residuals and velocity perturbations, respectively. The data uncertainties are included in the evaluation of the data variance.

RC2: Regarding Fig. S4. I wonder that the authors get increasing data variance and decreasing model variance for decreasing damping factor. I would expect exactly the opposite. This strange behaviour needs to be explained by the authors.

**Reply:** The strange behaviour of the data and model variances in Fig. S4 is caused by a mistake in manual (re-)colouring of the points of the trade-off curve. The model variance should really decrease

with an increasing damping factor. Thank you for noticing that. We corrected the plotting mistake in Fig. S4.

*RC2:* Why do the authors only consider 2 iterations? Did they try more iterations and how does the inversion behave then?

We performed one more iteration, but overall the imaged perturbations remain similar, thus we decided to stop calculations after the second iteration to save the time.

**Results:**

RC2: In the "Data" section two dataset were presented. Which one was used to create the model shown in the figures? Fig. S5 which compares vertical sections obtained from the 2 data sets is only mentioned once. The issue is never discussed in the results or the interpretative section.

**Reply:** The main features of the perturbations are the same in both models (Fig. S5), but models containing rays from the west and east could be potentially biased from heterogeneities outside the model. Reasoning for limiting rays into the northern and southern fans is expressed in section Data. All results presented, tested, discussed or interpreted are based on this data set. Modification of the text included in Section Data.

RC2: Regarding the vertical sections in fig. 3, I recognize significant perturbations in the gray-shaded upper 50 km of the model which appears to be the crustal domain. As the effect of the crust was subtracted from the travel-time residuals, why do the authors still allow perturbations in the upper 50 km? Theoretically, after crustal correction, the travel-time residuals should represent pure mantle structure and perturbations in the crust should be suppressed. How would the model change if perturbations in the upper 50 km were forced to zero? In particular, what would happen with the model between 50 km and 100 km depth, just beneath the crustal domain? The interpretation of a northward dip strongly depends on the velocity perturbations between 50 km and 100 km depth. Below the high velocity anomalies are rather vertical. Whether HV-EA is delaminated or not, also depends on the anomalies in this depth range. All the conclusions following in the paper about polarity flip, northward dip and detachment of the Eastern Alpine slab depend on this issue.

**Reply:** Relative residuals corrected for the crustal deviations (including sediments, Moho depth, velocity) relative to the IASP91 represent our tomography input. For each rays, refractions in the crust according to incidence angle and azimuth are considered. First, absolute residuals relative to the IASP91 are corrected for the real crust deviations from the models, then the array-average residual, calculated from the crust corrected residuals of an event, is subtracted from the residuals at each station which recorded the event. The significant perturbations in cross-sections come from our plotting error, which we correct now in all the figures. Because the residuals are corrected for the deviations in the crust, they are assumed to represent pure mantle structure, under the condition that the crustal model is correct. Insufficient corrections map perturbations into the upper 100km of the mantle. On the other hand, an overcorrection can erase or substantially reduce positive perturbations in the upper 100 km (see the text modification in the last part of Section 6). We invert at node depths of 60-420 km (13 grid levels). We do not invert at 30km, neither at the deepest 450 a 480 km. Perturbations related to grid nodes at 60 km correspond to depth range of 45-75km. Therefore, our new images are plotted from depth of 45 km. We do not invert for the crust. The two deepest layers of nodes together with the nodes at shallower depths, where we also do not invert for velocity perturbations, surround the volume studied to stabilize the tomography.

RC2: For a connection to crustal levels, one could plot the perturbations of the crustal model relative to a 1D-model of the crust into the vertical cross sections. But then, what to do with the anomalies that are already there?

**Reply:** The original submission contains Fig. S3 with information on the crust. We added the Moho depth considered in the crustal corrections into cross-section in Fig. 3. There are no anomalies in the crust, as explained at several places and in our answer to the previous question, we corrected the plotting error.

**RC2:* What is the criterion for gray-shading in the model domain? Is it derived from the resolution matrix?**

**Reply:** Boundary of the gray-shaded area follows a smoothed contour of diagonal elements of resolution matrix (RDE) equal to 0.15. The contour approximately separate the well and less well-resolved regions in the synthetic tests. This has been specified in the caption.

**Resolution Tests**

RC2: I would also like to see the results from a checkerboard test (with gaps) which nicely shows lateral and vertical smearing. In particular, vertical smearing at shallow levels should be investigated because it may hide a detachment of the slab or falsely connect the high vp anomaly below 100 km with a shallower one further to the south giving the impression of a significant northward dip.

**Reply:** We are aware of both the advantages and the disadvantages of the checkerboard test. We complement the Section 5 with the checkerboard test and include in the supplements a new Figure, S7a,b, as requested.

Besides the specific tests described above we also performed standard checkerboard tests to assess resolution capability of the network (Fig. S7- new). The checkerboard test confirms the positive and negative perturbations are retrieved well down at least to 240 km with a weak vertical smearing (Fig. S7a – horizontal slices). Also the vertical cross-section through the central part of the model (Fig.S7b) images the synthetic perturbations reliably.

RC2: The remark that the polarity flip is more or less accepted is certainly a misconception. If I read Paffrath et al. (2021) correctly, they see a rather vertical, detached eastern Alpine slab and favour the interpretation of European provenance because its down-dip length can only be explained by the Tertiary shortening in the Eastern Alps accommodated by south-dipping subduction of European lithosphere. Basically, what is seen in the tomography is only the slab dip but tomography does not tell us the provenance of the slab. Even if it were dipping clearly northwards (in the tomographies by Mitterbauer (P4), Zhao and Paffrath, it is nearly vertical) it could still be overturned European lithosphere. Independent data are needed to decide this issue. The use of the term "polarity flip", however, already implies the interpretation of Adriatic provenance of the slab.

**Reply:** Tomography images as referenced in our original ms. show steep northward-dipping highvelocity heterogeneity beneath the E. Alps within the upper mantle, down to ~250-300km. We interpret the images, especially in section of synthetic tests, not any concept, and we test the resolution of tomography images we get from available rays. In Section 6, we compare different tomography images, showing the northward dip, though they are interpreted differently. We agree that additional information are needed to decide the issue. We modify our formulations from polarity flip/or reversal to a change of a slab dip direction (as also recommended by RC1) to avoid misunderstanding, though both these terms as used in literature in a broader sense.

*RC2:* Resolution test 1 also mimics the real pattern quite well below 100 km depth while test 3 misses the increasing dip angle of the real pattern with depth. My remark regarding the treatment of crustal structure also applies to the resolution test. Since crustal corrections were subtracted before inversion of the real data, resolution test data should also be free of crustal contributions and anomalies in the crustal domain should be forced to zero.

**Reply:** We see substantial differences between the real pattern and that of Test 1 and disagree that the Test 1 mimics the real pattern. Moreover, the test is designed to show that the rays are able to distinguish between one or two heterogeneities (lines 179-180 of the original submission). Resolution test are free of crustal contributions, anomalies in the crustal domain are zero. We correct the previous error in plotting and provide the corrected figures.

**Imaging the high-velocity perturbations in different tomography model:**

RC2: 2nd paragraph: If I read Paffrath et al (2021) correctly they do not postulate a polarity flip for the Central Alpine slab. They state a steep SE dip of the slab and also do not associate it with Adria. This should be corrected.

**Reply:** This question touches the same topic as the one raised by C1 concerning lines 222-224. Please see our answer there. We have never associated the Central Alps with Adria.

*RC2:* In general, I find this section not well structured. The polarity flip issue is discussed in several paragraphs interrupted by a discussion of the  $2_{nd}$  high velocity anomaly underneath the BM. There seems to be no real ordering of thoughts and arguments. This part should be streamlined and rewritten in a concise and non-repetitive way with clear order and structure of arguments.

**Reply:** We modify this section related to images of the E. Alps slab in different recognized tomography models. The main concerns are depths of the positive heterogeneities beneath the E. Alps and Bohemian Massif, dip directions and their provenance. These aspects are hard to separate completely, which can lead to a "no ordering of thoughts and arguments" opinion. We have improved the structure of this section.

RC2: I am strongly worried by the resolution tests shown in Fig. S7. For each one of the detached slab test models there are strong smearing artifacts reaching up to the surface. The artifacts are strongest in the crustal layer and they mimic an either northward or southward dipping continuous slab. I cannot see that this resolution test proves that a detachment can be resolved by the inversion. It rather suggests the opposite. I wonder how the result of these tests would look like if crustal perturbations were forced to zero. Possibly, all artificial positive anomalies between 50 km and 100 km could become much stronger pretending a dipping high velocity feature.

**Reply:** We apologize for the earlier plotting mistake, which we have explained in our answer above. The profiles are redrawn to correct the mistake in plotting, which created false perturbations above 45 km, and which we have not considered during the interpretation. There are no crustal perturbations in any figure. Moreover, we include better tests (new Fig.S7) to show whether our tomography is able or not to image the gap above the delaminated (European) slab beneath the E. Alps.

We also add into the main text a new Fig7, in which we mimic the observed perturbations with two (3% and 5%) heterogeneities, the northern one beneath the BM and southern one beneath the E. Alps. In all our interpretations we respect and follow the body-wave tomography principle that perturbations refer to model velocities within each "layer" (grid plane).

---

## Author Comment (AC6)

Reply to EC1

Dear Editor,

Thank you for your editorial work and comments on our manuscript. We revised our manuscript and figures according to your and reviewer's recommendations and requests. We extended references to your papers. We addressed all the comment from the reviewers in detail in our replies, in a trackable way as you can find in the respective files.

**To your minor comments:**

**EC1:** *Please note that the anomaly you describe below the Bohemian Massif is very similar to the previously reported one in Kästle et al. 2018 ("Surface-wave tomography..."), why also imaged two fast anomalies in the uppermost mantle: one below the eastern Alps and a second anomaly below southern Germany and the southern part of the Bohemian Massif: "A second fast anomaly, also subparallel to the Periadriatic Fault, is visible approximately 1° farther to the north. It has its highest amplitude in southeastern Germany; hence, we denominate it the Eastern Alpine Northern Anomaly (Figure 12)." This would support your finding of a secondary slab/thickened and cold lithosphere north of the Alps.*

**Reply:** thanks for bringing this into our attention. We were aware of this second heterogeneity north of the Alps, but we lost our remark on your et al. 2018 paper. We incorporated into Section Introduction sentences as follows:

Kästle et al. (2018) identified approximately 1° to the north of the Periadriatic Fault a similar subparallel fast velocity heterogeneity in their surface-wave tomography.

Finding a secondary slab/thickened and cold lithosphere north of the Alps strengthens our motivation to determine its proper location and discuss hypotheses regarding its origin.

Modification of the last sentence of Introduction:
Besides linking the shallow heterogeneity beneath the Eastern Alps to subduction of the Adriatic plate, we also present and discuss in our paper three potential scenarios of the origin of the positive heterogeneity located beneath the southern Bohemian Massif (HV-BM).

**EC1:** *l. 37 For a review of different slab models under the Alps, Kästle et al. (2020, "Slab break-offs...") would also be a good reference that discusses the entire Alpine arc.*

**Reply:** We added the reference to Kästle et al. (2020) as recommended

**EC1:** *l. 271 Dando et al. 2011 never clearly state that they attribute the eastern Alpine slab to the subducting Adriatic plate. Instead they report a "Continuity of the Pannonian fast anomaly with the East Alpine fast anomaly" and state "The interpretation that emerges from our images is that a continuous collision zone extended from the Alps through present-day western and central Hungary. [...] When extension began in the Pannonian Basin, the higher velocity material detached from the lithosphere, indeed the Pannonian Basin extension may have been triggered by the detachment of this cold material produced by prior convergence." This would rather be in agreement with the European subduction that is interpreted to be continuous from Alps to Carpathians prior to 20 Ma (e.g., Handy et al. 2015).*

**Reply:** Similarly to our explanation in our replies to the reviewers, referring to Dando et al. (2011) relates to the northward dip of their heterogeneity beneath the E. Alps. The sentence was reformulated to:

Alps from data of regional passive experiments (Dando et al., 2011; Mitterbauer et al., 2011; Karousová et al., 2013) also retrieved the northward dipping high-velocity heterogeneity of similar geometries (Fig. 5) in 250 km of the upper mantle.

---

## Referee Report (RR1)

Reply to EC1

Dear Editor,

Thank you for your editorial work and comments on our manuscript. We revised our manuscript and figures according to your and reviewer's recommendations and requests. We extended references to your papers. We addressed all the comment from the reviewers in detail in our replies, in a trackable way as you can find in the respective files.

**To your minor comments:**

**EC1:** *Please note that the anomaly you describe below the Bohemian Massif is very similar to the previously reported one in Kästle et al. 2018 ("Surface-wave tomography..."), why also imaged two fast anomalies in the uppermost mantle: one below the eastern Alps and a second anomaly below southern Germany and the southern part of the Bohemian Massif: "A second fast anomaly, also subparallel to the Periadriatic Fault, is visible approximately 1° farther to the north. It has its highest amplitude in southeastern Germany; hence, we denominate it the Eastern Alpine Northern Anomaly (Figure 12)." This would support your finding of a secondary slab/thickened and cold lithosphere north of the Alps.*

**Reply:** thanks for bringing this into our attention. We were aware of this second heterogeneity north of the Alps, but we lost our remark on your et al. 2018 paper. We incorporated into Section Introduction sentences as follows:

Kästle et al. (2018) identified approximately 1° to the north of the Periadriatic Fault a similar subparallel fast velocity heterogeneity in their surface-wave tomography.

Finding a secondary slab/thickened and cold lithosphere north of the Alps strengthens our motivation to determine its proper location and discuss hypotheses regarding its origin.

Modification of the last sentence of Introduction:
Besides linking the shallow heterogeneity beneath the Eastern Alps to subduction of the Adriatic plate, we also present and discuss in our paper three potential scenarios of the origin of the positive heterogeneity located beneath the southern Bohemian Massif (HV-BM).

**EC1:** *l. 37 For a review of different slab models under the Alps, Kästle et al. (2020, "Slab break-offs...") would also be a good reference that discusses the entire Alpine arc.*

**Reply:** We added the reference to Kästle et al. (2020) as recommended

**EC1:** *l. 271 Dando et al. 2011 never clearly state that they attribute the eastern Alpine slab to the subducting Adriatic plate. Instead they report a "Continuity of the Pannonian fast anomaly with the East Alpine fast anomaly" and state "The interpretation that emerges from our images is that a continuous collision zone extended from the Alps through present-day western and central Hungary. [...] When extension began in the Pannonian Basin, the higher velocity material detached from the lithosphere, indeed the Pannonian Basin extension may have been triggered by the detachment of this cold material produced by prior convergence." This would rather be in agreement with the European subduction that is interpreted to be continuous from Alps to Carpathians prior to 20 Ma (e.g., Handy et al. 2015).*

**Reply:** Similarly to our explanation in our replies to the reviewers, referring to Dando et al. (2011) relates to the northward dip of their heterogeneity beneath the E. Alps. The sentence was reformulated to:

Alps from data of regional passive experiments (Dando et al., 2011; Mitterbauer et al., 2011; Karousová et al., 2013) also retrieved the northward dipping high-velocity heterogeneity of similar geometries (Fig. 5) in 250 km of the upper mantle.

Dear Reviewer 1,

Thank you for your comment on our manuscript. We report all your comments in red below, followed by our respective answers. The main manuscript is revised accordingly, some of the modified sentences are include below (underlined text).

Reply to RC1

Introduction:
**Lines 31-32:** "The classical concept assumed the European lithospheric slab subducted southeastward/southward along the entire Alpine chain (Mueller, 1982) without any fragmentation."

I would specify "Alps" rather than "the entire Alpine chain" because Mueller was referring only to the Alps sensu stricto, not to other parts of the Alpine chain which also includes the Apennines and Carpathians. The latter involve subduction of Adriatic lithosphere. It would also be appropriate to cite colleagues who previously proposed lithospheric subduction in the Alps (e.g., Laubscher 1970) inspired by the much earlier "classical" work of Ampferer and Argand.

**Reply:** We have modified the sentence as suggested:

The classical concept assumed the European lithospheric slab subducted south-eastward/southward along the entire Alps (Laubscher 1975; Mueller, 1982) without any segmentation.

*Line 40:* "…reversed polarity…" Here, I would recommend using the descriptive term "opposite dip" rather than "reverse polarity". The latter is more interpretative and implies a change in the direction of subduction.

**Reply:** We see your point and have modified the sentence as suggested, though "opposite" is not the exact expression and the term "reversed polarity" in sense of a change in dip direction is generally used in a broader sense than suggested in the review (e.g., Vignarolli et al., 2008). We substitute the word fragment, fragmented with more propper segment/segmented throughout most of the text.

For the first time, Babuška et al. (1990) imaged the Alpine slabs separated into two segments, one beneath the Western and second beneath the Eastern Alps, with opposite polarity and a gap in between them.

*Line 43:* "….with a gap between the two Alpine keels…" Please define what you mean by "keel". You probably mean slabs. Note that Mitterbauer et al. (2011) preferred an interpretation in which there is no slab gap between the Central and Eastern Alps.

**Reply:** we change word keel to slab as suggested, though the word keel is often used in many papers for dipping/thickened edge of the lithosphere (the whole lithosphere, not only crust; in that case the word "root" is usually used). We attempted to avoid frequent usage of word slab. We have seen the (vertical) slab gap even in our old studies in the 80s, which was later documented in other papers as well, e.g., Lippitsch et al. (2003).

New wording:

…. with a gap between the two Alpine slabs,……

*Line 58:* Nomenclature – "the south-eastern part of the BM (referred to as HV-BM throughout the paper)…" Please keep things simple by using BM = Bohemian Massif. Or if you really need this long abbreviation (HV-BM), then write it completely to make your usage clear.

**Reply:** Abbreviation "BM" is introduced for the Bohemian Massif as a whole in the abstract of the original submission (line 15) and Introduction (line 57). Abbreviation HV-BM is introduced as a logical expression for the "high-velocity heterogeneity beneath the southern part of the BM", i.e., only a part of the BM and at depth. Repeating "high-velocity heterogeneity beneath the southern part of the BM" throughout the text would be long and non-effective. Moreover, not all the BM mantle is fast, but rather slow (e.g., Piromallo & Morelli 2003; Amaru 2007; Koulakov et al. 2009). Using only BM for the high-velocity heterogeneity beneath a part of the BM would be confusing for those who are familiar with the slow sub-lithospheric upper mantle beneath the BM, in general. Therefore, we will keep the abbreviation HV-BM and complement the captions.

**Resolution tests:**

*Lines 201-205*: "The above performed synthetic tests corroborate that the data from the AlpArray and EASI networks are able to image two separate northward dipping sub-parallel slabs beneath the E. Alps and southern rim of the BM. The two slabs are separate from each other, and the northern one is not connected with the shallow parts of the lithosphere (above ~100km). The flip of the subduction polarity beneath the E. Alps relative to the W. Alps is undoubtedly real and it is not produced by potential smearing due to ray geometry." I would distinguish between slab dip (the direction that the slab is presently slanted) and subduction polarity (the original direction of subduction). The steep northward dip of the slab beneath the Eastern Alps is a robust feature of all models so far, but the original direction of subduction is debatable (see comments below). For comparison of end-member models with different slab dips, e.g., resolution tests of Paffrath et al. (in rev.).

**Reply:** We modify the sentence as suggested,

The difference between the dip directions of the subduction beneath the E. Alps and the subductions beneath the W. and central Alps is undoubtedly real and it is not produced by potential smearing due to ray geometry.

**Connection of surface to subsurface dynamics:**
*Lines 226-229:* "The uplift rates in the W. and C. Alps exhibit at least 50% contribution by convective processes (due to slab detachment) and dynamic contributions (due to the sub-lithospheric mantle flow), while isostatic response due to ice unloading during deglaciations dominates in the E. Alps (Sternai, 2019)." There are other modelling studies that differ from those of Sternai 2019 in both their approach and their conclusions; they warrant reading and citation in this context (e.g., Champognac et al. 2007, Fox et al. 2015, Mey et al. 2016).

**Reply:** We chose to delete this paragraph, which does not concern the main topic of the study. More explanation would make the ms. unnecessarily longer.

**Slab polarity and the boundary between the Eastern and Central Alps:**
*Lines 222-224:* "In recent studies, Paffrath et al. (2020) suggest the reversed slab polarity relative to the Western Alps already in the Central Alps, as opposed to the formerly documented polarity reversal further to the east - beneath the E. Alps (e.g., Babuška et al., 1990, Lippitsch 2003; Zhao et al., 2016)." This is a misunderstanding/misreading, because Paffrath et al. don't claim that there was a reversal in slab polarity below the Central Alps! Please read carefully. Rather, they refer to the part of the slab beneath the Eastern Alps when they write „The new model does not require a slab polarity switch due to the detachment from the crust and the nearly vertical dip of the eastern Alpine slab. Tectonic arguments (in Handy et al., in rev. with Solid Earth) rather suggest a European provenance of the slab".

**Reply:** We have read carefully both the submitted paper by Paffrath et al. as well as studied the presentation at EGU2020 (reference given), where on slide 15 the authors compare their results with results of Lippitsch et al. (2003) along slice B  running through the C. Alps and  show:

Discrepancy in lithospheric slab orientation, slab connected to Adria?
Dipping direction seems to be reversed (NW) in comparison, however structure of this slab is highly complex with eastern part showing tendency to dip SE

 Our sentence "In recent studies, Paffrath et al (2020) suggest the reversed slab polarity relative to the Western Alps already in the Central Alps, as opposed to the formerly documented polarity reversal further to the east - beneath the E. Alps (e.g., Babuška et al., 1990, Lippitsch 2003; Zhao et al., 2016)" refers correctly to this presentation.  The reference to Zhao et al., 2016 might be deleted as they do not have sufficient resolution beneath the E.Alps.  We can also delete the entire sentence in case statements in slide 15 of the presentation are no more valid, as it is not essential for our ms., in which we focus on the E. Alps and the BM. At this stage of our ms. we do not say anything about the slab provenance.

*Lines 224-226:* "Mock et al. (2020) pointed out the discordance between the slab geometry at depth and the boundary between the Eastern and Central Alps observed in the surface geology and, similarly to Rosenberg et al. (2018), shift the boundary between the E. and C. Alps further to the east, at the Giudicare-Brenner fault system." The authors cited do not redefine the boundary between Eastern and Central Alps, they merely note a discordance between the slab geometry at depth and the boundary between the Eastern and Central Alps observed at the surface. The boundary between the E. and C. Alps is defined at the surface in map view by the western edge of the continuous Austroalpine nappe stack located in Eastern Switzerland. This traditional definition does not take into account the effect of erosion which removed Austroalpine nappes originally exposed somewhat further to the west. The former existance of these nappes to the west is supported by the provenance of components in the Molasse Basin and by the Austroalpine Klippe in the Central Alps. Regarding the Alpine slab, the boundary between the central and eastern slab segments (as defined either by a slab gap or a change in dip direction of the slab) coincides generally with the Giudicarie-Brenner faults. In light of recent evidence for a laterally continuous Alpine slab with a change in dip (Handy et al., in review with SE), it would be clearer to avoid relating surface and subsurface boundaries between Central and Eastern Alps.

**Reply:** Wee see your point and chose to keep a shortened delimitation of the C./E. Alps boundary, regardless of the interpretation. The rest of the paragraph has been deleted:

Mock et al. (2020) note a discordance between the slab geometry at depth and the boundary between the Eastern and Central Alps observed at the surface. The boundary between the central and eastern   slab segments of the Alps, as defined either by slab gaps, due to delamination (Handy et al.,  2021)  or a change in dip direction of the subductions, coincides with the Giudicare-Brenner fault system, in general (e.g,  Rosenberg et al.,  20   ).

*Lines 238-241:* "Later tomography of the upper mantle which included the E. Alps from data of regional passive experiments (Dando et al., 2011; Mitterbauer et al., 2011; Karousová et al., 2013) also retrieved the northward dipping high-velocity heterogeneity of similar geometries (Fig. 5) and associated it mostly with the Adria plate subduction." It is important to avoid equating a switch in the dip direction (polarity) of the Alpine slab with a gap in this slab. In the interpretation of Handy et al. (in rev. with Solid Earth), the slab changes its dip direction along strike without evidence for lateral separation into two segments. Rather, the along-strike change in dip direction of the slab coincides generally with an eastward transition in its degree of detachment from the orogenic lithosphere.

**Reply:** We modified the sentence, in which we delete the last part. Reference to Dando et al. (2011) for the slab relation to the Adriatic plate was included by a mistake.

Later tomography of the upper mantle, which included the E. Alps, from data of regional passive experiments (Dando et al., 2011; Mitterbauer et al., 2011; Karousová et al., 2013) also retrieved the northward dipping high-velocity heterogeneity of similar geometries  (Fig. 5) in the 250 km of the upper mantle.

To the comment segmentation vs. change in dip direction:

Along-strike change in dip direction of the slab (without a gap) is conditioned by the slab detachment to the east.  In case of no or "thin" slab detachment (irresolvable at the current 30x30x30 km grid), as imaged in several tomographies including ours, the change in subductions calls for slab segmentation, or a "vertical" gap (separation) between the two subductions (regardless how they are interpreted). When interpreting results of body-wave tomography, we strongly follow principles of P- velocity perturbation images, i.e., deviations from a 1D reference velocity model. The damped deviations referes to each layer separately and do not allow direct comparison with absolute velocities in other models. One has to be careful with interprations of vertical cross-sections as well.

**Lines 241-246:** "….suggested three main phases in building the E. Alps keel: (1) NW translation of the Adria and its thrusting over the (subducting?) European plate in the W. Alps, (2) fragmentation of northern Adria along a deep-seated fault (possibly the Giudicarie Fault, or at least a spatially nearby structure) and (3) counter-clockwise rotation of the Adria and its subduction beneath the European plate in the Eastern Alps, with a triple-junction of three crustal terranes in its eastern rim proposed by Brückl et al. (2010), although the deformation style between the E. Alps and the Pannonian Basin is usually considered diffuse on the surface." Two points need clarification here: (1) Based on recent P-wave tomography from the newest Alp Array data (Paffrath et al. in rev), the length of slab beneath the Eastern Alps, irrespective of its dip direction, far exceeds the amount of shortening in the eastern Southern Alps. This precludes a primarily Adriatic origin of this slab, as discussed in Handy et al. in rev, Solid Earth; (2) there is no unequivocal evidence at the surface or at depth to support the idea of three "crustal terrains" with distinct Moho boundaries near the transition from

E. Alps to the Pannonian Basin. Rather, the Pannonian Basin is underlain by thinned European and Adriatic crust and lithosphere that acquired their reduced thicknesses in Miocene time (e.g., Ustaszewski et al. 2008). The decrease in Moho depth going from the Eastern Alps to the Pannonian Basin is gradual, not abrupt (e.g., Horvath et al. 2015), indicating extensional flow of the lower crust and upper mantle.

**Reply:** ad comment (1): we refer to our interpretation from the '90s, reflecting the LAB mapping and results of P-wave tomography, both showing differences between the Western and Eastern Alps. NW movement of Adria, its counterclockwise rotation and collision with the European plate is not questioned (e.g., Le Breton et al., 2017), including the amount of shortening (Ustaszewski et al., 2008). We modify the text as follows:

The imaged triangular shape of the LAB model (e.g., Babuška and Plomerová 1992) beneath the thickened lithosphere of the E. Alps detached from that beneath the W. and C. Alps led Babuška et al. (1990) to suggest three main phases in building the E. Alps mantle-lithosphere root (1) NW translation of the Adria and its thrusting over the (subducting?) European plate in the W. Alps, (2) fragmentation of northern Adria along a deep-seated fault (possibly the Giudicarie Fault, or at least a spatially nearby structure) and (3) counter-clockwise rotation of the Adria resulting in a start of its subduction after a collision with the European plate in the Eastern Alps (Ustaszewski et al., 2008), potentially above a delaminated European lithosphere residing at greater depth (Paffrath et al., 2021; Handy et al., 2021).

Ad (2) triple junction (Brückl)

R**eply:** In tectonically complex regions, as the Alps and surrounding area, one can hardly expect that only one fixed opinion exists. We modified the end of the sentence related to the crust and complemented the references as suggested.

Complexity of this region is reflected in ambiguous views on the crust, being interpreted as a triple-junction of three crustal terranes (Brückl et al., 2010), although the deformation style between the E. Alps and the Pannonian Basin is usually considered diffuse on the surface. Instead, e.g., Ustaszewski et al.(2008) and Horvath et al. (2015) assume the Pannonian Basin underlain by gradually thinned European and Adriatic crust, which has been recently imaged by Kalmár et al. (2021).

**Lines 267-277:** "The north oriented dip of the EA subduction, imaged in early tomography studies…..Most of them agree in interpreting it being of Adriatic plate origin (Dando et al., 2011; Karousová et al., 2013). However, Mitterbauer et al. (2011), and similarly in recent tomography by Paffrath et al., (2021 this issue), the positive perturbations are associated with a delaminated EU slab. Kästle et al. (2020) relate the HV-EA mainly with the European plate subductions as well, and leave no or only minor role to the Adriatic subduction. The authors explain the northward subduction modelled beneath the E. Alps by imaging a combination of the short Adriatic and deep delaminated, potentially overturned European slabs." Here, the authors should mention that recent P-wave tomography based on the newest Alp Array data (Paffrath et al. in rev) supports a primarily European origin of the north-dipping slab beneath the Eastern Alps (see discussion in Handy et al., in rev. with Solid Earth). This conclusion is based on the length of imaged slab beneath the Eastern Alps (several hundred km) which far exceeds the amount of shortening (c. 50 km) in the overlying crust of the eastern Southern Alps. In this interpretation, the steep northward dip of the slab segment beneath the E. Alps is attributed to asthenospheric flow during or after Adriatic indentation (Handy et al., rev.).

**Reply:** The requested formulation is included in the original submission (see above)**.** We extend the sentence and add the Handy et al., 2021 this issue refence as well).

… and similarly in recent tomography  based on the newest Alp Array data by Paffrath et al., (2021 this issue), the positive perturbations 🔖w 150km are associated with a delaminated European slab (see also Handy et al., 2021 this issue).

**Lines 401-402:** "The northward-dipping lithosphere keel is imaged down to ~200-250 km, without signs of delamination, and we associate it with the Adriatic plate subduction." As above, it is unclear what the authors mean by "keel". It appears to mean slab, but this should be clarified.

**Reply:** The sentence was modified (keel deleted):

The northward-dipping slab is imaged down to ~200-250 km, without signs of delamination from the Adriatic plate. Therefore, associations of the high-velocity heterogeneity with the Adriatic plate subduction seem to be obvious.

**Interpretations of anomalies beneath the Bohemian Massif**

**Lines 281-285:** "To understand the positive perturbations beneath the southern BM, we compare it with results from the large-scale Paffrath et al. (20201) tomography and with the regional tomography of the BM (Karousová et al., 2013) of similar resolution to ours (Fig. 6). The strongest positive perturbations related to that heterogeneity overlap in the models, though they are of unrealistically large extent in the Paffrath's et al. model (2021, this issue)." Why is this extent in the Paffrath et al. model unrealistic? "There, they include also the ER with the thinnest BM lithosphere, well imaged by negative perturbations in the EASI-AA and in BOHEMA models (Karousová et al., 2013)."

**Reply:** We modify the sentence to make it clearer and added reference Plomerova et al. (2016).

Paffrath et al. (2021) show the strong positive perturbations also beneath the north-western part of the BM (beneath the Eger Rift), where the lithosphere thins significantly (~80 km), which is well imaged by negative perturbations in the EASI-AA and in BOHEMA models (Karousová et al., 2013; Plomerova et al., 2016).

**Lines 308-311:** "The role of applying proper crustal corrections is significant in teleseismic regional tomography. Not applying any crustal corrections or applying inadequate ones can strongly affect velocity perturbations within the upper ~100 km of the upper mantle (e.g., Karousová et al., 2013), which is the zone, where the models discussed above differ. From this point of view, developing a uniform detailed and reliable model of the European crust is urgently needed." I agree and find the comparison of models in this chapter valuable. I've sent the authors some PPT slides of a recent talk showing the correspondence between models regarding the degree of connectivity of slabs with their orogenic lithosphere. Both the recent P-wave tomography employing 3D crustal models (Paffrath et al. in rev.) and the surface wave models of Kaestle et al. (2018, 2020) indicate slab detachment in the Western and Eastern Alps, with local slab attachment only in the western Central Alps. This coincidence of images with different methods suggests that the crustal correction of the Pwave model of Paffrath et al. is robust.

**Reply:** We extended the paragraph by a sentence as follows

"Overcorrecting" the travel times due to a crustal model used, regardless of a method of correcting itself, can delete/erase or substantially reduce positive perturbations in the upper 100 km, if too slow and thick crust would be considered and vice versa.

**Lines 332-333:** "The simplest explanation would be to consider it as a fragment of the delaminated part of the European plate subductions, as suggested in Handy et al. (2015) (Fig.7a )." This appears to be a misunderstanding: The +Vp anomaly below the BM is not the anomaly referred to in Handy et al. 2015 (their Fig. 11a), which was a European slab hanging directly below the Eastern Alps in Miocene time. Note that Handy et al. 2015 never considered a +Vp anomaly below the BM in their model. Instead, the Miocene +Vp anomaly below the Eastern Alps in their figure is torn European lithosphere that sank and no longer exists. The model of Handy et al. 2015 is actually consistent with the statement in Line 338 that "Therefore, an association of the HV-BM with the delaminated fragment of the EU subduction is not likely."

**Reply:** Our formulation was not clear enough, if it lead to the idea described by the referee. We did not mean that at all. We reformulate the sentence.

The simplest explanation would be to consider it as a small fragment of the delaminated part of the European plate subductions, as suggested in Handy et al. (2015) for the heterogeneity beneath the Eastern Alps at greater depth (Fig. 7a, now Fig. 8a ).

**Points pertaining to rock physics:**

**Lines 347-348:** "The Phanerozoic continental mantle lithosphere, composed of originally lighter rocks than those in the asthenosphere (Mantle lithosphere is generally denser rather than lighter than asthenospheric mantle) and becomes denser due to metamorphic phase changes as it subducts" (reference?)

**Reply:** The mantle lithosphere alone is on average a bit (ca. 70 kg/m3) denser then the asthenosphere. The mantle lithosphere remains above the asthenosphere as it is mechanically coupled to the crust, so the overall lithosphere density is less than the asthenosphere density. However, as soon as the upper or the full crust is decoupled from the mantle lithosphere, the lower lithosphere (denser than the asthenosphere) is able to sink. The ms. text is updated accordingly:

Updated text:

The Phanerozoic continental mantle lithosphere may mechanically decouple from the upper or the full crust, and since this lower lithosphere is denser than the asthenosphere it subducts. Densification of the lower crust through metamorphic reactions may enhance this process if the convergence is fast (e.g., 2 cm/yr in Hetényi et al., 2011).

New reference: Hetényi G, Godard V, Cattin R, Connolly JAD (2011) Incorporating metamorphism in geodynamic models: the mass conservation problem. *Geophys J Int* **186**:6-10. doi:10.1111/j.1365-246X.2011.05052.x

**Lines 360-363:** "The roll-back subduction of the Carpathians, accompanied by a substantial asthenospheric flow, could open a space between the E. Alpine and Carpathian slabs for the northwestward "transportation" of a purely oceanic lithosphere or a mix of oceanic and continental

lithosphere fragment into the mantle beneath the BM." No reason or mechanism is provided here to explain why oceanic or mixed ocean-continent lithosphere should be "transported" northward. Please elaborate.

**Reply:** there is no current data to demonstrate this, like unequivocal anisotropy due to mantle flow, as such signals have been overprinted. Nevertheless, our proposal here is based on Kissling and Schlunegger (2018)'s model of the Alps, accompanied by complex asthenosphere flow in context of slab edges (e.g., Vignaroli et al., 2008, their Figure 8 for the Western Alps. (https://doi.org/10.1016/j.tecto.2007.12.012) and other examples. For the context discussed in our manuscript, such flows could have happened around the retreating and /or delaminated slabs. The manuscript text now explicitely says this.

Manuscript change, the new sentences read as follows:
The Carpathians front curved significantly [mo] migrated to the north. Differences in the roll-back subductions of the Alps (e.g., Kissling and Schlunegger, 2018 and reference therein) and the Carpathians, northward push of Adria and European slab delamination beneath the EA could have formed complex flows in the asthenosphere (e.g., Vignaroli et al., 2008), which  could "transport"  a purely oceanic lithosphere or a mix of oceanic and continental lithosphere fragments through the open space be[w]en the E. Alpine and Carpathian slabs north-northeastward into the mantle beneath the BM. The rotational displacement of the Adriatic Plate indenter  provided an additional driving force for modifications of the Alpine-Carpathian-Dinaridic orogenic system.

New references:
Ustaszewski et al., 2008
Kissling and Schlunegger, 2018.
Vignaroli G, Faccenna C, Jolivet L, Piromalo C, Rossetti F (2008) Subduction polarity reversal at the junction between the Western Alps and the Northern Apennines, Italy. *Tectonophys* **450**:34-50. doi:10.1016/j.tecto.2007.12.012

**Lines 380 - 396:** "The roll-back subduction of the Carpathians, accompanied by a substantial asthenospheric flow, could open a space between the E. Alpine and Carpathian slabs for the northwestward "transportation" of a purely oceanic lithosphere or a mix of oceanic and continental lithosphere fragment into the mantle beneath the BM." No reason or mechanism is provided here to explain why oceanic or mixed ocean-continent lithosphere should be "transported" northward. Please elaborate.

**Reply:** The continental mantle lithosphere is formed by domains with their own large-scale fabric, oriented generally in 3D, i.e., with inclined symmetry axes of anisotropy ( e.g., Babuska and Plomerova, 2020). For example, In case of two adjacent lit[os]here domains with fabrics, in which the dipping high-velocity directions face each other (i.e., they are oriented convergently), an artificial positive heterogeneity could be produce at greater depth in isotropic images of the upper mantle.

We added an explanation after the quoted sentence
"Ignoring seismic anisotropy and assuming isotropic wave propagation or considering only azimuthal and/or radial anisotropy leads to significant isotropic and anisotropic imaging artefacts that may lead to spurious interpretations (VanderBeek and Faccenda, 2021)."

In this study of the broader region around the E. Alps we have applied, in the first step,  the isotropic mode of  a coupled anisotropic–isotropic teleseismic P-wave tomography developed by Munzarová

et al. (2018).  In spite of the general good agreement with the high-resolution large-scale isotropic tomography (Paffrath et al., 2021, this issue), the images can be biased due to seismic anisotropy (Eken et al., 2012; Qorbani et al., 2015, 2016; Bokelmann et al., 2021).  Laterally varying anisotropy, which correlates with tectonics of the region, has been indicated in shear-wave splitting (e.g. Link and Rumpker, 2021).  After collecting sufficient amount and well-distributed high-quality data, we will run the coupled anisotropic-isotropic mode of the code and retrieve 3D anisotropic model of the region, which will allow for laterally and vertically varied anisotropy with axes oriented generally (inclined) in 3D. This further investigation may help in deciding among the drafted scenarios for the origin of the HV-BM, or point to new ones.

New reference: Link and Rumpker, 2021, feart, doi: 10.3389/feart.2021.679887

**Figures and Captions:**

**Fig. 1** (right side): Label the main faults and use transparent colour to mark the Alps between the Northern and Southern Alpine Fronts.

**Reply:** We add labels to main faults**,** but do not mask the Alps with any schematized colour symplification.

**Fig. 2** (right panel): Label the Southern Alpine Front (SAF), use transparent colour to mark the Alps between the Northern and Southern Alpine Fronts. It would help to mark the outlines of the Tauern Window (TW). I would eliminate the triple junction, because there is no structural evidence for a Pannonian Plate sensu Brückl et al. (see comments above).

**Reply:** The same as for Fig.1. Labels added, the Tauern window contoured. The triple junction deleted.

**Fig. 3**: Label the Southern Alpine Front (SAF) on all cross sections. In the caption, indicate what DF means (this is not included in any of the earlier captions). Indicate the traces of all cross sections on a map or inset map. Include coordinates of the endpoints in the caption or directly on the figures. This is important for others who wish to compare their results with the authors'.

**Reply:**  SAF Label added at all cross-sections, DF included among explanations in Figs.1 and 2, traces of all cross-section marked in the horizontal slice at 120km (new upper right inset). Geographic coordinates of endpoints of all the along longitudes crosssections are complemented in the captions.

**Fig. 4**: Label all the faults shown in the other cross sections. Include coordinates of the endpoints in caption or directly on the figures. Again, this is important for others who wish to compare their results with the authors'.

**Reply:** Labels complemented, coordinates added in the caption.

**Fig. 5**: The labelling of the cross sections should be made clearer. Label each section only once so it corresponds to the letters of the traces on the map. The orogen-parallel cross section in a) should have the longitudinal degrees correspond to the longitudinal units on the map on the lower right.

**Reply:** Required change from landscape to portait destroyed the figures, which escaped my attention. Figure is redrawn with correct labelling. For better visibility of the heterogeneities we keep part (a) with a broader extent of longitudes.

**Fig. 6**: Add Variscan unit boundaries on the main horizontal slice of Paffrath to enable better comparison with the other slices shown. Distinguish Alpine and Variscan boundaries, either by using lines with different thicknesses (Alps thick, Variscan thin) or colours.

**Reply:** We prefer keeping the Paffraths et al. plots at as close as possible to their original form, the Alpine and Variscan boundaries are distinguished in the two other parts of our figure.  We label all important faults and units.

**Fig. 7**: Check the position of the cross sections with respect to the seismic anomalies, because they look misaligned. The northern tip of the delaminated EU slab in a) is too far to the north in b) with respect to the Northern Alpine Front (NAF). Handy et al. 2015 proposed that this slab sank beneath the core of the Alps, not that it migrated to the north with respect to the front. However, if this is your idea (meaning the authors of this text), then please state this clearly.

**Reply:** This is probably a misunderstanding. We slightly corrected the position of the delaminated slab  in the schematic cartoon plotted across  backround from  Handy et al. (2015) and complemented the caption.

**Fig. 8**: Show the end points of the cross sections on an inset map so that the location of the. anomalies and unit boundaries can be compared, otherwise the model can't be compared with other data and the reader is helplessly lost.

 In b), please also consider other interpretations of Late
Devonian to late Carboniferous paleogeography and subduction directions of the Rheic Ocean, e.g., Schulmann et al. 2009, 2014, Ballevre et al. (2009), Zeh & Will (2011) or in Franke et al. (2000). What effect do these scenarios have on your interpretation of the 3% positive Vp anomaly beneath the Moldanubikum?

**Reply:**  Figure 7b is part of the schematic cartoon (not to scale) with  reference on Babuska and Plomerova (2013), where details of the cross-sections are presented and other interpretations as requested are discusseed and referred in Section 6.  We complement the captions.

We present some of definitely more potential scenarios, but at present stage, we do not have enough information to favour only one of them, neither dispute with other  interpretation of the Reic Ocean subductions. Arguments for our model of BM we have presented, e.g., in  Babuska and Plomerova (2013) and reference it in this ms.

Dear Reviewer 2,

Thank you for your comment on our manuscript. We report all your comments in italic below (preceded by "RC2"), followed by our respective answers. The main manuscript is revised accordingly, some of the modified sentences are include below (underlined text).

Reply to RC2

**Introduction:**

*RC2: The introduction is surprisingly short. I miss a proper description of the state of the art (what is known), of controversial issues (what is debated) and of how the manuscript contributes to answering some of these open questions. The introduction does not pose any research questions and thus entirely lacks a motivation for the work presented. I also miss a proper description of previous work done with EASI data and its relevance for the current work. At the end, a brief description of the contents of the paper would be welcome. Instead, the last paragraph rather resembles an abstract.*

**Reply:** We added sentences on actual stage of debates on the E. Alps slab (scientific questions), our motivation for the work and reformulate description of the content of the paper as requested. With that, we consider our description of the classical view and state-of-art of the tomography images of the Alps in the first two paragraphs of the Introduction as representative and sufficient.

While resolution of early tomography of the Alpine region allowed researchers to deal only with the most distinct heterogeneities in the upper mantle, accumulation of high-quality data from the dense AlpArray network motivated us to search finer images of the upper mantle and answers on segmentation of the Alps, dip directions of the subductions, their relevance to European or Adriatic plate, extent of slab delamination and particularly, to elucidate the smaller size heterogeneity beneath the Bohemian Massif north of the E. Alps subduction.

*RC2: The meaning of the lines in the tectonic map in Fig 1 needs to be explained in the caption.*

**Reply:** Captions complemented.

**Data:**

*RC2: I assume that also data of the complementary EASI experiment are used. This should be made clear in the text.*

**Reply:** Usage of the AlpArray-EASI data is presented in the 1$^{st}$ sentence of the abstract and in the 3$^{rd}$ par. of the Introduction of the original submission, as well as in the Section Data. Nevertheless we modified the original lines 69-72 of Section 2. Data. It sounds in the revised version as follows:

We collected recordings from stations of the AlpArray Seismic Network (AASN, doi.org/10.12686 /alparray/z3_2015) and AlpArray-EASI network (doi.org/10.12686 /alparray/xt_2014). All the AASN

stations installed, in a 200km-wide band (Fig. 1) along the densely spaced stations of the AlpArray-EASI complementary experiment (Hetényi et al., 2018b), were selected for this study.

*RC2: Which procedures for checking data quality have been applied? The paper should be self-contained to some extent.*

**Reply:** We checked data for its completeness and correct timing (uncorrected leap second, failures in clock synchronizations) and station metadata to fix several mistakes (e.g., wrong channel name, station mislocation, poles and zeros given in Hz instead of radians/s).

We have applied several procedures to check the data quality (Vecsey et al., 2017), particularly data completeness and correct timing, to eliminate periods with the uncorrected leap seconds or failures of lock synchronizations. In station metadata files we fixed for example wrong channel naming, station mislocation, and incorrect units for poles and zeros (mixing Hz and radians/s units).

*RC2: The description of the picking scheme and the event distributions should be put into main paper.*

**Reply:** We consider this part as valuable for deep specialists and we prefer keeping it in the supplementary materials, which are easily accessible, for those who are interested. We do not want to extend further the main text due to an earlier experience with Solid Earth about manuscript length.

*RC2: Which trace is used as reference for cross-correlation and beam forming?*

***Reply:*** explanation added

A low-noise beam trace created from stacked cross-correlated and shifted traces of an event serves as a reference in the second cross-correlation step and beam forming in the P-arrival time picking. This means, there is no subjective a priori selected single reference trace.

RC2: *Quantitative statements regarding error estimation of the picks and their probabilistic combinations should be made.*

**Reply:** We modified the text, added a new part to Fig.S2 (part S2b, Histograms of uncertainties, means, medians) and extended caption of Figure S2:

The final time of each extreme (green P1, P2 in Fig. S2a) and its error estimate is computed from the normal distribution which approximates a mixture of normal distributions of partial picks.

Figure S2b shows uncertainties of the measured P-wave arrivals, means and medians for both the complete dataset as well as for events selected for tomography (see below).

Each of the red, black and blue picks is complemented by its error estimate defined as a standard error of the normal distribution. Time error of the red extreme depends on a signal noise level (see cyan basins, their height is given by a noise magnitude - red dashed lines), errors of the black and blue correlation picks come from coherence of the signal with the beam. The final time of each extreme (green P1, P2 ) and its error estimate is computed from the normal distribution which approximates a mixture of normal distributions of partial picks (red, black and blue P1, P2), ……

*RC2:* *The description of the enhanced data set is confusing. Is the enhanced dataset just the original one with 201 events (in Figure S1 209 events) plus another 43 events within the 60 degree back-azimuth cone? Or is it a completely new selection from the overall available events containing only events within the 60 degree cone? This should be clarified in the manuscript. Regarding the figures S1,* **the latter seems to be the case**.

**Reply:** We stated on lines 86-88 of the original submission
„To enhance the resolution in direction of the subducted plates, we selected further rays coming from the northern and southern 60° wide azimuth bins to be included in the tomographic inversion."

Of course, the reason to take only rays from the northern and southern 60° wide azimuth bins was to minimize effects of heterogeneities east and west outside the elongated model. Thus you are right: "the latter is the case". Besides showing the event distributions in Fig. S1, we also include Fig. S5 (original submission, and text there on L90-91), which shows the enhanced resolution along the vertical N-S cross-sections, if all rays are included (left, rays as in FigS1a) and if rays from sides of the model are excluded (right, rays as in Fig. S1b). This means, the original submission has already contained the above recommendation of the RC2. We modify the text and caption of Fig. S5 to prevent misunderstanding:

To eliminate mapping effects of heterogeneities aside the model into its internal part and to enhance the resolution in direction of the subductions, we selected additional rays coming from the northern and southern 60° wide azimuth bins. Only rays propagating through the model within the two azimuthal fans (see Fig. S2) have been included in the final tomographic inversions, tested, discussed and interpreted further. This data comes from 244 earthquakes, each of them being recorded by 120 stations on average, i.e., by 50% of stations in the region.

**Figure S5**: Velocity perturbations along Profile EASI through models calculated for rays from all directions (left) and for rays from the northern and southern 60° wide azimuth bins (right, see also Fig. S1). Relatively less-well resolved regions along the profile are shaded.

*RC2: There are two figures S1 in the supplement. Renumber to S1 and S2.*
**Reply:** Figure S1 has parts a) and b). The second label was missing in the second part of the figure of the original submission, now complemented. Caption and numbering of following figures remain correct.

*RC2: Information on crustal structure is taken from several sources. How is the 3D crustal model put together? How are transitions between subregions treated? How are crustal discontinuities treated, as transition regions or as real discontinuities? At least two horizontal sections through the 3D crustal velocity model should be shown. To which depth does the crustal model reach? Does it also contain parts of the uppermost mantle?*
*How is the correction for crustal structure done? One correction for all events assuming vertical incidence or event-wise corrections taking correct incidence angles into account?*

**Reply:** we compile information for each station and its surrounding, there are no transitions between subregions, individual crust parameters go to Moho depth, i.e., it does not contain the uppermost mantle. The absolute residuals relative to the IASP91 model are corrected for the real crust deviations from the model. For each ray, refractions in the crust according to incidence angle and azimuth are considered. We never applied one correction for all events assuming vertical incidence, but event-wise corrections, which take correct incidence angles into account.

*RC2: The meaning of the end of the following sentence (line 100) "…. proper tomographic inversion in our target region to resolve structures, including the spatial limits of our images" is unclear? What are the spatial limits of the images? Do the authors want to say that resolution is also good at the model boundaries?*

**Reply:** Of course, we did not want to say anything like that. We simplify the sentence.

With this approach we gathered a high-quality and uniform dataset of travel time residuals for a proper tomographic inversion to resolve structures in the target region.

*RC2: Estimated uncertainties of the travel time residuals should be documented in the manuscript. Show a histogram of the uncertainty distribution and give average and median values.*
**Reply:** As above, new Figure S2b with histograms of uncertainties, mean and medianhas been included in the revised version.

***METHOD***

*RC2: It is stated that the matrix W_m in equation (4) provides horizontal smoothing. What about vertical smoothing?*

*REPLY:* There is only horizontal smoothing. There is no option for vertical smoothing in the original code for teleseismic tomography that we use. Both the Telinv and the AniTomo codes have been created without such a possibility in order not to enhance sub-vertical smearing, which is given by the ray geometry.

*RC2: Is there also a damping term included in W_m , and if yes, what is the weight of smoothing relative to damping?*

**Reply***:* Smoothing matrix forces the model parameters at each inverted node to be close to an average of the model parameters in the surrounding inverted nodes. Such a form of the smoothing matrix is fixed in AniTomo and in Telinv and there is no option to change any weight there. Damping factor epsilon**2 is just a constant multiplying every element of the smoothing matrix.

*RC2: With 13 cells in vertical direction, I calculate a model depth of 13\*30=390 km and not 435 km as stated in the text. Is there a specific reason for choosing 390 km (or 435 km) as the bottom of the model. Is this depth still warranted by intersection of rays given the length of the EASI profile?*

**REPLY:** The number of 13 refers only to layers of grid nodes, in which the inversion for velocity perturbations is allowed, i.e., at the 60 km, 90 km, …, 390 km and 420 km. The depth of 435 km corresponds to the bottom of the grid cells that belong to the nodes at 420 km depth. We clarified this in the text. There are still a lot of rays at the depth of 420 km, crossing mainly in the central part of the array. In the northern and the southern parts of the array, the ray crossing is less good compared to the upper parts, of course. See also the new Fig. S7 showing the whole model from the checkerboard test.

The area of about 400 000 km2, centred at 13.3°E 48.5°N, is approximated by 30-by-30 km cell size, horizontally, and with 30 km spacing, vertically. The images are calculated down to 435 km depth on a vertical grid of 30 km spacing. To minimize creating false perturbations, we invert for the velocity perturbations only in the central 5 x 25 x 13 cells, which are well-sampled by criss-crossing rays (Fig. S1), i.e., in nodes between depths of 60 km and 420 km. The model covers the Eastern Alps and a core of the BM, an area of ca. 140 400 km2 in total. Data variance reduction of the final model for the chosen damping parameter attains 66% (Fig. S4).

*RC2: What is the damping factor in Fig. S4? Is it epsilon\*\*2 from equation (4)?*

**Reply:** Yes, damping factor is epsilon \*\*2 from equation (4).

*Why do the authors plot data variance instead of misfit which is normalized to the data uncertainties. What is model variance? Is it the squared norm of the velocity perturbations? A definition of this quantity should be provided. The authors should provide a value of misfit normalized to the picking uncertainties to get an impression whether there is overfitting or maybe even severe underfitting.*

**Reply:** Evaluation of data variance in tomographic codes AniTomo and Telinv includes normalization to the data uncertainties. We modified caption of Fig. S4. Reduction of data variance of 66 % has been mentioned at the end of Section 3.

**Figure S4:** Data and model variance trade-off curve evaluated for various values of damping of the isotropic-velocity perturbations and numbers of iterations. The data variance and model variance are squared norms of the time residuals and velocity perturbations, respectively. The data uncertainties are included in the evaluation of the data variance.

*RC2: Regarding Fig. S4. I wonder that the authors get increasing data variance and decreasing model variance for decreasing damping factor. I would expect exactly the opposite. This strange behaviour needs to be explained by the authors.*

**Reply:** The strange behaviour of the data and model variances in Fig. S4 is caused by a mistake in manual (re-)colouring of the points of the trade-off curve. The model variance should really decrease

with an increasing damping factor. Thank you for noticing that. We corrected the plotting mistake in Fig. S4.

*RC2: Why do the authors only consider 2 iterations? Did they try more iterations and how does the inversion behave then?*

We performed one more iteration, but overall the imaged perturbations remain similar, thus we decided to stop calculations after the second iteration to save the time.

**Results:**

*RC2: In the "Data" section two dataset were presented. Which one was used to create the model shown in the figures? Fig. S5 which compares vertical sections obtained from the 2 data sets is only mentioned once. The issue is never discussed in the results or the interpretative section.*

**Reply:** The main features of the perturbations are the same in both models (Fig. S5), but models containing rays from the west and east could be potentially biased from heterogeneities outside the model. Reasoning for limiting rays into the northern and southern fans is expressed in section Data. All results presented, tested, discussed or interpreted are based on this data set. Modification of the text included in Section Data.

*RC2: Regarding the vertical sections in fig. 3, I recognize significant perturbations in the gray-shaded upper 50 km of the model which appears to be the crustal domain. As the effect of the crust was subtracted from the travel-time residuals, why do the authors still allow perturbations in the upper 50 km? Theoretically, after crustal correction, the travel-time residuals should represent pure mantle structure and perturbations in the crust should be suppressed. How would the model change if perturbations in the upper 50 km were forced to zero? In particular, what would happen with the model between 50 km and 100 km depth, just beneath the crustal domain? The interpretation of a northward dip strongly depends on the velocity perturbations between 50 km and 100 km depth. Below the high velocity anomalies are rather vertical. Whether HV-EA is delaminated or not, also depends on the anomalies in this depth range. All the conclusions following in the paper about polarity flip, northward dip and detachment of the Eastern Alpine slab depend on this issue.*

**Reply:** Relative residuals corrected for the crustal deviations (including sediments, Moho depth, velocity) relative to the IASP91 represent our tomography input. For each rays, refractions in the crust according to incidence angle and azimuth are considered. First, absolute residuals relative to the IASP91 are corrected for the real crust deviations from the models, then the array-average residual, calculated from the crust corrected residuals of an event, is subtracted from the residuals at each station which recorded the event. The significant perturbations in cross-sections come from our plotting error, which we correct now in all the figures. Because the residuals are corrected for the deviations in the crust, they are assumed to represent pure mantle structure, under the condition that the crustal model is correct. Insufficient corrections map perturbations into the upper 100km of the mantle. On the other hand, an overcorrection can erase or substantially reduce positive perturbations in the upper 100 km (see the text modification in the last part of Section 6). We invert at node depths of 60-420 km (13 grid levels). We do not invert at 30km, neither at the deepest 450 a 480 km. Perturbations related to grid nodes at 60 km correspond to depth range of 45-75km. Therefore, our new images are plotted from depth of 45 km. We do not invert for the crust. The two deepest layers of nodes together with the nodes at shallower depths, where we also do not invert for velocity perturbations, surround the volume studied to stabilize the tomography.

*RC2: For a connection to crustal levels, one could plot the perturbations of the crustal model relative to a 1D-model of the crust into the vertical cross sections. But then, what to do with the anomalies that are already there?*

**Reply:** The original submission contains Fig. S3 with information on the crust. We added the Moho depth considered in the crustal corrections into cross-section in Fig. 3. There are no anomalies in the crust, as explained at several places and in our answer to the previous question, we corrected the plotting error.

*RC2: What is the criterion for gray-shading in the model domain? Is it derived from the resolution matrix?*

**Reply:** Boundary of the gray-shaded area follows a smoothed contour of diagonal elements of resolution matrix (RDE) equal to 0.15. The contour approximately separate the well and less well-resolved regions in the synthetic tests. This has been specified in the caption.

**Resolution Tests**

RC2: I would also like to see the results from a checkerboard test (with gaps) which nicely shows lateral and vertical smearing. In particular, vertical smearing at shallow levels should be investigated because it may hide a detachment of the slab or falsely connect the high vp anomaly below 100 km with a shallower one further to the south giving the impression of a significant northward dip.

**Reply:** We are aware of both the advantages and the disadvantages of the checkerboard test. We complement the Section 5 with the checkerboard test and include in the supplements a new Figure, S7a,b, as requested.

Besides the specific tests described above we also performed standard checkerboard tests to assess resolution capability of the network (Fig. S7- new). The checkerboard test confirms the positive and negative perturbations are retrieved well down at least to 240 km with a weak vertical smearing (Fig. S7a – horizontal slices). Also the vertical cross-section through the central part of the model (Fig.S7b) images the synthetic perturbations reliably.

RC2: *The remark that the polarity flip is more or less accepted is certainly a misconception. If I read Paffrath et al. (2021) correctly, they see a rather vertical, detached eastern Alpine slab and favour the interpretation of European provenance because its down-dip length can only be explained by the Tertiary shortening in the Eastern Alps accommodated by south-dipping subduction of European lithosphere. Basically, what is seen in the tomography is only the slab dip but tomography does not tell us the provenance of the slab. Even if it were dipping clearly northwards (in the tomographies by Mitterbauer (P4), Zhao and Paffrath, it is nearly vertical) it could still be overturned European lithosphere. Independent data are needed to decide this issue. The use of the term "polarity flip", however, already implies the interpretation of Adriatic provenance of the slab.*

**Reply:** Tomography images as referenced in our original ms. show steep northward-dipping high-velocity heterogeneity beneath the E. Alps within the upper mantle, down to ~250-300km. We interpret the images, especially in section of synthetic tests, not any concept, and we test the resolution of tomography images we get from available rays. In Section 6, we compare different tomography images, showing the northward dip, though they are interpreted differently. We agree

that additional information are needed to decide the issue. We modify our formulations from polarity flip/or reversal to a change of a slab dip direction (as also recommended by RC1) to avoid misunderstanding, though both these terms as used in literature in a broader sense.

*RC2: Resolution test 1 also mimics the real pattern quite well below 100 km depth while test 3 misses the increasing dip angle of the real pattern with depth. My remark regarding the treatment of crustal structure also applies to the resolution test. Since crustal corrections were subtracted before inversion of the real data, resolution test data should also be free of crustal contributions and anomalies in the crustal domain should be forced to zero.*

**Reply:** We see substantial differences between the real pattern and that of Test 1 and disagree that the Test 1 mimics the real pattern. Moreover, the test is designed to show that the rays are able to distinguish between one or two heterogeneities (lines 179-180 of the original submission). Resolution test are free of crustal contributions, anomalies in the crustal domain are zero. We correct the previous error in plotting and provide the corrected figures.

**Imaging the high-velocity perturbations in different tomography model:**

RC2: 2nd paragraph: If I read Paffrath et al (2021) correctly they do not postulate a polarity flip for the Central Alpine slab. They state a steep SE dip of the slab and also do not associate it with Adria. This should be corrected.

**Reply:** This question touches the same topic as the one raised by C1 concerning lines 222-224. Please see our answer there. We have never associated the Central Alps with Adria.

*RC2: In general, I find this section not well structured. The polarity flip issue is discussed in several paragraphs interrupted by a discussion of the 2$_{nd}$ high velocity anomaly underneath the BM. There seems to be no real ordering of thoughts and arguments. This part should be streamlined and rewritten in a concise and non-repetitive way with clear order and structure of arguments.*

**Reply:** We modify this section related to images of the E. Alps slab in different recognized tomography models. The main concerns are depths of the positive heterogeneities beneath the E. Alps and Bohemian Massif, dip directions and their provenance. These aspects are hard to separate completely, which can lead to a "no ordering of thoughts and arguments" opinion. We have improved the structure of this section.

RC2: I am strongly worried by the resolution tests shown in Fig. S7. For each one of the detached slab test models there are strong smearing artifacts reaching up to the surface. The artifacts are strongest in the crustal layer and they mimic an either northward or southward dipping continuous slab. I cannot see that this resolution test proves that a detachment can be resolved by the inversion. It rather suggests the opposite. I wonder how the result of these tests would look like if crustal perturbations were forced to zero. Possibly, all artificial positive anomalies between 50 km and 100 km could become much stronger pretending a dipping high velocity feature.

**Reply:** We apologize for the earlier plotting mistake, which we have explained in our answer above. The profiles are redrawn to correct the mistake in plotting, which created false perturbations above 45 km, and which we have not considered during the interpretation. There are no crustal perturbations in any figure. Moreover, we include better tests (new Fig.S7) to show whether our tomography is able or not to image the gap above the delaminated (European) slab beneath the E. Alps.

We also add into the main text a new Fig7, in which we mimic the observed perturbations with two (3% and 5%) heterogeneities, the northern one beneath the BM and southern one beneath the E. Alps. In all our interpretations we respect and follow the body-wave tomography principle that perturbations refer to model velocities within each "layer" (grid plane).

[Figure]

**Figure 7**: Velocity perturbations along the five vertical along-longitude cross-sections calculated from real data (upper row) and those (middle row) calculated from two synthetic 3% and 5% heterogeneities (bottom), plotted over the retrieved perturbations. Green dots mark Moho depths in the model used for calculation of crustal corrections (see also Fig. S3). The along-longitude cross-sections run from 51.65N to 45.35N.

[Figure]

**Figure S2b**: Histograms of uncertainties of the full dataset (left panel) and of events selected for tomography (right panel). Mean and median of uncertainties, width of quality bins as well as uncertainty of the maximum of picks are indicated by vertical lines. The uncertainties are categorized into five quality classes ranging from 1 (the best) to 5 (the worst). Only arrival times measured with quality 1-3, with the mean 0.08s and median 0.07s, input the tomography.

[Figure]

**Figure S7:** Velocity perturbations along the N-S cross-section in the center of the array from real data (upper row) and from synthetic data (middle row) calculated for models of the steep detached slab beneath the E. Alps (lower row). The top of the heterogeneity migrates upward (bottom row) from 150 km (detachment as in Paffrath et al., 2021) to 60 km (representing no detachment). The slab detachment larger than the 30km grid would be revealed in the upper 200 km of the EASI-AA model. A potential leakage does not overprint the images.

.

**Figure S8:** Pairs of horizontal slices (a) and vertical cross-section (b) through the checkerboard model (right images in the pairs) and retrieved perturbations (left images in the pairs), plotted for all inverted node levels. The same mask as in Fig. 3 is used for shading in part (b).

[Figure]

Figure 8: continuation, part (b)

---

## Referee Report (RR2)

**Review Plomerova et al., SE-2021-56, first revised version**

While several remarks of my first review were appropriately dealt with, there are still remaining issues regarding crustal correction, inversion procedure, resolution tests and interpretation.

**Introduction:**

There is one strange sentence in the introduction ("We show that thanks to data from the AlpArray-EASI ….") which should be reformulated. I also do not understand the relation between the HV-BM and the HV-anomaly found by Kästle et al. located 1 degree north of the PAL.

**Data:**

Line 94: A low-noise beam is formed by stacking cross-correlated time shifted traces. To do that some reference is needed relative to which the traces are correlated and shifted. So why is there no subjective choice of a reference trace?

Line 106: "aside" -> outside.

Line 108: Does this mean that the dataset with evenly distributed events was only constructed to show that the one with events in the 60 degree cones provides sharper images?

In Fig. S5, the velocity perturbations still reach up into the corrected crustal domain.

Fig. S3: The authors show Moho thickness as well as velocity and thickness of sediments but not velocities in the crust used for crustal corrections. And they only show these parameters for the station locations. The authors should show one vertical section along the EASI line with the assumed crustal velocities. They should also document in the manuscript (not only in the Reply) how the crustal corrections are computed. Is the 30 km gridding also used for calculating crustal corrections or is a more finely resolved model used. If the 30 km gridding is used, is the crustal model which certainly shows smaller-scale structures smoothed to the much coarser gridding? And finally, why not plot the a priori crustal perturbations relative to IASP91 into one or several of your vertical sections in addition to the Moho?

It is stated that the travel-time residuals are normalized to the array average. I understand from the reply that the average residual is subtracted. Normalisation would imply division which does not make much sense in this context.

**Method:**

Some explanations regarding the forward problem from the teleseismic source to the receivers would be helpful. How are travel times calculated? How is the region outside the inversion domain treated, how is the transition into the inversion domain managed?

It could be mentioned in the text that the inversion code does not offer an option for vertical smoothing.

I understand that the model domain is made of grid nodes at 30 km depth intervals and, horizontally, by the yellow and green nodes in Fig. 1. It is stated that the data are only inverted for velocity perturbations at the green nodes and not at the yellow nodes. My question is: what is then the purpose of the yellow nodes? Why are they needed at all? Is the velocity at the yellow nodes fixed to the values of the reference model? Moreover, would it not be favourable to allow perturbations at the yellow nodes to avoid mapping of heterogeneities into the central region (green nodes) due to rays which spend a significant path length in the domain of the yellow nodes, especially at greater depths? There are many of these as Fig S1b shows. I urgently recommend doing that because all ignored heterogeneities lying in the region of the yellow nodes will produce artificial perturbations at the green nodes. Are there contributing rays that propagate even outside the area covered with nodes?

Why do the authors still plot data variance instead of misfit which is normalized to the data uncertainties. In the caption of Fig. S4 it is stated that the data uncertainties are included in the calculation of the data variance. But how? If the residuals are divided by the uncertainty a dimensionless quantity results. Why do the authors not simply plot misfit over N defined as $1/N*sum((d-s)/sigma)**2$ as is standard in tomographic work?

Why was the inversion stopped after only 2 iterations? What happens if more iterations are done? Please enter some results for further iterations into Fig. S4.

Instead of plotting data variance versus model variance, the authors should plot misfit over N versus model roughness as the latter is used to regularize the inversion and the trade-off occurs between misfit and model roughness (and not model variance).

Results:

What is the criterion for grey-shading in the model domain? Is it derived from the resolution matrix? You give the information in the reply but I do not find it in the manuscript. The caption of Fig. 3 just says that less-well resolved regions are shaded. This is not very informative.

In Fig. 3 the HV anomalies are surrounded by thick dashed lines. What was the criterion for placing these lines. They do not seem to follow a velocity contour. In Fig. 3 the dashed lines do not honour the small shallow area of less positive or negative perturbations which splits the HV-EA anomaly down to about 100 km. This area nicely coincides with the depth maximum of the Moho. In the interpretation later, the entire HV-EA is attributed to the Adriatic plate. Wouldn't it make more sense to attribute the left part of HV-EA to Europe and the right part to Adria given the shape of the Moho there?

Why did the authors saturate the colour scale? I prefer unsaturated scales.

Resolution tests:

In Fig. S6a-d, the vertical sections through the model obtained from real data shows (shaded) perturbations in the crust. I thought that the first layer of nodes at 30 km was not involved in the inversion.

The evaluation of the checkerboard resolution test is a bit optimistic. The vertical section shows oblique smearing owing to the ray geometry which could nicely mimic dipping slabs. Given this result and the massive artefacts in the test shown in Fig. S8, I would not be that confident in the difference of the dip directions as stated in line 225.

Imaging the high-velocity perturbations in different tomography model:

The paper by Paffrath et al. (2021) is erroneously referenced two times as Paffrath et al. (2020).

The HV-BM anomaly is described as trending SW-NE. In Fig.2 I rather see a NW-SE trend of the anomaly at 120 km and 150 km depth that seems to rotate to SW-NE at 180 km depth.

I am seriously worried by the resolution tests shown in Fig. S8. For each one of the detached slab test models there is massive leakage reaching up to the surface which mimics an either northward or southward dipping continuous slab. Apparently, it is hard to distinguish between a slab reaching up to 45 km and a detached one with top as deep as 150 km! Inversion for test models all deliver a shallow north-dipping HV anomaly. In view of Figure S8, I am not at all convinced of the author's arguments for northward dip and for attachment of the slab. I would be interested in explanations for this massive leakage.

This leakage also affects the HV-BM anomaly as shown in Fig. 7. Moreover, the test in Fig. 7 cannot reproduce the shallow splitting of the HV-EA. What would be the result of a resolution test which takes the configuration in Fig. 14c of Paffrath et al. 2021 as test model (a fast shallow Adriatic lithosphere and a detached EU slab)?

---

## Author Response (AR2)

*Dear Editor, dear Emanuel,*

*Many thanks to you and both Referees for their work on our manuscript. We have now carefully revised all points that have been raised, and made changes to the manuscript together with replying to the comments below in italics. We appreciate the second reviewer encouraged us to include our thoughts on a possible dual origin of the high-velocity heterogeneity beneath the Eastern Alps.*

*We truly hope that the revised version will be to your satisfaction, and if you find further points to fix, please let us know.*

*Best regards,*

*Jarka, Helena, György and Ludek*

Reply to Referee 1 letter and commented reply to previous revision.

Dear Editor, dear Emanuel,

The authors have made a big effort to amend the text and figures. This has made their article much more accessible for the AlpArray and the earth science community, to which they have contributed so much.

Yet, there are first-order issues that were addressed in the first review that still need fixing and clarification. These are commented on in annotations to the authors´response to the editor.

The statement in lines 245-247 is still not correct. As a co-author of Paffrath et al. (2021) and its companion interpretational paper (Handy et al. 2021), I can say for sure that the slab anomaly in our images across the Central Alps dips to the SSE, not to the N (please see Fig. 14b in Paffrath et al. 2021, as well as many vertical sections in Handy et al. 2021, e.g., their Fig. 3c, sections B and 6 in their Appendix). In fact, the S-dipping positive anomaly is also clear in our TRANSALP section (their Fig. 4A). This latter section is across the westernmost part of the Eastern Alps, just east of the Giudicarie-Brenner Fault and certainly not in the Central Alps. At the risk of being forward, I have proposed a formulation in the annotation.

*As recommended, we corrected the text according to Paffrath et al. (2021) and Handy et al. (2021) papers and deleted reference to Paffrath et al. (2020) presentation.*

Lines 250-252: Perhaps it would be useful for the authors to state before (rather than after) lines 245-247 that they place the boundary between the Central Alps and Eastern Alps at the western end of the Tauern Window, i.e., at the northward prolongation of the Giudicarie Fault, as already proposed by Handy et al. (2015) and other more recent authors to separate different crustal (lower crustal wedges) and mantle (slab gap, slab dips) structures.

*The paragraph has been reorganised and now reads as:*

Though the upper mantle structure is less diverse in comparison with the crust, in general, ongoing studies of the Alpine upper mantle continue to reveal new and more detailed features in geometry of the lower lithosphere, dip direction of the Alpine slabs, tears or detachments of the slabs and interactions of the Alps with the Apennines and Dinarides. The current stage of knowledge from results of various disciplines – seismology, geology, petrology, tectonics, paleomagnetism, geochemistry, GPS studies etc. - reflect differences in the segmented slab responses to the acting forces. The complex structure of the fragmented Alpine slab(s) and the broader Europe/Adria collision zone is now visualized in tomography snapshots. The boundary between the central and eastern slab segments of the Alps, is placed at the western end of the Tauern Window, i.e., at the northward prolongation of the Giudicare Fault, as already proposed by Handy et al., (2015) and other more recent authors to separate different crustal (lower crustal wedges) and mantle (slab gaps, slab dips) structures (e.g, Lippitsch et al., 2003; Schmid et al., 2004; Rosenberg et al., 2018). In recent studies, Paffrath et al. (2021) and Handy et al. (2021) document a change from a S-dipping slab to a N-dipping slab beneath the western part of the Eastern Alps (i.e., beneath the Tauern Window). This is in agreement with the previously proposed geometries of Babuska et al. 1990 and later by Lippitsch et al. 2003 and Zhao et al. 2016. Mock et al. (2020) note a discordance between the slab geometry at depth and the boundary between the Eastern and Central Alps observed at the surface.

Lines 373-375: Instead of "…as suggested by Handy et al. (2015)…", I would substitute the words "…, analogous to the scenario proposed by Handy et al. (2015) for delamination of European lithosphere further to the S beneath the Eastern Alps (Fig. 8a)." This would distinguish the models more clearly.

*Text was modified as suggested:*

The simplest explanation would be to consider it as a fragment of the delaminated part of the European plate subductions, analogous to the scenario proposed by Handy et al. (2015) for delamination of European lithosphere further to the South beneath the Eastern Alps (Fig.8a).

Lines 404-409: For the purposes of their text, I would recommend that the authors cite Royden & Burchfiel (1989, Tectonics, 8, 1, 51-61) and/or Royden (1993, Tectonics, 12, 629-638) instead of Kissling & Schlunegger (2015). Although both papers use the term "rollback subduction", they invoke quite different mechanisms. The Carpathians-Pannonian system is generally regarded as a prime example of oceanic and continental subduction in the absence of significant plate convergence, as discussed in numerous papers of Frank Horvath, Wiki Royden & Clark Burchfiel (ob cit). The consequence of this is that significant upper plate (Pannonian) extension and mantle flow are required to accommodate lateral retreat of the downgoing (European) plate in a mantle reference frame. I believe this is exactly (or close to) what Jarka Plomerova and her co-authors mean in their discussion of "complex flows of the mantle". Note that this is not identical to the mechanism proposed by Kissling & Schlunegger (2015) which involves minor slab-hinge

retreat due to slab steepening at the very end of continent-continent collision. The amount of mantle flow in their model is minimal, if not negligible, at least compared to the model of Royden et al. It is rather unfortunate that Kissling and Schlunegger (and in their papers since) also termed this rollback subduction (I would have called it "late orogenic slab steepening").

*Thank you for this good suggestion; the reference used (Kissling & Schlunegger, 2015) was substituted by the two new references:*

Differences in the roll-back subductions of the Alps (e.g., Royden and Burchfiel, 1989; Royden 1993) and the Carpathians, northward push of Adria and European slab delamination beneath the EA could have formed complex flows in the asthenosphere (e.g., Vignaroli et al., 2008), which could "transport" a purely oceanic lithosphere or a mix of oceanic and continental lithosphere fragments through the open space between the E. Alpine and Carpathian slabs north-northeastward into the mantle beneath the BM.

Figures: The labelling is much improved, but still falls short of desirable, as commented in the annotations. This is the now the editor`s prerogative to decide how much is enough. Surely, plotting boundaries on a map to help locate anomalies with respect to the surface geology is much less arduous that processing seismic data to determine the anomalies themselves! The benefits for the community would be immense. The trace of the cross section in Fig. 8A should be plotted on a map, as recommended already in my first review (though please excuse me if I have overlooked them). It would also be a great help to have longitudinal coordinates above the cross section in this figure.

*We continue keeping tectonic boundaries and do not plot political boundaries as they have no geophysical meaning in map views with sufficiently dense geographical coordinates. We modified caption of Fig. 8a (there are no exact coordinates in Handy et al., 2021). In the first revision we chose the proposed option and included the coordinates into caption of cross-sections through our tomography, not to overload the figures with labels.*

The manuscript can be accepted after minor revisions along the lines recommended above.

**Reply on "commented on in annotations to the authors´response"**

*We have incorporated the overwhelming majority of suggested changes in the manuscript (and do not report all of them here unless there is a thought to add); similarly, other very tiny changes are in the manuscript as well.*

May I suggest the following minor change to make it even clearer:

"...imaged two segments of Alpine slab, one dipping S beneath the Western Alps and another dipping N beneath the Eastern Alps, with a gap between them." Note: "slab" instead of "slabs" -plural- which can be misinterpreted to mean that two slabs were each separated into segments.

*Text modified as suggested.*

*The word "keel" is no more used to avoid any potential misunderstanding.*

HV-BM   HV-EA

Ok, fine. The readers will follow you if you explain the terms like you do here.

*The text where these abbreviations are introduced has been coplemented.*

Dip change C./E. Alps

*Reference to Paffrath et al. 2020 was deleted and text modified as recommended.*

*Recommended references for C/E Alps boundary added.*

Ad (2) triple junction (Brückl)

*This was a heading in our reply, with which you agreed, to your part 2 of the paragraph:*

*The suggested formulation "….speculating the anomaly beneath BM could be derived from subduction and lateral transport in the mantle of the oceanic embayment of Alpine Tethys."  was incorporated.*

Regarding Fig. 8A (cross sections), the authors appear to have ignored the recommendation to show the location of the cross sections on a map. This would be necessary to check the superposition of the anomalies that are compared (Paffrath, this work, model of Handy et al. 2015). One asks oneself if this science is reproducible...

Note that the anomaly labelled EU in the cross section needs an equivalent in the legend (is it labeled EU-BM in the caption?).

*We complemented the figure, its legend and caption. We do not have the proper coordinates of the profiles from Handy et al (2015) as taking it from their figure 10 would be very difficult and imprecise; instead, we refer to their figure 10 for the location. The sizes of the high-velocity heterogeneities shown (contours) are modulated to fit the depth and lateral scales according to the NCA, TW, PL locations.*

Reply to Referee 2 comments.

**Introduction:**

There is one strange sentence in the introduction ("We show that thanks to data from the AlpArray-EASI ….") which should be reformulated. I also do not understand the relation between the HV-BM and the HV-anomaly found by Kästle et al. located 1 degree north of the PAL.

*This is now clarified in the reformulated the text:*

Based on data from the AlpArray-EASI (Hetényi et al., 2018b) and AlpAray Seismic (AASN) networks (AlpArray Seismic Network, 2014; 2015; Hetényi et al., 2018a), our tomography shows a small-size high-velocity heterogeneity at ~100 - 200km depths beneath the south-eastern part of the BM (referred to as HV-BM throughout the paper), sub-parallel to and distinct from the E. Alps high-velocity heterogeneity (HV-EA). Kästle et al. (2018) identified in their surface-wave tomography approximately 1° to the north of the Periadriatic Fault a high-velocity heterogeneity similar, similar to HV-BM, but further to the west.

**Data:**

Line 94: **A low-noise beam i**s formed by stacking cross-correlated time shifted traces. To do that some reference is needed relative to which the traces are correlated and shifted. So why is there no subjective choice of a reference trace?

*Answer: The reason for that is simply to avoid subjectivity in the selection of the reference trace.*

*Text in revision 1:*

A low-noise beam trace created from stacked cross-correlated and shifted traces of an event serves as a reference in the second cross-correlation step and beam forming in the P-arrival time picking.  This means, there is no subjective a priori selected single reference trace.

*Modification for Revision 2:*

Instead of a subjective selection of a reference trace, we cross-correlate all pairs of traces, in the first step. For each trace a time shift related to maxima of the cross-correlation function is determined. Traces  at individual stations are then shifted by a weighted average of the time-shifts gained from cross-correlations with all other stations which recorded the event. The low-noise beam trace, created as a stack of the shifted traces at all stations, form the reference trace for performing new beam forming and arrival time picking of an event in the second cross-correlation step.

In the second step, we correlate traces of all stations with the reference low-noise beam trace. The waveforms are aligned according to the times related to the maxima of the cross-correlation functions. The new low-noise beam is computed as a median of all aligned traces and the P-wave onset on the beam is determined automatically (see Fig. S2). Then, arrival times at individual

stations are derived from differences between times of corresponding extremes. Arrival times on the station signals are measured by three different methods. The final times of individual extremes (green P1 and P2 in Fig. S2a) and their error estimates are computed from the normal distribution, which approximates a mixture of normal distributions of partial picks (for details see Fig. S2a).

############

Line 106: "aside" -> outside.

*Answer:*

*We understand "outside" as generally external to the region of interest, whereas "aside" here refers to the area that is adjacent and just out of the longer sides of the region, i.e., west and east of the regions (along Alpine strike structural variations). We therefore will keep aside in the text here.*

#############

Line 108: Does this mean that the dataset with evenly distributed events was only constructed to show that the one with events in the 60 degree cones provides sharper images?

*Answer:*

*We have explained in the text the reasons for limiting data to 60 degree wide cones:*

*"To eliminate mapping effects of heterogeneities **aside** the model (meaning adjacent and along the long sides of the array) into its internal part and to enhance the resolution in direction of the subductions, we selected additional rays coming from the northern and southern 60° wide azimuth bins."*

*The last sentence of the paragraph was modified as:*

Figure S5 demonstrates that potential bias coming from heterogeneities west and east of the N-S oriented elongated array is weak and that the events from the 60deg. cones illuminate better the Bohemian Massif (BM) - Eastern Alps (EA) structures.

#########

In Fig. S5, the velocity perturbations still reach up into the corrected crustal domain.

*We apologize to have forgotten to correct the plotting error in the supplementary Fig. S5. This is now corrected and this figure is compatible with the figures in the main text.*

############

Fig. S3: The authors show Moho thickness as well as velocity and thickness of sediments but not velocities in the crust used for crustal corrections. And they only show these parameters for the station locations. The authors should show one vertical section along the EASI line with the assumed crustal velocities. They should also document **in the manuscript** (not only in the Reply) how the crustal corrections are computed. Is the 30 km gridding also used for calculating crustal corrections or is a more finely resolved model used. If the 30 km gridding is used, is the crustal

model which certainly shows smaller-scale structures smoothed to the much coarser gridding? And finally, why not plot the a priori crustal perturbations relative to IASP91 into one or several of your vertical sections in addition to the Moho?

*Answer: We have rewritten the last paragraph of Data section to address all these queries:*

Teleseismic data cannot resolve velocities in the crust itself due to their sub-vertical propagation at shallow depths. To avoid mapping effects from the crust into the velocity perturbations in the upper mantle (e.g., Karousová et al., 2012), one has to introduce crustal corrections. Unfortunately, up to now, there is no uniform, sufficiently detailed model of the crust for Europe, neither for the AlpArray region. For our body-wave studies in Europe we have collected accessible information on the crust for each station from different sources: from Karousová et al. (2012, and references therein) for the BM mostly based on results of control source seismics; from, e.g., Di Stefano et al. (2011), Hua et al., (2017), Tesauro et al.(2008) for areas south of the BM; and from Hetényi et al. (2018b) along the EASI transect. The crust is characterized beneath each station by the depth of Moho, the dip angle and dip direction if the Moho is not flat, the Moho jump if there is any, velocity in the crust, thickness of sediments and sediment velocity (Fig. S3). Sometimes there are significant differences between different models, in their overlapping parts, therefore, we do not attempt creating any kind of „fine"-gridded model of the crust. Instead, we tune the corrections individually beneath each station and correct the travel time residuals along each individually traced ray for the difference in the „real" crust and the crust of the reference model. Carefully pre-processed P-wave travel-time residuals calculated relative to the IASPEI'91 velocity model (Kennett and Engdahl, 1991), corrected for the crust, normalized to the average residual per event and cleaned from outliers serve as input to the inversion in which we do not invert for the crust. With this approach we gathered a high-quality and uniform dataset of travel time residuals for a proper tomographic inversion to resolve structures in our target region and below the crust. Of course, all crustal models remain idealized models and aren't hundred percent correct. Therefore, small "uncorrected" perturbations remain in each teleseimic tomography model of the upper mantle, which one needs to consider when interpreting the results.

#########

It is stated that the travel-time residuals are normalized to the array average. I understand from the reply that the average residual is subtracted. Normalisation would imply division which does not make much sense in this context.

*Answer:*

*The term "normalization" is used since decades (even since pre-digital recording era) for computing relative travel-time residuals. The relative residuals can be calculated from absolute travel times relative to (1) an array travel time average, or to (2) the travel time of a reference station, or to (3) an average travel time of a subset of stations. Each of the procedures has its advantages and disadvantages, which are reflected in the accuracy of the relative residuals as well as the stability of the reference level (see e.g. Babuska and Plomerova, 1992). Regardless of*

*the choice of this reference level, it is always subtraction and never division that is applied in this type of normalization.*

######

**Method:**

Some explanations regarding the forward problem from the teleseismic source to the receivers would be helpful. How are travel times calculated? How is the region outside the inversion domain treated, how is the transition into the inversion domain managed?

*Answer:*

*It is unusual to describe basic information on traveltime calculation in each scientific paper. They can be found in any tomography text book, e.g. "Seismic Tomography: Theory and Practice", by Iyer and Hirahara. Regarding the region outside the inversion domain: Each teleseismic regional tomography has to deal with heterogeneities outside the studied volume. It is standardly done by the normalization as described above. Transition into the inversion domain: Bottom entrance to the volume is fixed according to the reference velocity model and then the ray is traced according to ray-bending technique by Steck and Prothero (1991). The forward task is solved only within the region studied.*

################################

It could be mentioned in the text that the inversion code does not offer an option for vertical smoothing.

*Answer:*

*Sentence incorporated: „There is no vertical smoothing in the code."*

*Comment: Because teleseismic body-wave tomography optimizes velocity perturbations in horizontal layers, and because the ray geometries may already cause sub-vertical leaking, vertical smoothing in teleseismic tomography is generally not useful and not applied.*

#################

I understand that the model domain is made of grid nodes at 30 km depth intervals and, horizontally, by the yellow and green nodes in Fig. 1. It is stated that the data are only inverted for velocity perturbations at the green nodes and not at the yellow nodes. My question is: what is then the purpose of the yellow nodes? Why are they needed at all? Is the velocity at the yellow nodes fixed to the values of the reference model? Moreover, would it not be favourable to allow perturbations at the yellow nodes to avoid mapping of heterogeneities into the central region (green nodes) due to rays which spend a significant path length in the domain of the yellow nodes, especially at greater depths? There are many of these as Fig S1b shows. I urgently recommend doing that because all ignored heterogeneities lying in the region of the yellow nodes will produce

artificial perturbations at the green nodes. Are there contributing rays that propagate even outside the area covered with nodes?

*Answer:*

*Basic principles of body-wave tomography say that only blocks/grid-node surroundings with crossing rays can be considered as resolved. Which is not the case of the yellow points, but those grid nodes are still used in ray tracing in the entire volume.*

*The question in the last sentence is not clear, unless the answer is a trivial "no". In teleseismic tomography with foci outside the region, only rays entering through the bottom of the volume contribute to computation, contributions from ray-paths "outside the area" are not considered.*

##################

Why do the authors still plot data variance instead of misfit which is normalized to the data uncertainties. In the caption of Fig. S4 it is stated that the data uncertainties are included in the calculation of the data variance. But how? If the residuals are divided by the uncertainty a dimensionless quantity results. Why do the authors not simply plot misfit over N defined as 1/N*sum((d-s)/sigma)**2 as is standard in tomographic work?

*Answer:*

*The formula for evaluation of data variance in the code is similar to what you write in your comment. It is 1/(N-1)*sum((residual - average of residuals)*weight)**2, i.e., instead of dividing by sigma, the numerator is multiplied by a unitless weight. The weight of each travel time residual depends on the measurement uncertainty, determined during the arrival time picking (see the decription in the text).*

*Caption of Fig.S4 is modified.*

##########################

Why was the inversion stopped after only 2 iterations? What happens if more iterations are done? Please enter some results for further iterations into Fig. S4.

*Answer:*

*In our response to the previous revision we have written that we performed one more iteration, but overall imaged perturbations remained without significant changes, therefore we decided to stop calculations after the second iteration to save the time.*

#########################

Instead of plotting data variance versus model variance, the authors should plot misfit over N versus model roughness as the latter is used to regularize the inversion and the trade-off occurs between misfit and model roughness (and not model variance).

*Answer:*

*We plot the trade-off curve in the same way as numerous authors of tomography models using the codes Telinv or AniTomo, e.g., Lippitsch et al. 2003; Sandoval et al. 2004; Shomali et al. 2006;*

*Eken et al. 2007; Karousová et al. 2012; 2013; Plomerová et al. 2016; Silvennoinen et al. 2016; Chyba et al. 2017; Munzarová et al., 2018b. This enables a closer comparison with their results.*

####################################

**Results:**

What is the criterion for grey-shading in the model domain? Is it derived from the resolution matrix? **You give the information in the reply but I do not find it in the manuscript.** The caption of Fig. 3 just says that less-well resolved regions are shaded. This is not very informative.

*Answer:*

*Yes, the shading of less-well resolved regions was set according to the resolution matrix, ray coverage grid cells and derivatve weight sums. The contour follows the value of 0.15 of diagonal elements of the resolution matrix.*

*Caption of Fig. 3  is complemented accordingly.*

#############################

In Fig. 3 the HV anomalies are surrounded by thick dashed lines. What was the criterion for placing these lines. They do not seem to follow a velocity contour. In Fig. 3 the dashed lines do not honour the small shallow area of less positive or negative perturbations which splits the HV-EA anomaly down to about 100 km. This area nicely coincides with the depth maximum of the Moho. In the interpretation later, the entire HV-EA is attributed to the Adriatic plate. Wouldn't it make more sense to attribute the left part of HV-EA to Europe and the right part to Adria given the shape of the Moho there?

*Answer:*

*We have deleted the contours in Fig.3, which followed smoothly the 0% perturbations, and have newly denoted the high-velocity heterogeneities as labels using Roman numerals I, II, III (the text of the 3rd para. of the Results section has been modified accordingly).  We have also added a new paragraph at the end of the section pointing to the possible dual source of the high-velocity heterogeneity beneath the E. Alps. Corresponding tiny modifications are in the Abstract and in the Conclusions.*

*New text:*

Looking at perturbations of the HV-EA heterogeneity at greater details, one can recognize its potential dual source. The positive heterogeneities I and II (Fig. 3) are separated just beneath the TW, where we modelled the European and Adriatic crust contact (Hetényi et al., 2018b). Dip directions of the heterogeneities slightly change – HV-EA-part II seems to immerse southward (Fig.3a,b),  whereas  HV-EA- part I dips to the north. This allows us to argue for a mixing of a detached EU slab fragment and the shallow Adria slab connected at depth, as already suggested in Babuška et al. (1990) using a much coarser model inferred from P residuals.

############################################

Why did the authors saturate the colour scale? I prefer unsaturated scales.

*A saturated scale in mapping velocity perturbations is frequently used. The reason is not to mask variations in a real range of perturbations by possible outliers, mostly at the edges of the region. In our example here, we have inverted at 1625 nodes, out of which only 6 % exceed the (-4, 4%) dv range with the applied damping.*

##############################################

**Resolution tests:**

In Fig. S6a-d, the vertical sections through the model obtained from real data shows (shaded) perturbations in the crust. I thought that the first layer of nodes at 30 km was not involved in the inversion.

*Answer:*

*The same as for Fig.S5: the first layer of nodes at 30km was not involved in the inversion. We apologize but forgot to correct a plotting mistake in the supplementary figures, including FigS6a-d. Now these are all corrected.*

############################################

The evaluation of the checkerboard resolution test is a bit optimistic. The vertical section shows oblique smearing owing to the ray geometry which could nicely mimic dipping slabs. Given this result and the massive artefacts in the test shown in Fig. S8, I would not be that confident in the difference of the dip directions as stated in line 225.

*Answer:*

*Checkerboard tests are always requested, but inversions with targeted synthetic anomalies, as e.g., in our Figs. 4 and S6a-d, document the specific resolution and provide more plausible grounds for evaluating the results (E. Kissling, personal communication).*

*There is some leakage in the presented tomography, but it is far from producing massive artefacts.*

##################################################

**Imaging the high-velocity perturbations in different tomography model:**

The paper by Paffrath et al. (2021) is erroneously referenced two times as Paffrath et al. (2020).

*Answer:*

*Reference to Paffrath et al. (2020) has been erased thorough the text and in References.*

####################################

The HV-BM anomaly is described as trending SW-NE. In Fig.2 I rather see a NW-SE trend of the anomaly at 120 km and 150 km depth that seems to rotate to SW-NE at 180 km depth.

*Answer:*

*We do not derive the trend of the HV-BM anomaly only from its shape retrieved in our tomography itself, as it would be speculative due to its size and the array width, let alone to speculate about a change with depth. We derive the trend from the comparison with two larger tomography results (Karousova et al., 2013; Paffrath et al., 2021). The text stays as:*

*Considering the NE continuations of the HV-BM as imaged in body-wave tomography of a larger extent (e.g., Karousová et al., 2013; Paffrath et al., 2021), the heterogeneity strikes with the SW-NE trend, in parallel with the boundary of the Moldanubian (MD) and Brunovistulian (BV) mantle lithosphere in the BM, and the westernmost part of the Carpathian front.*

##############################

I am seriously worried by the resolution tests shown in Fig. S8. For each one of the detached slab test models there is massive leakage reaching up to the surface which mimics an either northward or southward dipping continuous slab. Apparently, it is hard to distinguish between a slab reaching up to 45 km and a detached one with top as deep as 150 km! Inversion for test models all deliver a shallow north-dipping HV anomaly. In view of Figure S8, I am not at all convinced of the author's arguments for northward dip and for attachment of the slab. I would be interested in explanations for this massive leakage.

*Answer:*

*Due to leakage, which exists in each teleseismic P-wave tomography, one has to be careful when interpreting and concentrate only on distinct features. Figure S8 demonstrates that the dark blue perturbations in the model with shallow heterogeneity reproduce the dark blue perturbations in the model from real data better than the model with heterogeneity below 150 km depth. This test aims at verifying whether our tomography is able to image and distinguish between an attached and a detached heterogeneity, and is not targeting the dip. Tests focusing on the respective dips of the heterogeneities are presented in Figs. 4 and S6a-d.*

*Text complemented as:*

Due to the leakage, which exists in each teleseismic P-wave tomography, one has to be careful and concentrate only on distinct features. Figure 8 demonstrates that the dark blue perturbations in the model with shallow heterogeneity reproduce the dark blue perturbations in the model from real data better than the model with heterogeneity below 150 km depth.

############################################

This leakage also affects the HV-BM anomaly as shown in Fig. 7. Moreover, the test in Fig. 7 cannot reproduce the shallow splitting of the HV-EA. What would be the result of a resolution test which takes the configuration in Fig. 14c of Paffrath et al. 2021 as test model (a fast shallow Adriatic lithosphere and a detached EU slab)?

*Answer:*

*Fig. 7 shows that leakage of the model perturbations of the HV-BM towards the surface is weak and acts against the negative perturbations imaged from real data. There is no surprise that a simple prism model cannot reproduce details of the complex HV-EA heterogeneity.*

*We have performed a variety of resolution tests, and continuing could be an endless process. The region of the Eastern Alps is complex and tomography studies will definitely continue investigating this area. We will consider suggested tests, as well as special tests focused on effects of crustal corrections in future work.*

Text complemented:

The leakage of the model perturbations of the HV-BM towards the surface is weak and acts against the negative perturbations imaged from real data. There is no surprise that a simple prism model cannot reproduce tiny details of the complex HV-EA heterogeneity.

##############################################################